

# Regulation of N₂O emissions from acid organic soil drained for agriculture: Effects of land use and season

**Arezoo Taghizadeh-Toosi[1], Lars Elsgaard[1], Tim J. Clough[2], Rodrigo Labouriau[3], Vibeke Ernstsen[4] and Søren O. Petersen[1]**

[1] Department of Agroecology, Aarhus University, Tjele, Denmark

[2] Faculty of Agriculture and Life Sciences, Lincoln University, Christchurch, New Zealand

[3] Applied Statistics Laboratory, Department of Mathematics, Aarhus University, Aarhus, Denmark

[4] Geological Survey of Denmark and Greenland, Copenhagen, Denmark

*Correspondence to*: Arezoo Taghizadeh-Toosi (Arezoo.Taghizadeh-Toosi@agro.au.dk)

## Abstract

Drained organic soils are extensively used for cereal and high-value cash crop production or as grazing land, but emissions of nitrous oxide (N₂O) are enhanced by the drainage and cultivation. A study was conducted to investigate the regulation of N₂O emissions in a raised bog area drained for agriculture. The area has been classified as potentially acid sulfate soil, and we hypothesised that pyrite (FeS₂) oxidation was a potential driver of N₂O emissions through microbially mediated reduction of nitrate (NO₃⁻). Two sites with rotational grass, and two sites with a potato crop, were equipped for monitoring of N₂O emissions, as well as sub-soil N₂O concentrations at 5, 10, 20, 50 and 100 cm depth, during spring and autumn 2015. Precipitation, air and soil temperature, soil moisture, water table (WT) depth, and soil mineral N were recorded during weekly field campaigns. In late April and early September, intact cores were collected to 1 m depth at adjacent grassland and potato sites for analysis of soil properties, which included acid volatile sulfide (AVS) and chromium-reducible sulfur (CRS) to quantify, respectively, iron monosulfide (FeS) and FeS₂, as well as total reactive iron (TRFe) and nitrite (NO₂⁻). Soil organic matter composition and total reduction capacity was also determined. The soil pH varied between 4.7 and 5.4. Equivalent soil gas phase concentrations of N₂O ranged from around 10 μL L⁻¹ at grassland sites to several hundred μL L⁻¹ at potato sites, in accordance with lower soil mineral N concentrations at grassland sites. Total N₂O emissions during 152-174 days were 3-6 kg N₂O-N ha⁻¹ for rotational grass, and 19-21 kg N₂O-N ha⁻¹ for potato sites. Statistical analyses by graphical models showed that soil N₂O concentration in the capillary fringe was the strongest predictor for N₂O emissions in spring, and for grassland sites also in the autumn. For potato sites in the autumn, nitrate (NO₃⁻) availability in the top soil, together with temperature, were the main controls on N₂O emissions. Pyrite oxidation coupled with NO₃⁻ reduction could not be dismissed as a source of N₂O, but the total reduction capacity of the peat soil was much higher than explained by the FeS₂ concentration, and the concentrations of TRFe were much higher than pyrite concentrations. The potential for chemodenitrification being a source of N₂O during WT drawdown in spring is discussed. In contrast, the N₂O emissions associated with rapid soil wetting and WT rise in autumn were consistent with biological denitrification. Soil N availability and seasonal WT changes were important controls of N₂O emissions.



**Key words:** Drained peat, potentially acid sulfate soil, rotational grass, potato, nitrous oxide, reactive iron


## 1 Introduction

Worldwide, 25.5 million ha of organic soils have been drained for agricultural use, mainly as cropland (Tubiello et al., 2016), which accelerates decomposition of soil organic matter and net carbon (C) and nitrogen (N) mineralisation above the water table (WT) (Schothorst, 1977). Drained organic soils are significant net sources of greenhouse gas (GHG)

emissions due to emissions of carbon dioxide ($CO_2$) and nitrous oxide ($N_2O$) (Goldberg et al., 2010; Maljanen et al., 2003). A recent supplement to the 2006 IPCC Guidelines for National Greenhouse Gas Inventories on Wetlands (IPCC, 2014) proposed average annual emission factors of 4.3 and 8.2 kg $N_2O$-N ha$^{-1}$ yr$^{-1}$ for temperate grassland on drained organic soil with low and high nutrient status, respectively, and an emission factor of 13 kg $N_2O$-N ha$^{-1}$ yr$^{-1}$ for cropland. For soil C losses, the emission factors proposed for the three land use categories were between 5.3 and 7.9

Mg $CO_2$-C ha$^{-1}$ yr$^{-1}$ (Hiraishi et al., 2014). This implies that site conditions are potentially more critical for $N_2O$ than for $CO_2$ emissions.

Site conditions are defined by land use, management, inherent soil properties and climate (Mander et al., 2010; Leppelt et al., 2014). Both WT drawdown (Aerts and Ludwig, 1997) and WT rise (Goldberg et al., 2010) may enhance $N_2O$ emissions, but such effects depend on soil nitrogen status (Martikainen et al., 1993; Aerts and Ludwig, 1997).

Maljanen et al. (2003) found that WT, $CO_2$ emissions and temperature at 5 cm depth, together, explained 55% of the observed variability in $N_2O$ emissions during a two-year field study on a drained organic soil, whereas the response to N fertilisation was limited, and they suggested that N released by soil organic matter mineralisation was the main source of $N_2O$. In a study comparing GHG emissions from organic soil with different land uses in three regions of Denmark, Petersen et al. (2012) also found that site conditions such as WT, pH and precipitation contributed

significantly to explain $N_2O$ emission dynamics.

Extremely high $N_2O$ emissions corresponding to 38 and 61 kg N ha$^{-1}$ were observed from arable sites in two of the three regions studied by Petersen et al. (2012). There are several processes, which can lead to $N_2O$ formation in acid organic soil: Biotic processes include ammonia oxidation by archaea or bacteria (Herrmann et al., 2012; Herold et al., 2012; Stieglmeier et al., 2014), as well as nitrifier denitrification and heterotrophic denitrification by bacteria or fungi

(Liu et al., 2014; Maeda et al., 2015; Wrage-Mönnig et al., 2018). Abiotic $N_2O$ production can occur through chemodenitrification (Van Cleemput and Samater, 1996; Jones et al., 2015) or abiotic codenitrification (Spott et al., 2011). Characterising soil profiles with respect to potential electron donors and acceptors of putative chemical or microbial processes may inform about the potential role of individual pathways, which can then be evaluated in controlled experiments (Jørgensen et al., 2009). The sites showing extreme $N_2O$ emissions had both developed from

marine forelands and were categorised as potentially acid sulfate soil, i.e., saturated to poorly drained soil containing pyrite ($FeS_2$) that, upon oxidation, could lead to acid production in excess of the soil's neutralising capacity (Madsen and Jensen, 1988). The capillary fringe of organic soils represents an interface between saturated and unsaturated soil conditions, and it was speculated that oxidation and reduction of iron sulfides could have interacted with N transformations during periods with changing groundwater level. Drainage will promote oxidation of $FeS_2$, a process



which may be linked to microbially mediated nitrate ($NO_3^-$) reduction (Jørgensen et al., 2009; Torrento et al., 2010).
The complete reduction of $NO_3^-$ to dinitrogen ($N_2$) can proceed as follows:

$$30NO_3^- + 10FeS_2 + 20H_2O \rightarrow 15N_2 + 20SO_4^{2-} + 10Fe(OH)_3 + 10H^+ \qquad (1)$$

However, in the capillary fringe residual oxygen or, alternatively, the acidification produced by $FeS_2$ oxidation, could
favour incomplete denitrification with accumulation of the intermediate $N_2O$ (Torrento et al., 2010):

$$30NO_3^- + 8FeS_2 + 13H_2O \rightarrow 15N_2O + 16SO_4^{2-} + 8Fe(OH)_3 + 2H^+ \qquad (2)$$

Nitrate reduction *via* the reaction described in Eq. 2 could potentially have contributed to the very high $N_2O$ emissions
reported by Petersen et al. (2012) from the two arable sites, where groundwater sulfate concentrations were also
consistently high.

We studied four agricultural sites within a raised bog area with acid soil conditions, i.e., two sites with rotational
grass and two sites with a potato crop. The study covered spring and autumn periods, where high emissions of $N_2O$
were observed in previous studies (Petersen et al., 2012; Kandel et al., 2018). Dynamic variables, including $N_2O$
emissions, soil mineral N, temperature and WT level, were monitored weekly, whereas reactive iron and iron sulfides
were determined in soil profiles collected in April and/or September. We hypothesised that $FeS_2$ oxidation coupled with
$NO_3^-$ reduction was a potential driver of $N_2O$ emissions, but more so in fields with a potato crop than in undisturbed
grasslands with a continuous plant cover, where the plant N uptake is more efficient. It was further hypothesised that
seasonal variation in $N_2O$ emissions could be explained as interactions of environmental factors (soil temperature,
precipitation, WT depth) with N availability.

**2 Materials and methods**

**2.1 Study sites**

The sites selected for this study were located in Store Vildmose, which is a 5,000 ha raised bog in northern Jutland,
Denmark. The area was, until 150 years ago, the largest raised bog in Denmark, and largely unaffected by human
activity. The bog overlies a marine plain formed by the last marine transgression; the sea retreated around 8000 BC, and
peat later developed in wet parts of the landscape, attaining a maximum depth of 4.5 to 5.3 m in central parts of the bog
(Kristensen, 1945). Between 1880 and 2010, the peat has generally subsided by at least 2 m due to drainage for
agriculture or peat excavation (Regina et al., 2016), and today the peat depth is mostly 1-2 m, but in some locations
even less (Kandel et al., 2018). The peat and underlying sand is acidic and has been categorised as a potentially acid
sulfate soil (Madsen and Jensen, 1988).

Four sites were selected that were distributed along an east-west transect (Figure 1a). One arable site (*AR1*) was
cropped with second-year potato in 2015, while the adjoining site (*RG1*) had second-year rotational grass; these two
sites were also represented in a previous monitoring program (Petersen et al., 2012) as sites *N-AR* and *N-RG*,
respectively. Land use treatments (i.e., potato and rotational grass) were replicated at other sites, referred to as *AR2* and



*RG2*; site *AR2* was located 4.6 km to the west, and site *RG2* was located 1.7 km to the east of the paired *AR1-RG1* sites (Figure 1a and S1).

### 2.2 Experimental design

In January 2015, an area of 10 m × 24 m was defined at each site. Sampling positions were georeferenced using a Topcon HiPer SR geopositioning system (Livermore, CA). On 25 February 2015, each site was fenced and three 10 m × 8 m experimental blocks defined (Figure 1b). Each site was further divided along its longitudinal axis to establish two 5 m × 24 m fertilisation subplots.

For monitoring of WT depth, piezometer tubes (Rotek A/S, Sdr. Felding, Denmark) were installed to 150 cm depth at the centre of each block. On either side of the piezometers, at 2.7 m distance, collars of white PVC (base area: 55 cm × 55 cm, height: 12 cm [*RG*] or 24 cm [*AR*]) were installed to between 5 and 10 cm depth (Figure 1). The higher collars used at *AR* sites were at level with the ridges during the growth period. The collars, which were fixed to the ground by four 40 cm pegs, had a 4 cm wide flange extending outwards 2 cm from the top to support gas flux chambers. Platforms (60 cm × 100 cm) of perforated PVC were placed in front of each collar as boardwalk to prevent soil disturbance during

gas sampling. The exact headspace of each collar was determined from 16 individual measurements of distance from the upper rim; this procedure was repeated whenever collars had been removed and reinstalled in order to enable field operations.

Sets of five stainless steel diffusion probes for soil gas sampling at 5, 10, 20, 50 and 100 cm depth were installed vertically within 0.5 m of the flux measurement positions in two blocks (Block 2 and 3) at sites *AR1* and *RG1*, while at

sites *AR2* and *RG2* diffusion probes were installed in one block (Block 2). The stainless steel probes were constructed as described in detail by Petersen (2014), with a 10 cm$^3$ cell connected to the surrounding soil *via* a 3 mm diameter opening (at the sampling depth) that was covered by a silicone membrane, and connected to the soil surface via two 18G steel tubes with Luer Lock fittings (Figure S1).

A HOBO Pendant Temperature Data Logger (Onset Computer Corp., Bourne, MA) was installed at 5 cm depth in

Block 2 at each site. A mobile weather station (Kestrel 4500; Nielsen-Kellerman, Boothwyn, PA) was mounted at 170 cm height at site *RG1* for hourly recording of air temperature, barometric pressure, wind speed and direction, and relative humidity. Daily precipitation was recorded at <10 km distance from the monitoring sites at a meteorological station, from where data to fill a gap in air temperature were also obtained.

### 2.3 Management

Management within the fenced experimental sites followed the practices adopted by the respective farmers, e.g., with respect to fertiliser application, grass cuts, potato harvest and soil tillage. One exception to this was N fertilisation, which was only given to one of the two subplots in each block (Figure 1b). Fertilised subplots of the *RG1* site received 350 kg ha$^{-1}$ NS 27-4 fertiliser on 16 April (DOY106), corresponding to 94.5 kg N ha$^{-1}$. Site *RG2* was fertilised with 20-25 Mg ha$^{-1}$ acidified cattle slurry (pH 6) on 5 May (DOY125), and again on 2 July (DOY183), each time corresponding



to 90-110 kg total N ha$^{-1}$. At the second slurry application, *RG2* further received 50 kg N ha$^{-1}$ as NS 27-4 fertiliser, which was applied by mistake to both fertilisation subplots. The *AR1* site received 100 kg N ha$^{-1}$ as liquid NPS 20-3-3 fertiliser on 21 May (DOY141), while the *AR2* site received 110 kg N ha$^{-1}$ as NS 21-24 pelleted fertiliser on 30 April (DOY120). The NS fertilisers contained equal amounts of ammonium ($NH_4^+$) and nitrate ($NO_3^-$), while N in the NPS fertiliser was mainly as $NH_4^+$.

At the *RG1* site, the grass was cut in late August, while at the *RG2* site the grass was cut in late June and on 9 September (DOY252). Potato harvest at the *AR1* site took place in mid-September, with interruptions due to heavy rainfall. At the *AR2* site, the potato harvest took place on 23 September (DOY266).

**2.4 Field campaigns**

A monitoring program was conducted during spring, from 3 March (DOY63) to 16 June (DOY169), and autumn, 145 from 3 September (DOY245) to 10 November (DOY314). Weekly measurement campaigns were conducted at each of the four sites insofar as field operations permitted. During spring, there were 14, 12, 14 and 15 weekly campaigns at the *RG1*, *AR1*, *RG2*, and *AR2* sites, respectively, the differences being due to interruptions for field operations. During autumn, there were 10, 10, 7 and 10 weekly campaigns at the *RG1*, *AR1*, *RG2*, and *AR2* sites, respectively. Two sites were visited during each field trip, either *AR1 + RG1* or *AR2 + RG2*. Campaigns included registration of weather 150 conditions and WT depth, soil sampling, soil gas sampling, and N$_2$O flux measurements. With few exceptions, each campaign was initiated between 9:00 and 12:00; the order of sites visited in each trip alternated from week to week.

**2.4.1 Climatic conditions**

Air temperature, relative humidity and barometric pressure were logged at the weather station located at *RG1*. During field campaigns, the WT depth was first determined in each of the three piezometers using a Model 101 water level 155 meter (Solinst; Georgetown, Canada). At *AR1* and *AR2*, WT depth in Block 3 was further recorded at 30-minute time resolution for a period during autumn using MaT Level2000 data loggers (MadgeTech; Warner, NH, USA). Soil temperatures at 5, 10 and 30 cm depth were measured in each block using a high precision thermometer (GMH3710, Omega Newport, Deckenpfronn, Germany), and in addition continuous measurements of soil temperature at 5 cm depth were collected in block 2 at each site using HOBO Pendant Temperature Data Loggers (Onset Computer Corp., Bourne, 160 MA).

**2.4.2 Soil sampling**

During all field campaigns, soil samples were collected separately from fertilised and unfertilised subplots by random sampling of six 20 mm-diameter cores to 50 cm depth. Each core was split into 0-25 and 25-50 cm depth, and the six subsamples from each depth were pooled. The pooled samples were transported back to the laboratory in a cooling box 165 and stored at -20°C for later analysis of mineral N and gravimetric water content.



On 23 April (DOY113), and again on 2 September (DOY245), undisturbed soil cores (50 mm diameter, 30 cm segments) were collected to 1 m depth within 1 m distance from the positions of flux measurements in Block 3 of sites *RG1* and *AR1* (cf. Figure 1b). A stainless steel corer (04.15 SA/SB liner sampler, Eijkelkamp, Giesbeek, Netherlands) equipped with a transparent plastic sleeve was used. The steel corer's lower end was capped with a 4 cm long cutting

head, and hence sampling depths were 0 to 30 cm, 34 to 64 cm and 68 to 98 cm. The intact cores were capped and sealed, and transported in a cooling box to the laboratory, where they were stored at -20°C.

### 2.4.3 Soil gas sampling

Soil gas samples were collected in 6 mL pre-evacuated Exetainers (Labco Ltd, Lampeter, UK) as described by Petersen (2014) and shown in Figure S2. In brief, the diffusion probes were flushed *via* the inlet tube with 10 mL $N_2$ containing

50 $\mu$L $L^{-1}$ ethylene (AGA, Enköbing, Sweden) as a tracer. A three-way valve, mounted on the outlet tube, was fitted with a 10 mL glass syringe and an Exetainer. The displaced gas was quantitatively collected in the glass syringe from where the soil gas sample, now partly diluted by the flushing gas, was transferred to the Exetainer. After gas sampling, the probe was flushed with $2 \times 60$ mL $N_2$ to remove ethylene, and the Luer Lock fittings were capped. Samples of the $N_2$/ethylene gas mixture used for sample displacement were also transferred directly to Exetainers for gas

chromatographic analysis ($n = 3$) as reference for the calculation of dilution factors (Petersen, 2014). Sampling for soil gas was done in parallel with flux measurements, but less frequently, since equipment had to be removed during periods with field operations.

### 2.4.4 Nitrous oxide flux measurements

Gas fluxes were measured with static chambers (60 cm × 60 cm × 40 cm) constructed from 4-mm white PVC and

equipped with a rubber gasket (Emka Type 1011-34; Megatrade, Hvidovre, Denmark) as seal during chamber deployment. Chambers were further equipped with a 12V fan (RS Components, Copenhagen, Denmark) for headspace mixing that was connected to an external battery (Yuasa Battery Inc.; Laureldale, PA), as well as a vent tube with outlet near the ground to minimise effects of wind (Conen and Smith, 1998; Hutchinson and Mosier, 1981). Also, chambers were equipped with an internal temperature sensor (Conrad Electronic SE; Hirschau, Germany), and a butyl rubber

septum on top of each chamber for gas sampling. Handles attached to the top were used for straps fixing the chamber firmly against the collar. Gas samples (10 mL) were taken with a syringe and hypodermic needle immediately after chamber deployment, and then 15, 30, 45 and 60 minutes after closure. Gas samples were transferred to 6 mL Exetainer vials, leaving a 4 mL overpressure.

### 2.4.5 Soil analyses

Soil samples collected during the weekly campaigns were sieved (6 mm) and subsampled for determination of soil mineral N and gravimetric water content. Approximately 10 g fresh weight soil was mixed with 40 mL 1 M potassium chloride (KCl) and shaken for 30 min. Concentrations of $NH_4^+$ and $NO_2^- + NO_3^-$ in filtered KCl extracts were determined by autoanalyser (Model 3; Bran+Luebbe GmbH, Norderstedt, Germany) using standard colorimetric



methods (Keeney and Nelson, 1982). Gravimetric soil water content was determined after drying of soil samples at
80°C for 48 hours.

Additional soil characteristics were determined on the intact soil cores collected in April and September at *AR1* and
*RG1*. Five cm sections were subsampled from selected depths and analysed for water content, pH, electrical
conductivity (EC), total soil organic C and N, and $NO_2^-$. Soil pH and EC were measured with a Cyberscan PC300
(Eutech Instruments; Singapore) in a soil:water solution (1:2.5, w/v). Total soil organic C and total N were measured by
high temperature combustion with subsequent gas analysis using a vario MAX cube CN analyser (Elementar
Analysensysteme GmbH; Langenselbold, Germany). Soil $NO_2^-$-N concentrations were analysed in soil:water extracts
(1:5, w/v) using a modified Griess-Ilosvay method (Keeney and Nelson, 1982). Total organic C and total N were further
determined in bulk soil samples (0-25 cm and 25-50 cm depth) collected at *RG2* and *AR2* in the same weeks as
sampling of intact cores took place at *AR1* and *RG1*.

The intact soil cores from the September sampling were further analysed for acid volatile sulfides (AVS) and
chromium reducible sulfur (CRS) as indices of FeS and $FeS_2$, respectively. Quantification of AVS and CRS was based
on passive distillation adapted from Ulrich et al. (1997) and Burton et al. (2008). Briefly, 0.5 g soil and a trap with 4
mL alkaline Zn-acetate solution (5%) was placed in 120 mL butyl-stoppered (and crimp-sealed) serum bottles, which
were evacuated (1 kPa) and pressurised with $N_2$ (150 kPa) three times to remove $O_2$, eventually leaving the headspace
with $N_2$ at atmospheric pressure. Acid volatile sulfide (primarily FeS) was liberated and trapped as ZnS after injection
of 12 mL anoxic 2 M HCl followed by sonication (0.5 h) and incubation (24 h) on a rotary shaker (20ºC). Using the
same approach with replicate soil samples, combined AVS and CRS (primarily elemental S and $FeS_2$) was trapped after
injection of 12 mL 1 M $Cr^{2+}$ in 2 M HCl, prepared by reduction of $CrCl_3$ (Røy et al., 2014). Trapped sulfide (ZnS) in
the two traps was measured colorimetrically using diamine reagent (Cline, 1969), and CRS was then calculated by
difference.

The concentration of total reactive Fe (TRFe) at selected depth intervals was determined in the samples from both
April and September samplings of intact soil cores. The analysis of TRFe was done using a dithionite-citrate extraction
(Carter and Gregorich, 2007; Thamdrup et al., 1994) followed by $Fe^{2+}$ analysis with the colorimetric ferrozine method,
which included hydroxylamine as reducing agent (Viollier et al., 2000). The extraction dissolves free (ferric) Fe oxides
(except magnetite, $Fe_3O_4$), as well as (ferrous) Fe in FeS, but not $FeS_2$.

Finally, the total reduction capacity of the peat at depths of 27-30 cm, 61-64 cm and 95-98 cm was determined. In
brief, a suspension (soil:solution, 1:25; w/v) of oven dried (105°C) sieved soil (<2 mm) and 25 mM cerium(IV) sulfate,
$Ce(SO_4)_2$ in 5% sulfuric acid ($H_2SO_4$) was shaken horizontally for 24 h at 275 rounds per minute (rpm). After
centrifugation at 2,000 rpm, residual Ce(IV) was measured by end-point titration using a solution of 5 mM $FeSO_4$ in 5%
$H_2SO_4$. The amount of reduced compounds was calculated and expressed as meq $kg^{-1}$.

### 2.4.6 Gas analyses



Nitrous oxide concentrations were analysed on an Agilent 7890 gas chromatograph (GC) with a CTC CombiPal auto-sampler (Agilent, Nærum, Denmark). The instrument had a 2 m back-flushed pre-column with Hayesep P connected to a 2 m main column with Poropak Q. From the main column, gas entered an electron capture detector (ECD). The carrier

was $N_2$ at a flow rate of 45 mL min$^{-1}$, and Ar-CH$_4$ (95%/5%) at 40 mL min$^{-1}$ was used as make-up gas. Temperatures of the injection port, columns and ECD were 80, 80 and 325°C, respectively. Concentrations were quantified with reference to synthetic air and a calibration mixture containing 2013 nL L$^{-1}$ $N_2O$. Soil profile $N_2O$ concentrations were frequently at several hundred µL L$^{-1}$; linearity of the EC detector response was ascertained up to 1600 µL L$^{-1}$, but the entire range was not included in analytical runs as a standard practice, and therefore the higher equivalent gas phase

concentrations are relatively uncertain.

Ethylene concentrations in soil gas samples and flushing gas were analysed following a separate injection with an extended run time. All GC settings were as described earlier, except that gas from the main column was directed to a flame ionisation detector supplied with 45 mL min$^{-1}$ H$_2$, 450 mL min$^{-1}$ air, and 20 mL min$^{-1}$ N$_2$; the detector temperature was 200°C.

**2.5 Data processing and statistical analyses**

Individual $N_2O$ fluxes were calculated in R (version 3.2.5, R Core Team, 2016) using the package HMR (Pedersen et al., 2010). This program analyses non-linear concentration-time series with a regression-based extension of the model of Hutchinson and Mosier (1981), and linear concentration-time series by linear regression (Pedersen et al., 2010). Statistical data ($p$ value, 95% confidence limits) are provided by HMR for both categories of fluxes. The choice to use a

linear or non-linear flux model was made based on scatter plots and the statistical output.

The temporal dynamics of $N_2O$ fluxes were analysed using a generalised linear mixed model defined with the identity link function, the gamma distribution (see Jørgensen and Labouriau, 2012; McCullagh and Nelder, 1989), and Gaussian random components. The model contained a fixed effect representing the interaction between crop, fertilisation and sampling day, and random effects representing site and sampling position. Standard Gaussian linear

mixed models could not be applied, since these models failed to pass standard model control checks (e.g., the Shapiro-Wilk test of normality applied to the residuals and the Bartlett test). Moreover, standard transformations such as logarithm, square-root, inverse, and the entire family of Box-Cox transformations, also failed to pass the basic model control techniques referred to above. Cumulative $N_2O$ emissions were therefore estimated and treatment effects analysed by specially designed linear contrasts as described in detail by Duan et al. (2017), who showed that models

with untransformed responses (when using adequate distributions) allow simple statistical inference of the time-integrated $N_2O$ emissions.

The dependence structure of variables that were potential drivers of $N_2O$ fluxes were studied using the class of multivariate models called "graphical models" (Whittaker, 1990, see also Labouriau et al., 2008a,b; and Lamandé et al., 2011 for applications in soil science). These models represent the dependence of variables using an undirected graph

(not to be confounded with the word "graph" used to refer to a plot), i.e., the mathematical structure composed of vertices, represented by points, and edges connecting pairs of vertices, represented by lines connecting points,



according to the convention explained below. In graphical models, the variables of interest are the vertices of the graph (represented as labelled points). Here the variables used were: soil temperature at 5 cm depth (Temp5); soil temperature at 30 cm depth (Temp30); $NH_4^+$ and $NO_3^-$ concentrations in the top soil (AmmoniumT and NitrateT); $N_2O$

concentration of the soil gas diffusion probe closest to, but above the WT, i.e., in the capillary fringe ($N_2$OWT); and finally, the $N_2O$ flux ($N_2$O-flux). The dependence structure of these variables was characterised by the conditional covariances between each pair of variables given the other variables. Those conditional covariances were simultaneously estimated using the available data according to a statistical model. The graph representation of the model is constructed by connecting the pairs of vertices (i.e., pairs of variables) by an edge when the conditional

correlation of the two corresponding variables, given all the other variables, is different from zero. It is possible to show that two variables directly connected in the graph carry information on each other that is not already contained in the other variables (see Whittaker, 1990, Jørgensen and Labouriau, 2012). Moreover, the absence of an edge connecting two vertices indicates that (even a possible) association between the two corresponding variables can be entirely explained by the other variables. According to the general theory of graphical models, if two groups of variables, say A

and B, are separated in the graph by a third group of variables, say C (i.e., every path connecting an element of A with an element of B necessarily contains an element of C), then A and B are conditionally uncorrelated given C (see Lauritzen, 1999). This property, called the separation principle, will be used below to draw non-trivial conclusions on the interrelationship between $N_2$O-flux related variables. The graphical models were inferred by finding the model that minimised the BIC (Bayesian information criterion, i.e., a penalised version of the likelihood function) as implemented

in the R package gRapHD (Abreu et al., 2010). This inference procedure yields an optimal representation of the data in the sense that the probability of correct specification of the model, when using this penalization, tends to one as the number of observations increases (see Haughton, 1988). The confidence intervals for the conditional correlations were obtained by a non-parametric bootstrap procedure (Davidson and Hinkley, 1997) with 10,000 bootstrap samples. Separate analyses were conducted for each combination of season and crop, since different dependency patterns appear

in those groups.

## 3 Results

### 3.1 Climatic conditions

During the spring monitoring period, the daily mean air temperature varied between 1 and 15°C, with an increasing

trend over the period, and total rainfall was 220 mm. During the autumn monitoring period, the daily mean air temperature declined from 15 to 5°C, and total rainfall was 148 mm; the most intense daily rain events during spring and autumn were 16.9 and 33.2 mm, respectively. For 2015 as a whole, the annual mean air temperature in the area was 8.7°C, and annual precipitation was 920 mm.

Soil temperature at 5 cm depth showed a clear diurnal pattern (Figure S3), but at all four sites the temperature at the

time of chamber deployment was close to the daily mean temperature at this depth. Thus, across the four sites the



average deviation ranged from 0.2 to 0.9°C, and the largest deviations on a single day were -2.0 and 2.1°C, respectively.

### 3.2 Soil characteristics

Soil characteristics were determined by analyses of intact cores collected in late April 2015 (Table 1). At all sites the

soil was acidic, with pH ranging from 4.7 to 5.4. At the paired sites *AR1* and *RG1*, a weak decline in pH was indicated at 40-50 cm depth. Electrical conductivity at *AR1* and *RG1* sites ranged from 0.15 to 0.91 mS cm$^{-1}$, with no obvious trends in the data; the highest value (0.91 mS cm$^{-1}$) occurred at site *AR1* at 93-98 cm in a layer dominated by sand underlying the peat. Total organic C concentrations at sites *AR1* and *RG1* were 34-43% in the upper 0-40 cm, but then dropped to only 0.3-0.6% at *c.* 1 m depth in the sand. The peat was amorphous and well-decomposed at 0-20 cm depth,

while the underlying peat was dominated by intact plant debris. At site *RG2*, the process of peat degradation was evident even at 0-50 cm depth, where TOC concentrations only just met the requirements for being defined as an organic soil. At *RG2*, the organic C content was below 20 and 10% at 0-25 and 25-50 cm depth, respectively. Site *AR2* was characterised by a uniform peat layer (33-38% organic C) at 0-50 cm depth. The C:N ratios ranged between 14 and 26 in the organic soil layers.

Acid volatile sulfide ranged from 1.7 to 4.9 $\mu$g S g$^{-1}$ soil across the four sites and showed no clear relationship with soil depth. This was also the case for CRS, which ranged from 24 to 155 $\mu$g S g$^{-1}$ dry weight soil. Total reactive Fe (TRFe) concentrations in soil profiles from sites *AR1* and *RG1* ranged from 1.19 to 4.99 mg g$^{-1}$ dry weight soil at 0-50 cm depth, and hence concentrations of reactive Fe were up to 2500 times higher than concentrations of Fe in AVS (assuming this was FeS), and 25-90 times higher than Fe in CRS results (assuming this was FeS$_2$). At sites *AR1* and

*RG1*, TRFe declined below 20 cm depth and was close to zero in the sand below the peat layer (Table 1).

The highest concentrations of TRFe at sites *RG1* (Figure 2b) and *AR1* (Figure 2d) occurred at 20 cm depth on 23 April. At site *AR1*, a sink for TRFe at 40-60 cm depth was indicated. There were only minor differences in the distribution of TRFe between seasons. There was a strong correlation between TRFe and TOC across all sites ($r = 0.88$, $n = 16$).

At both *AR1* and *RG1*, the total reduction capacity at 30 cm depth was outside the range of the analytical method, but >11,500 meq kg$^{-1}$. The reduction capacity dropped to around 1000 meq kg$^{-1}$ at 60 to 65 cm depth with the declining organic matter content, and 50 to 100 meq kg$^{-1}$ in the sandy layer at 100 cm depth.

### 3.3 Soil mineral N dynamics

Soil concentrations of NH$_4^+$ and NO$_3^-$ at 0-25 and 25-50 cm depth were determined in connection with field campaigns (Tables S1-S4). At *AR* sites, there was an accumulation of mineral N at both depth intervals during May (Table S2, S4), although concentrations at *AR1* were much greater than at *AR2*, and at site *AR2* only NO$_3^-$ accumulated; high



concentrations in the fertilised subplot in May was due an external input of fertiliser N. Fertilisation increased $NH_4^+$-N and $NO_3^-$-N concentrations to generally 100-200 µg g$^{-1}$ dry weight soil at all sites except *RG2* (Table S3), where

acidified cattle slurry was surface applied. It is not clear if the slurry infiltrated to >50 cm, or if plant uptake was very effective. The residence time for mineral N in the soil solution was generally longer at *AR* compared to *RG* sites. Accumulation of $NO_3^-$ in the weeks after fertilisation was observed at all sites, and also evidence for some transport to 25-50 cm depth.

Nitrite-N concentrations were determined in undisturbed soil collected from *RG1* and *AR1* on 23 April and 2

September 2015. In April, the concentration of $NO_2^-$-N at both sites was highest (*c.* 10 µg g$^{-1}$ dry weight soil) around 40 cm depth and declined towards the surface and deeper layers (Figure 2a,c). A decline in $NO_2^-$-N concentration was indicated at 50 cm depth at site *AR1* relative to site *RG1*, and also a depletion of TRFe was indicated. However, there was also a lower concentration of peat (cf. TOC in Table 1), which may account for this difference. In September, $NO_2^-$-N concentrations were <1 µg g$^{-1}$ dry weight soil at both sites, while the much higher concentrations of TRFe were

comparable to those in April.

### 3.4 Groundwater table dynamics

During spring, WT depth at sites *RG1* and *AR1* ranged from 17 to 81 cm, with a steady decline until the end of April (DOY120) that was followed by a period with fluctuations around 60-80 cm depth due to frequent rainfall (Figures 3 and 4). During the first half of September (DOY246 to 259), rainfall caused the WT to rise from 80 to 40 cm depth

(Figures 5 and 6). The continuous measurements of WT depth (data not shown) revealed, however, that on two occasions (DOY248 and 260) the WT depth rose to 20 cm depth and only gradually declined during the following days. From mid-September there followed a period with a gradual WT decline until early November, where upon the WT rose from 90 to 45 cm depth during a week with intense rainfall.

At site *RG2*, the WT was mostly at 50-60 cm depth during spring, with a temporarily rise to 30 cm depth by 3 June

(DOY139; see Figure 3). In the autumn, sampling campaigns could not be initiated until DOY260 due to harvest. By this time the WT was close to the surface following intense rainfall, but then declined to 80-100 cm in the sandy subsoil (Figure 5). The WT at site *AR2* was consistently between 45 and 60 cm depth during spring except for a transient increase to 35 cm depth in early June (Figure 4). During autumn, the WT rose to the soil surface in September (DOY260), and then gradually withdrew until early November when rainfall caused a *c.* 40 cm increase (Figure 6), as

also observed at sites *RG1* and *AR1*.

### 3.5 Soil N$_2$O concentration profiles

Equivalent gas phase concentrations of N$_2$O, as determined by passive diffusion samplers, are presented as contour plots (Figures 3-6) based on data compiled in Table S5. A logarithmic grey scale had to be used in order to show trends within both *RG* and *AR* treatments, as concentrations sometimes differed by orders of magnitude, this was also true

between depths within individual profiles in many cases. Some gaps occur where diffusion probes could not be installed or were temporarily removed due to field operations.





Under the rotational grass at site *RG1*, soil $N_2O$ concentrations during spring were mostly between 0.1 and 3 $\mu L\ L^{-1}$ (Figure 3). A higher concentration (15 $\mu L\ L^{-1}$) was observed at 40-80 cm depth in the fertilised subplot around DOY139, but only in Block 3 of the field plot. At site *RG2,* the concentrations of $N_2O$ in the soil during spring were

generally similar to those of *RG1*, although there were more values in the 1-10 $\mu L\ L^{-1}$ concentration range (Table S5). However, on 3 June (DOY154) a significant increase in $N_2O$ concentration occurred in the fertilised part of the plot with a maximum of 560 $\mu L\ L^{-1}$ at 100 cm depth (i.e., well below the WT). This occurred during a period with frequent rainfall and could have been caused by $NO_3^-$ leaching from the top soil. Soil $N_2O$ concentrations in the unfertilised plot also increased around this time, but only to c. 15 $\mu L\ L^{-1}$ and mainly near the soil surface.

During autumn, $N_2O$ concentrations in the soil profile at the *RG1* and *RG2* sites varied between 0 and 12 $\mu L\ L^{-1}$ independent of fertilisation and with a tendency for highest concentrations at 10-20 cm depth (Figure 5).

The arable site *AR1*, with sampling positions located 10-20 m from those of site *RG1*, showed very different soil $N_2O$ concentration dynamics during spring (Figure 4). There was a consistent accumulation of $N_2O$ at 50 and 100 cm depth where seasonal concentrations averaged 340 and 424 $\mu L\ L^{-1}$, respectively. In contrast, at 5, 10 and 20 cm depth

the average $N_2O$ concentrations were 10-30 $\mu L\ L^{-1}$, and there was no clear response to fertilisation on DOY141 in terms of soil $N_2O$ accumulation. There was significant within-site heterogeneity in soil conditions, and the highest concentrations were observed in the unfertilised subplot. Between DOY75 and DOY100, the concentrations of $N_2O$ peaked at nearly 1500 $\mu L\ L^{-1}$ at 50 cm depth and were 2-3 fold higher than at 100 cm depth, indicating that $N_2O$ was produced in the capillary fringe as WT in this period was around 60 cm depth. At site *AR2,* the highest soil $N_2O$

concentrations during early spring were consistently observed at 20 cm depth, but then gradually declining to reach the background level of 0.3 $\mu L\ L^{-1}$ in mid-May (around DOY130). In the unfertilised field plot, the $N_2O$ concentration then increased again at 20 cm depth to reach 272 $\mu L\ L^{-1}$ following rainfall, and a WT rise to 35 cm depth. With fertilisation, soil $N_2O$ concentrations were even higher at 10 cm depth and reached nearly 400 $\mu L\ L^{-1}$ in mid-June.

September was characterised by heavy rainfall (114 mm in total), and at site *AR1* a substantial rise in the WT from

80 to 40 cm depth was observed (Figure 6). Soil $N_2O$ concentrations showed a dual pattern, with maxima at 10 and 100 cm depth through to DOY266 (end of September), and after this time soil $N_2O$ rapidly declined as the WT withdrew. Nitrous oxide concentrations equivalent to several hundred $\mu L\ L^{-1}$ were measured even at 5 cm depth during this period. During late autumn, the $N_2O$ concentration at 0-50 cm depth varied between 0 and 20 $\mu L\ L^{-1}$, whereas at 100 cm depth it remained high at 100-850 $\mu L\ L^{-1}$. At site *AR2,* the groundwater level was higher than at *AR1* and reached the soil

surface by mid-September. Soil $N_2O$ accumulated in both fertilised and unfertilised subplots following saturation of the soil, again with the highest concentrations at 20 cm depth. A secondary increase was observed near the soil surface at the last sampling on DOY314 in November, in response to a period with rainfall and a rapid WT rise.

### 3.6 Nitrous oxide emissions

At site *RG1*, $N_2O$ emissions during spring ranged from 0 to 550 $\mu g\ N_2O\ m^{-2}\ h^{-1}$, with no effect of fertiliser amendment

(Figure 3). Growth of the grass sward showed a strong response to fertilisation (not shown), and presumably there was a rapid uptake of the N added. At site *RG2*, however, a peak in $N_2O$ emissions occurred on DOY154, and the flux was





still elevated at the next two samplings. This high flux coincided with the elevated soil profile $N_2O$ concentrations described above. At site *AR1*, the $N_2O$ fluxes were generally much higher than at the *RG1* site (Figure 4). Fluxes during early spring reached 2000-6000 µg $N_2O$ $m^{-2}$ $h^{-1}$ and were higher than in late spring where, as for site *RG1*, no effect of

N fertilisation was observed. Hence, the higher emissions were associated with soil conditions and not fertilisation. The potato field at site *AR2* showed a different pattern, with $N_2O$ fluxes remaining low during early spring, and for several weeks after fertilisation. The highest emissions occurred, independent of fertilisation, in June when a WT rise to 35 cm depth was observed.

In the autumn, $N_2O$ fluxes from site *RG1* were consistently low (Figure 5). The first sampling at site *RG2* was on

DOY260 in mid-September, where a high flux of 3000 µg $N_2O$ $m^{-2}$ $h^{-1}$ was seen, which dropped to near zero within 1-2 weeks. Nitrous oxide emissions at site *AR1* were high during September at 4000-10,000 µg $N_2O$ $m^{-2}$ $h^{-1}$ independent of N fertilisation, and subsequently declined to near zero (Figure 6). The high fluxes coincided with a rise in the WT from 80 to 40 cm depth, and the decline in fluxes coincided with WT withdrawal. At site *AR2* the pattern in $N_2O$ emissions was similar, and again the dynamics of $N_2O$ fluxes aligned with WT dynamics.

Cumulative $N_2O$ emissions were calculated for the 99-105 days of monitoring in spring, and for the 47-69 d period in autumn (Table 2). At *RG* sites, the average $N_2O$ flux from fertilised grassland was significantly higher than from unfertilised grass (7.3 *vs.* 2.0 kg $N_2O$ $ha^{-1}$) during spring. At *AR* sites with potato, there was no significant effect of N fertilisation, but the cumulative $N_2O$ emissions of 15-17 kg $N_2O$ $ha^{-1}$ were much higher than from *RG* sites. In the autumn there were no residual effects of N fertiliser application in spring, and average cumulative emissions at the *RG*

and *AR* sites were 2 and 15 kg $N_2O$ $ha^{-1}$, respectively.

**3.7 Interrelationships between driving variables of $N_2O$ production**

Graphical models were used to study the dependence structure among selected soil variables and $N_2O$ fluxes. Interestingly, at *RG* sites in both spring (Figure 7a) and autumn (Figure 7b), and at *AR* sites in spring (Figure 7c), the only variable with a direct link to $N_2O$ flux was soil $N_2O$ concentration in the capillary fringe ($N_2O_{WT}$), indicating that

$N_2O_{WT}$ carried information on the $N_2O$ flux that could not be explained by indirect correlations between the other variables. Moreover, the variable $N_2O_{WT}$ separated $N_2O$ flux from the other variables in the graph which, according to the separation principle (an instance of the general theory of graphical models), indicates that information about this variable rendered all the other variables uninformative with respect to $N_2O$ flux. For example, in the analysis of *AR* sites in spring (Figure 7c), the variables $N_2O$ flux and Temp5 were not directly connected, and therefore any correlation

between Temp5 and $N_2O$ flux could be completely explained by other variables. The only exception to this pattern was *AR* sites in the autumn (Figure 7d), where instead two other variables showed a significant relationship with $N_2O$ flux; one variable was NitrateT, i.e., $NO_3^-$-N concentration in the top soil, and the other variable was soil temperature at 30 cm depth. All other relationships were unrelated to $N_2O$ flux, or could be accounted for by other variables.

**4 Discussion**



We investigated seasonal dynamics of $N_2O$ emissions and soil conditions in a region in Northern Denmark that was designated as a hotspot for $N_2O$ emissions in a meta-analysis of organic soils across Europe (Leppelt et al., 2014). Spring and autumn monitoring periods together covered 152-174 d, and cumulative $N_2O$ emissions during these periods were in total 3-6 kg $N_2O$-N ha$^{-1}$ for rotational grass, and 19-21 kg $N_2O$-N ha$^{-1}$ for arable sites with a potato crop. These numbers indicate that annual emissions were comparable to (*RG*), or clearly above (*AR*), the IPCC emission factors for drained organic soil of 8 and 13 kg $N_2O$-N ha$^{-1}$ yr$^{-1}$ for nutrient rich grassland and cropland, respectively (IPCC, 2014). Hence, the observations confirmed that organic soil drained for agriculture in this region constitutes a high risk for $N_2O$ emissions, but also showed that this risk depends on land use.

Leppelt et al. (2014) concluded that high $N_2O$ emissions are associated with cropped land having a pH below 4.7, C:N ratios below 30-35, and WT depths of 0.2-0.9 m, and they found a significant positive relationship with annual precipitation. This sites investigated here largely fit this description, but the specific mechanisms behind high $N_2O$ emissions are not easily derived from average annual conditions. The present study was therefore planned to examine high-emission periods at higher spatial and temporal resolution to elucidate environmental controls and possible pathways, such as $FeS_2$ oxidation being a driver of $N_2O$ emissions.

### 4.1 Nitrous oxide emissions and water table dynamics

It is well established that $N_2O$ emissions from organic soil may be enhanced by drainage (Martikainen et al., 1993; Taft et al., 2017). The response will appear within days, as shown by Aerts and Ludwig (1997) in an incubation study with an oscillating WT. A stimulation of $N_2O$ emissions was also observed by Goldberg et al. (2010) when simulating drought under field conditions, but in addition a pulse of $N_2O$ occurred after rewetting. In the present study, the response to WT drawdown was complex, i.e., at sites *RG1* and *AR1* there was a stimulation of $N_2O$ emissions as WT declined in early spring, while this was not evident at sites *RG2* and *AR2*. During autumn there was generally no effect of WT drawdown on $N_2O$ emissions. In contrast, rising WT and/or increasing soil wetness in late spring and in the autumn resulted in a consistent increase in $N_2O$ emissions at all sites. Hence, the relationship between WT depth and $N_2O$ emission showed seasonal patterns and site-specific effects, which indicated that other soil properties modified the effect of WT on $N_2O$ emissions.

### 4.2 Nutrient status and land use

The repeated increase in $N_2O$ emissions after WT drawdown reported by Aerts and Ludwig (1997) was observed only with eutrophic peat, whereas a mesotrophic peat showed no effect of WT treatment on $N_2O$ emissions, which were consistently low. A similar interaction between nutrient status and WT depth was observed in field studies comparing $N_2O$ emissions from minerotrophic and ombrotrophic boreal peatlands (Martikainen et al., 1993; Regina et al., 1996). Thus, nutrient status, and N availability in particular, was probably a driver for the higher $N_2O$ emissions at *AR* sites used for potatoes. The *RG* sites with rotational grass, in contrast, showed much lower $N_2O$ emissions despite similar soil conditions and N fertiliser input. Grasslands on organic soil generally show lower emissions of $N_2O$ compared to arable organic soil (Eickenscheidt et al., 2015; Petersen et al., 2012), presumably because plants compete successfully with microorganisms for available N. Schothorst (1977) estimated peat decomposition indirectly from the N-content in herbage yield of grassland and concluded that the soil supplied 96 kg N ha$^{-1}$ when the drainage depth was 25 cm, but



and 224 kg N ha$^{-1}$ with the WT in drainage ditches at 70 and 80 cm depth, respectively. In the present study, soil NH$_4^+$-N and NO$_3^-$-N concentrations at *RG1* remained mostly below 5 µg g$^{-1}$ dry weight soil except for a short period after fertilisation (Table S1). In contrast, at site *AR1* the NO$_3^-$-N concentrations were mostly at 15-25 µg g$^{-1}$ dry weight soil during spring, and it declined more slowly after fertilisation, where soil mineral N peaked at *c.* 500 and 200 µg g$^{-1}$ dry weight at 0-25 and 25-50 cm depth, respectively (Table S2). This does indicate that the grass sward effectively took up N mineralised from soil organic matter above the WT.

Independent of land use there was no immediate effect of N fertiliser application on emissions of N$_2$O. Other studies also found a limited response to fertilisation (Maljanen et al., 2003; Regina et al., 2004), although Regina et al. (2004) observed a peak in N$_2$O emissions in late spring after rainfall.

### 4.3 Nitrogen dynamics and N$_2$O concentration in soil profiles

Only pooled soil samples from 0-25 and 25-50 cm depth were available for characterisation of soil mineral N dynamics (Table S1-S4), but additional information can be derived from soil N$_2$O concentration profiles (Goldberg et al., 2008). The soil gas diffusion probes used in this study were installed vertically and thus did not disturb soil stratification prior to monitoring. At *RG* sites, soil N$_2$O concentrations were generally low and did not provide clear evidence for N transformations. In contrast, at *AR* sites there was during spring an accumulation of N$_2$O in the soil; the highest concentrations at *AR1* occurred at 50 to 100 cm depth, while at site *AR2* the highest concentrations were at 20 cm depth, in accordance with the higher groundwater table. This suggests that peat decomposition was a significant source of mineral N, and that biotic or abiotic processes led to extensive N$_2$O accumulation. At site *RG2,* accumulation of N$_2$O at 1 m depth in late May suggested that mineral N from the acidified cattle slurry had leached from the top soil (Figure 3). Soil N$_2$O concentration profiles thus indicated that emissions of N$_2$O at *AR* sites before fertilisation were due to an interaction between peat decomposition and declining WT. In the period following fertilisation, accumulation of N$_2$O in the soil profile was mostly associated with precipitation and rising WT.

Precipitation was high during September 2015, and the rapid rise in WT toward the soil surface resulted in accumulation of N$_2$O in the top soil at all sites. However, N$_2$O concentrations peaked at around 10 µL L$^{-1}$ at *RG* sites, as opposed to several hundred µL L$^{-1}$ at *AR* sites. Soil NO$_3^-$-N concentrations at 0-50 cm depth in early September (DOY145) were 5 µg g$^{-1}$ dry weight soil at site *RG1* (Table S1), but 40-150 µg g$^{-1}$ dry weight soil at site *AR1* (Table S2), which could have supported denitrification activity. It is not clear if the source of NO$_3^-$ was decomposing potato crop residues or accelerated peat decomposition following harvest, or both.

### 4.4 Environmental controls

The previous sections have indicated that effects of land use and climate on N$_2$O emissions (Leppelt et al., 2014; Mu et al., 2014) are modified by soil conditions that may vary across the year. We investigated possible drivers of N$_2$O emissions using a statistical method represented by graphical models, which identified N$_2$O concentration in the capillary fringe as the strongest predictor of N$_2$O emissions from both grassland and arable soil in spring, and from grassland soil in the autumn. The implication is that N transformations at depth in the soil, and not in the top soil (despite fertilisation in some treatments) were the main source of N$_2$O escaping to the atmosphere in these cases. This is



in accordance with Goldberg et al. (2010), who found that $N_2O$ emissions from a minerotrophic fen were produced at 30-50 cm depth. Peat decomposing in the capillary fringe during WT drawdown could thus have been a driver of $N_2O$ production, and indeed the highest concentrations of $N_2O$ were mostly observed at 20 or 50 cm depth (Table S5). On

the other hand, at site *AR1* the soil $N_2O$ concentration was high even at 100 cm depth, indicating that $N_2O$ was also produced in the saturated zone.

At the arable sites, the regulation of $N_2O$ emissions in the autumn was different from that in spring, since the graphical model identified $NO_3^-$ in the top soil, and soil temperature at 30 cm depth, as predictors of $N_2O$ emissions (Figure 7), although it should be noted that accumulation of $NO_3^-$ was much greater at site *AR1* compared to *AR2*.

Rainfall most likely triggered denitrification at the *AR* sites, by rapidly increasing soil water-filled pore space, impeding oxygen supply (Barton et al., 2008). This interpretation is supported by $N_2O$ concentrations increasing dramatically around the WT depth (Figure 6). In an annual study, conducted in other parts of the Store Vildmose bog, Kandel et al. (2018) also measured high peak emissions of $N_2O$ from a potato cropping system, i.e., around 2000 µg $N_2O$ $m^{-2}$ $h^{-1}$ in October 2014 and 6000 µg $N_2O$ $m^{-2}$ $h^{-1}$ in June 2015, which coincided with $NO_3^-$ accumulation.

**4.5 Possible pathways of $N_2O$ formation**

We hypothesised that $NO_3^-$ reduction coupled with $FeS_2$ oxidation could be a pathway of $N_2O$ formation in this acid organic soil, and $FeS_2$, measured as CRS, was quantified at selected depths (Table 1). Assuming a bulk density of peat in this area of 0.15 g $cm^{-3}$ (Schäfer et al., 2012), the amount of CRS at 0-50 cm depth at site *AR1* would correspond to around 180 mmol $FeS_2$ $m^{-2}$, whereas the $N_2O$ emission observed during spring and autumn monitoring periods together

constituted 145 mmol N $m^{-2}$. It is thus possible that the process described by Eq. 2 contributed to $N_2O$ emissions, though probably not during spring, where $N_2O$ emissions were unrelated to soil $NO_3^-$ dynamics. The total reduction capacity of the peat was much higher than that represented by $FeS_2$, i.e., >11,500 meq $kg^{-1}$ at 27-30 cm depth. Also, the concentration of total reactive Fe was 25-90 times higher than that of CRS. Together this indicates that reducing agents other than $FeS_2$ were more important. Subsequent incubation experiments with addition of $FeS_2$ together with different

electron acceptors also suggest that $FeS_2$ oxidation is not a driver of $N_2O$ emissions in this peat soil (manuscript in preparation), and hence alternative pathways should be considered.

Bacterial nitrification, denitrification, and nitrifier-denitrification are all potential pathways of $N_2O$ formation (Braker and Conrad, 2011), and the significant relationship with $NO_3^-$ at *AR* sites in the autumn (Figure 7) suggested that denitrification was the main source in this period. In early spring, however, the emissions were more strongly

related to $N_2O$ accumulating near the WT depth. Ammonia oxidising bacteria (AOB) are scarce in acid peat despite the presence of nitrite oxidising bacteria (NOB) (Regina et al., 1996), and some studies indicate that ammonia oxidising archaea (AOA) predominate in both abundance and activity (Herrmann et al., 2012; Stopnišek et al., 2010). Stieglmeier et al. (2014) isolated an AOA from soil that emitted $N_2O$ at a rate corresponding to 0.09% of the $NO_2^-$ produced independent of $O_2$ availability, but it is not known if this organism is present in acid organic soil. Stopnišek et al. (2010)

found that AOA activity was not stimulated by an external source of $NH_4^+$ and concluded that the activity was associated with N released from decomposing soil organic matter. Thus, in early spring the anaerobic conditions of





saturated peat may have been a limiting factor for N mineralisation and in turn ammonia oxidation, a constraint which could have been alleviated as the WT declined and oxygen entered deeper soil layers.

Ammonia oxidation may drive $N_2O$ emissions indirectly *via* production of $NO_2^-$ or $NO_3^-$. Nitrite accumulated at 20-50 cm depth in late April at both *RG1* and *AR1* (Figure 2). This was consistent with peat decomposition in connection with WT drawdown, but also suggests an imbalance between ammonia oxidation and nitrite oxidation activity. Estop-Aragonés et al. (2012) found that oxic-anoxic interfaces in peat soil were located above the WT depth, and hence the capillary fringe in this study may have been partly anoxic. Oxygen affinity differs between nitrifiers, with AOA>AOB>NOB (Yin et al., 2018), and thus oxygen limitation could have caused the accumulation of $NO_2^-$. In acid

soil, this would result in product inhibition by $HNO_2$, if there were no mechanism to remove $NO_2^-$; this would be especially true for *AR* sites, where mineral N accumulation was three to four times higher compared to *RG* sites (Tables S3-S6). Nitrifier-denitrification is one mechanism by which ammonia oxidisers can avoid $HNO_2$ accumulation, and this process leads to $N_2O$ formation (Braker and Conrad, 2011). Another potential sink for $NO_2^-$ is chemodenitrification, an abiotic reaction in which $NO_2^-$ reacts with $Fe^{2+}$ to produce $N_2O$ (Jones et al., 2015):

$$4Fe^{2+} + 2NO_2^- + 5H_2O \rightarrow 4FeOOH + N_2O + 6H^+ \qquad (3)$$

where $Fe(OH)_3$ is shown as anhydrous FeOOH. Some depletion of TRFe was indicated at 50 cm depth at site *AR1*, which coincided with a similar depletion in $NO_2^-$ (Figure 2). Nitrifier-denitrification and chemodenitrification are both sinks for $NO_2^-$, and therefore both pathways were potential sources of $N_2O$ emissions in spring.

The observation that TRFe concentrations were much higher than those of AVS or CRS (Table 1) was unexpected,
but makes it relevant to consider alternative reactions with a potential to produce $N_2O$ that involve iron oxides/hydroxides rather than $FeS_2$. One such pathway is Feammox, a process whereby ammonia oxidation coupled with ferric iron reduction can produce $NO_2^-$ below pH 6.5 (Yang et al., 2012):

$$6Fe(OH)_3 + 10H^+ + NH_4^+ \rightarrow 6Fe^{2+} + 16H_2O + NO_2^- \qquad (4)$$

Nitrate can also be produced under these conditions (Yang et al., 2012; Guan et al., 2018):

$$8Fe(OH)_3 + 14H^+ + NH_4^+ \rightarrow 8Fe^{2+} + 21H_2O + NO_3^- \qquad (5)$$

A shuttle of $Fe^{2+}$ between Feammox and chemodenitrification could have caused the accumulation of $N_2O$ under anoxic conditions in the saturated zone, where presumably the availability of $NH_4^+$ from peat mineralisation would be a limiting factor. The confirmation of pathways will require more detailed investigations that should also involve molecular analyses targeting microbial communities in the soil profile.


## 5 Conclusion

As hypothesised, there was an effect of land use on $N_2O$ emissions, which were clearly higher from arable sites with a potato crop compared to rotational grassland independent of fertilisation. There were strong seasonal dynamics in $N_2O$ emissions that were associated with WT dynamics. In spring there was no direct response to the input of fertiliser N,



and instead $N_2O$ emissions mainly reflected the accumulation of $N_2O$ near the WT. At sites used for a potato crop, $NO_3^-$ accumulated after harvest and was significantly related to $N_2O$ emissions. Pyrite was present at low concentrations, and hence some $N_2O$ emission from $NO_3^-$ reduction coupled with $FeS_2$ oxidation could not be dismissed, at least in the autumn. However, the total reduction capacity of peat was much higher than that represented by $FeS_2$, and reactive Fe was predominantly in forms other than pyrite, probably as oxyhydroxides. We propose that decomposing peat was the

main source of $N_2O$ during WT drawdown in spring, where ammonia oxidation together with chemodenitrification was a likely pathway to $N_2O$ formation. In the autumn, denitrification of $NO_3^-$ derived from residues or decomposing peat following WT rise after heavy rainfall was probably the main pathway. Mitigating $N_2O$ emissions from the acid organic soil investigated here is challenged by the apparent complexity of underlying processes. However, reducing surplus N in the soil, for example by ensuring a vegetation cover throughout the year, and stabilising the WT depth by effective

drainage, are potential strategies for curbing $N_2O$ emissions.

## 6 Acknowledgements

This study received financial support from the Danish Research Council for the project "Sources of $N_2O$ in arable organic soil as revealed by $N_2O$ isotopomers" (DFF – 4005-00448). We would like to thank the dedicated staff involved in field campaigns, including Bodil Stensgaard, Søren Erik Nissen, Sandhya Karki, Kim Johansen, Karin Dyrberg,

Holger Bak and Stig T. Rasmussen. We would also like to acknowledge the support of three farmers hosting the field sites: Poul-Erik Birkbak, Rasmus Christensen and Jørn Christiansen.

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





**Table 1.** Selected characteristics of soil profiles at the four monitoring sites with rotational grass (*RG1*, *RG2*) and arable soil used for potato (*AR1*, *AR2*). All analyses were done in triplicate; results shown represent mean and standard error of two soil profiles ($n = 2$). Soils for analyses were collected in late April except for AVS and CRS (early September). Abbreviations: EC, electrical conductivity; TOC, soil organic carbon; TRFe, total reactive iron; AVS, acid volatile sulfide; CRS, chromium reducible sulfur.

|  | Depth (cm) | pH | EC | TOC (g 100 g$^{-1}$) | Total N (g 100 g$^{-1}$) | C:N ratio | TRFe (mg Fe g$^{-1}$) | AVS (µg S g$^{-1}$) | CRS (µg S g$^{-1}$) |
|---|---|---|---|---|---|---|---|---|---|
| **RG1** | | | | | | | | | |
| Depth 1 | 2.5-7.5 | 5.1 (0.2) | 0.26 (0.10) | 37.4 (0.2) | 1.75 (0.00) | 21.3 | 3.63 (0.11) | 2.51 (0.86) | 155 (62) |
| Depth 2 | 7.5-12.5 | 5.3 (0.1) | 0.15 (0.02) | 38.2 (0.2) | 1.79 (0.01) | 21.3 | 4.03 (0.44) | NA | NA |
| Depth 3 | 17.5-22.5 | 5.3 (0.5) | 0.37 (0.18) | 39.7 (0.3) | 1.80 (0.04) | 22.1 | 4.14 (0.32) | NA | NA |
| Depth 4 | 36-40 | 4.8 (0.1) | 0.55 (0.02) | 43.1 (2.7) | 1.85 (0.03) | 23.3 | 3.04 (0.26) | 2.60 (0.87) | 133 (64) |
| Depth 5 | 47.5-52.5 | 5.1 (0.3) | 0.42 (0.13) | 31.0 (15.6) | 1.47 (0.64) | 21.1 | 2.50 (0.55) | 4.86 (1.07) | 24 (17) |
| Depth 6 | 93-98 | 5.4 (0.0) | 0.51 (0.06) | 0.6 (0.3) | 0.01 (0.01) | ND | 0.14 (0.04) | NA | NA |
| **RG2** | | | | | | | | | |
| Depth 1 | 0-25 | 5 | NA | 19.8 (3.4) | 1.34 (0.13) | 14.8 | 2.29 (0.56) | NA | NA |
| Depth 2 | 25-50 | 5.1 | NA | 8.9 (3.0) | 0.63 (0.23) | 14.2 | 4.48 (NA) | 1.71 (0.00) | 33 (7.3) |
| **AR1** | | | | | | | | | |
| Depth 1 | 2.5-7.5 | 5.0 (0.1) | 0.45 (0.04) | 35.9 (0.1) | 1.81 (0.02) | 19.9 | 4.57 (0.09) | 1.74 (0.02) | 141 (9) |
| Depth 2 | 7.5-12.5 | 5.2 (0.1) | 0.42 (0.06) | 34.2 (0.2) | 1.76 (0.02) | 19.4 | 4.66 (0.15) | NA | NA |
| Depth 3 | 17.5-22.5 | 5.2 (0.1) | 0.34 (0.04) | 41.0 (2.2) | 1.93 (0.11) | 21.3 | 4.99 (0.43) | NA | NA |
| Depth 4 | 36-40 | 4.7 (0.5) | 0.37 (0.05) | 41.1 (5.8) | 1.84 (0.05) | 22.4 | 3.23 (0.41) | 2.17 (0.29) | 49 (3) |
| Depth 5 | 47.5-52.5 | 4.7 (0.3) | 0.48 (0.08) | 5.9 (1.7) | 0.37 (0.13) | 16.3 | 1.19 (0.19) | 1.98 (0.41) | 137 (39) |
| Depth 6 | 93-98 | 5.4 (0.2) | 0.91 (0.03) | 0.3 (0.1) | 0.00 (0.00) | ND | 0.18 (0.02) | NA | NA |
| **AR2** | | | | | | | | | |
| Depth 1 | 0-25 | 5.1 | NA | 33.4 (1.2) | 1.45 (0.03) | 23.1 | 4.11 (0.03) | NA | NA |
| Depth 2 | 25-50 | 5.1 | NA | 38.4 (0.2) | 1.46 (0.02) | 26.2 | 3.78 (0.14) | 1.65 (0.02) | 45 (8) |

ND – Not determined due to TOC and total N concentrations being at the limit of detection.

NA - Not analysed.





**Table 2.** Cumulative emissions of $N_2O$ (kg $N_2O$ ha$^{-1}$) during the spring (99-105 days) and autumn (47-69 days) monitoring period. Estimation for each season was performed using the trapezoidal approximation of the integral of the emission curve. Numbers in parentheses indicate 95% confidence intervals, and significant differences, corrected for multiple testing by the single-step method, are indicated by asterisks. *RG*, rotational grass; *AR*, arable crop (potato); F, fertilised; NF, unfertilised.

| | Cumulative $N_2O$ | | *RG-NF* | *RG-F* | *AR-NF* |
|---|---|---|---|---|---|
| **Spring** | kg ha$^{-1}$ | | | | |
| *RG-NF* | 2.0 | (1.5-2.5) | | | |
| *RG-F* | 7.3 | (4.9-9.6) | ***§ | | |
| *AR-NF* | 17.1 | (13.9-20.2) | *** | *** | |
| *AR-F* | 15.0 | (12.2-17.8) | *** | *** | NS |
| **Autumn** | | | *RG* | | |
| *RG* | 2.2 | (1.6-2.7) | | | |
| *AR* | 14.8 | (11.6-17.9) | *** | | |

§ ***, $p < 0.001$; NS, not significant ($p > 0.05$)





**Figure captions**

**Figure 1.** A. Location of sites *AR1* and *RG1* (both at 57°13'59.7"N, 9°50'40.3E), *RG2* (57°13'55.9"N, 9°52'20.2E) and *AR2* (57°13'7.6"N, 9°46'26.9E). B. Experimental design at each of the four sites, with three blocks centered around piezometers (●) and two subplots, one of which received N fertiliser at the rate of the surrounding field. Six collars for gas flux measurements (S1-S6) were distributed as indicated, and sets of 5 diffusion probes for soil gas sampling were installed near collars in selected positions (see text).

**Figure 2.** Nitrite-N (a, c) and total reactive iron, TRFe (b, d), in undisturbed soil cores collected at sites *RG1* and *AR1* on 23 April (DOY113; white symbols) and 2 September (DOY245; grey symbols). Results shown are mean and standard error ($n = 2$). The dotted lines indicate WT level on the two sampling dates.

**Figure 3.** The top panel shows rainfall, air temperature and management (F – fertilisation) at sites *RG1* (left panels) and *RG2* (right panels) during spring. The middle section shows $N_2O$ fluxes (black circles; mean ± standard error, $n = 3$) and contour plots of soil $N_2O$ concentrations in fertilised subplots, and the lower section the corresponding results for unfertilised subplots. A logarithmic grey scale was used in order to show trends within both *RG* and *AR* treatments, and between depths. Soil gas sampling positions are indicated in the contour plots; numbers shown are $N_2O$ concentrations ($\mu L\ L^{-1}$). Grey lines show the WT depth (which varied slightly between blocks). B2 and B3 refer to block number of diffusion probe positions.

**Figure 4.** The top panel shows rainfall, air temperature and management (T – tillage; F – fertilisation) at sites *AR1* (left panels) and *AR2* (right panels) during spring. The middle section shows $N_2O$ fluxes (black circles; mean ± standard error, $n = 3$) and contour plots of soil $N_2O$ concentrations in fertilised subplots, and the lower section the corresponding results for unfertilised subplots. A logarithmic grey scale was used in order to show trends within both *RG* and *AR* treatments, and between depths. Soil gas sampling positions are indicated in the contour plots; numbers shown are $N_2O$ concentrations ($\mu L\ L^{-1}$). Gaps are indicated where soil gas sampling probes were installed late, or removed due to field operations. Grey lines show the WT depth (which varied slightly between blocks). B2 and B3 refer to block number of diffusion probe positions.

**Figure 5.** The top panel shows rainfall, air temperature and management (H - harvest) at sites *RG1* (left panels) and *RG2* (right panels) during autumn. The middle section shows $N_2O$ fluxes (black circles; mean ± standard error, $n = 3$) and contour plots of soil $N_2O$ concentrations in fertilised subplots, and the lower section the corresponding results for unfertilised subplots. A logarithmic grey scale was used in order to show trends within both *RG* and *AR* treatments, and between depths. Soil gas sampling positions are indicated in the contour plots; numbers shown are $N_2O$ concentrations ($\mu L\ L^{-1}$); the probes were absent in the unfertilised subplot after harvest. Grey lines show the WT depth (which varied slightly between blocks). B2 and B3 refer to block number of diffusion probe positions.

**Figure 6**. The top panel shows rainfall, air temperature and management (H - harvest) at sites *AR1* (left panels) and *AR2* (right panels) during autumn. The middle section shows $N_2O$ fluxes (black circles; mean ± standard error, $n = 3$) and contour plots of soil $N_2O$ concentrations in fertilised subplots, and the lower section the corresponding results for





unfertilised subplots. A logarithmic grey scale was used in order to show trends within both *RG* and *AR* treatments, and between depths. Soil gas sampling positions are indicated in the contour plots; numbers shown are $N_2O$ concentrations ($\mu$L L$^{-1}$). Grey lines show the WT depth (which varied slightly between blocks). B2 and B3 refer to block number of diffusion probe positions.

**Figure 7.** Statistical results from graphical models for the four combinations of crops (*RG*, *AR*) and season (spring, autumn). a. *RG*, spring; b. *RG*, autumn; c. *AR*, spring; and d. *AR*, autumn. The vertices ("points") and edges ("lines") indicate significant relationships between explanatory variables and the response variable, i.e., $N_2O$ flux. Key to variables: AmmoniumT – $NH_4^+$ at 0-25 cm depth; NitrateT – $NO_3^-$ at 0-25 cm depth; $N_2O$ WT – equivalent soil gas phase concentration closest to, but above the water table depth; Temp5 – soil temperature at 5 cm depth; Temp30 – soil temperature at 30 cm depth.



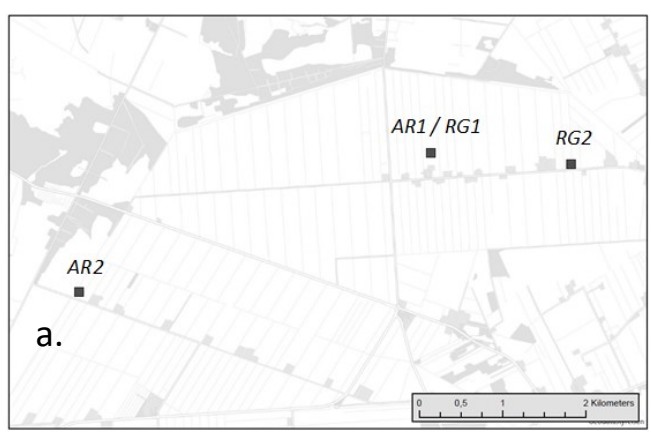

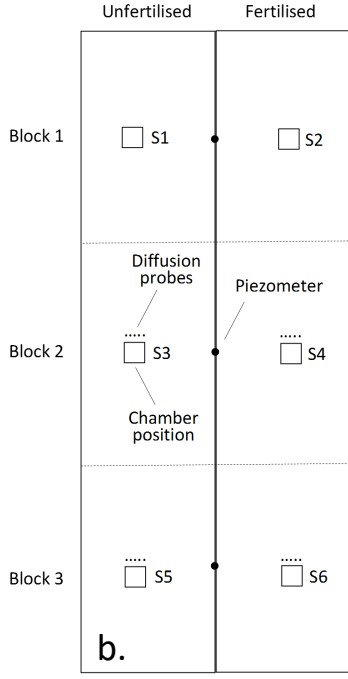

**Figure 1**





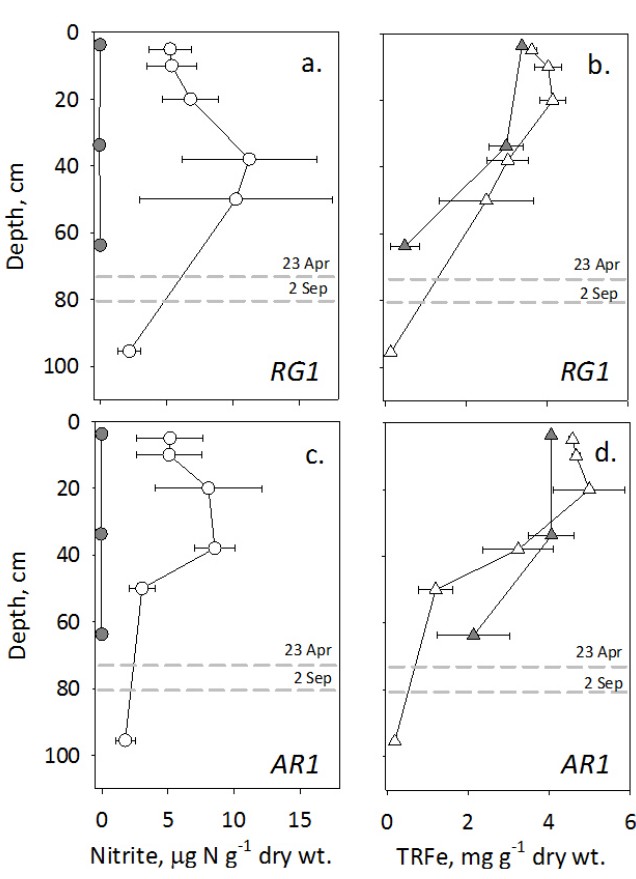

Figure 2.



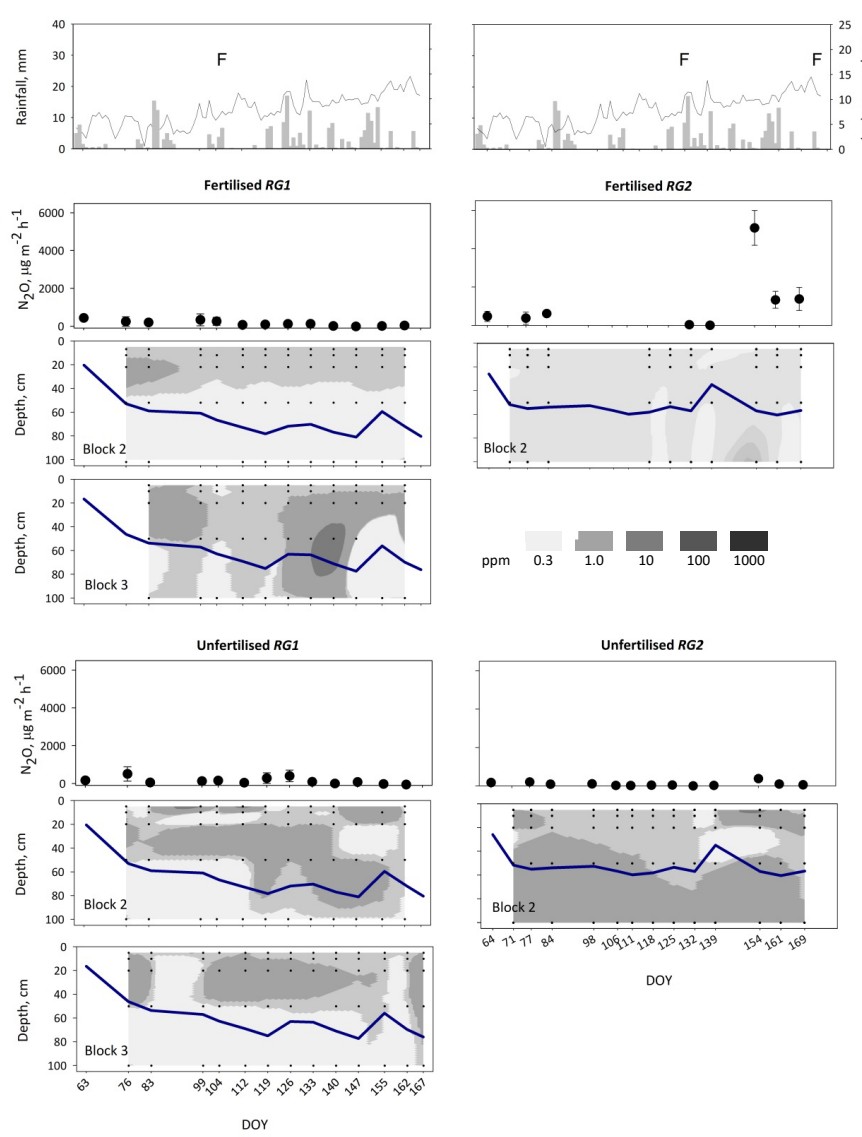

**Figure 3**



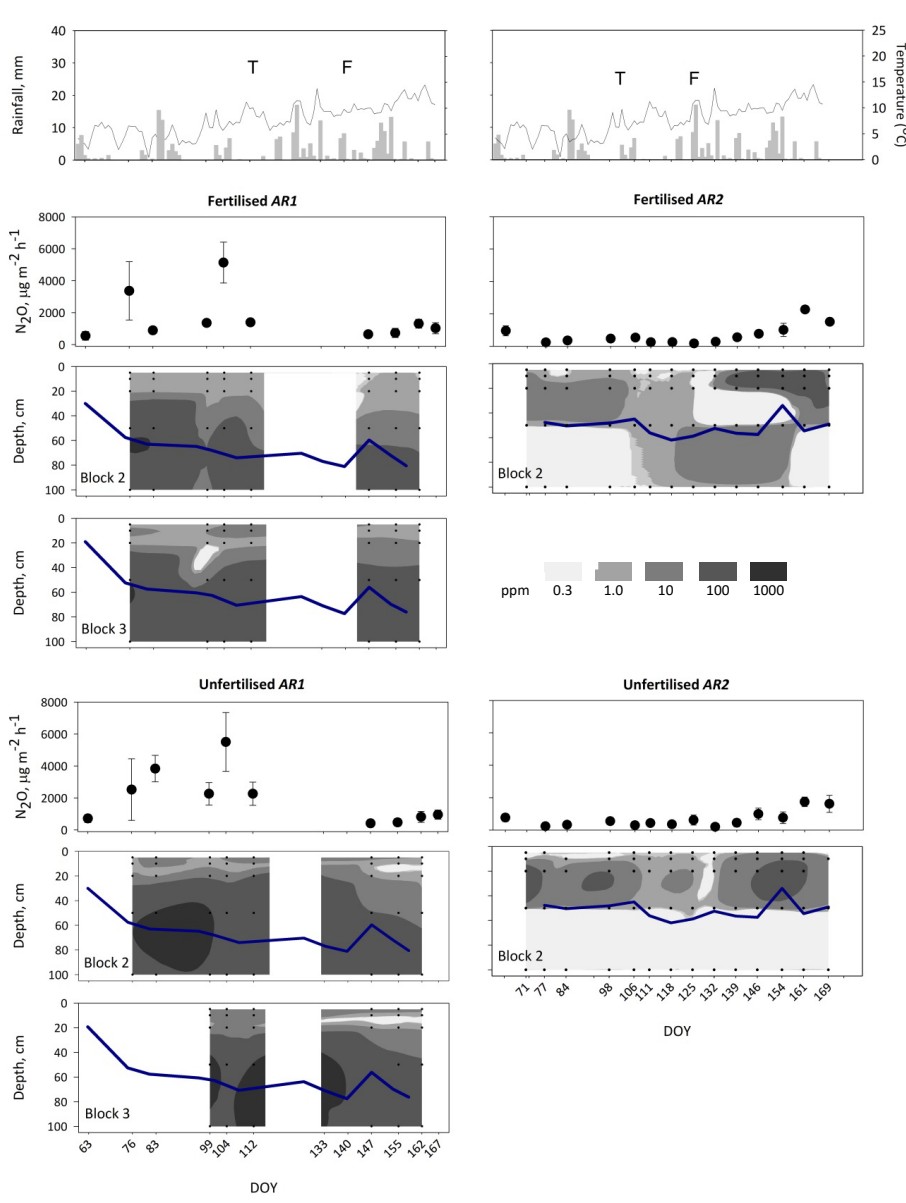

Figure 4





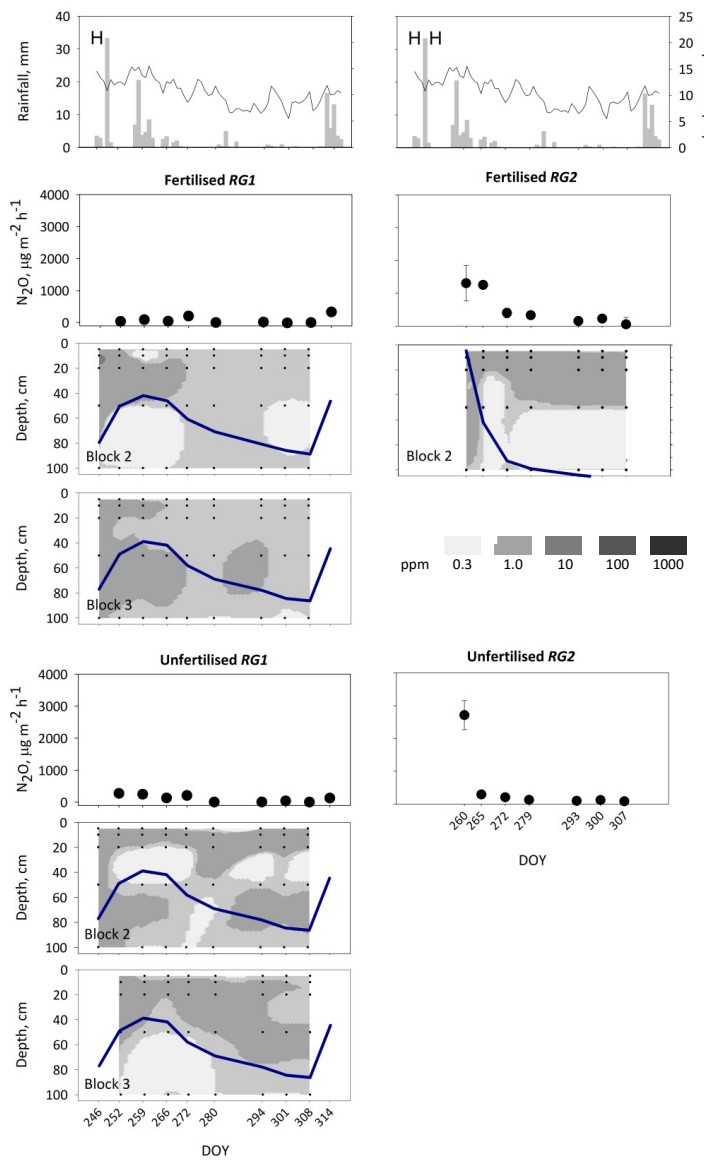

**Figure 5**



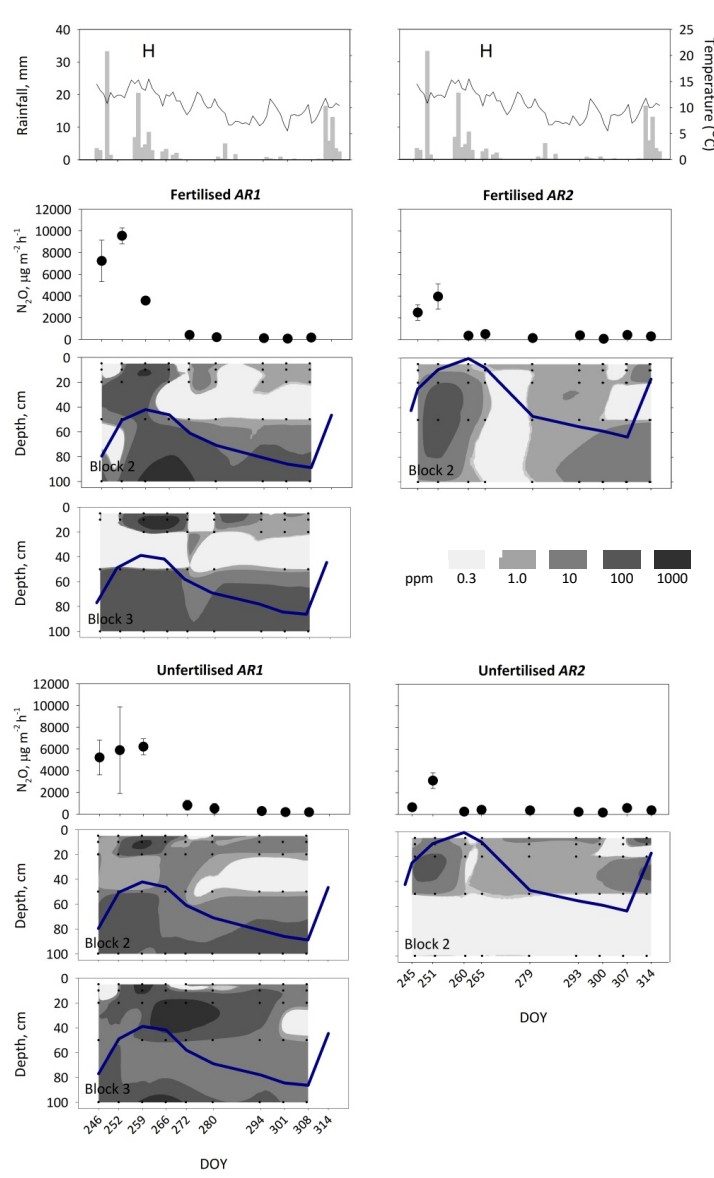

**Figure 6**





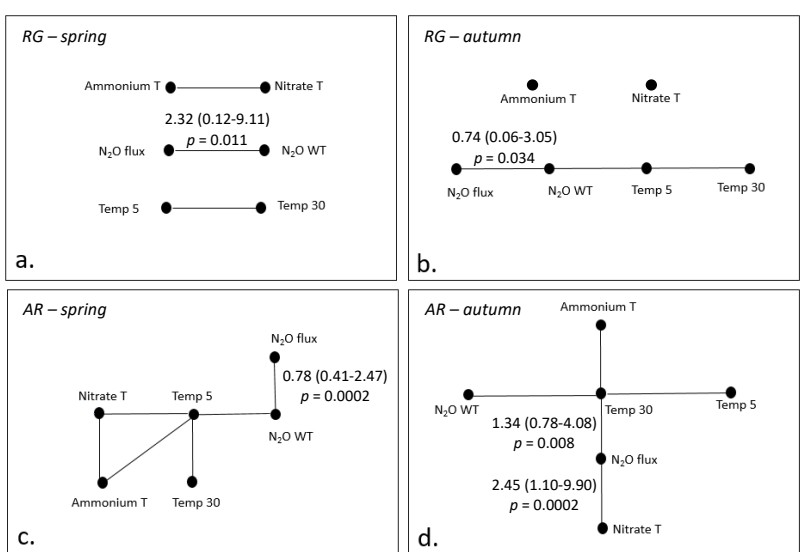

Figure 7