# Peer review of "Regulation of N2O emissions from acid organic soil drained for agriculture"

_Biogeosciences, 2019_

## Referee Comment (RC1) · Anonymous Referee #1 · 28 Feb 2019

This manuscript investigated N2O emissions and concentrations in peat soils under 2 agricultural crops: grassland and potato) at 2 distinct site locations during spring and autumn season of one year only. All combination (site x crop) treatment received different management in terms of fertilisation and harvesting etc. The N2O production measurements were characterized with static chambers and soil N2O diffusion probes placed at 5, 10, 20, 50 and 100 cm depths. All potential environmental factors (climatic or edaphic) were also monitored during this period. This manuscript has been re-submitted and is substantially improved and conclusions are now validated. It would appear that a lot of fieldwork and indeed field data have been processed and are not equally discussed here but is focused on the title of the manuscript. It is well written

and sufficient information provided to allow their reproduction. Some minor comments below would help clarify some details of the experiment and the results.

Line 11: rephrase or change word 'extensively' or 'intensive'; there is nothing extensive about growing cereals or potatoes on organic soils given the cultivation/fertilisation inputs. Perhaps it was meant to be 'widely' used? Line 24: emissions could be given per unit of time, either day or season. Where are those days in terms of season?

Line 79-83 belongs to methods; go straight to your hypothesis questioning the role of crop type and seasonal variation Line 130; the fertilisation treatment is different in each site and therefore do not act as replicate but different treatment. L149 each field trip being a day sampling so 2 sites were sampled per day maximum but all were sampled during the same week? Rephrase please. L258 it is not clear that cumulative N2O emissions are here total or on a daily average. Line 301 : average deviation of soil temp from air temp is given; could it be better described in terms of sign L304. It stats in Lin 166 that soil samples were taken at the start of each season April and Sept? Depth of total peat layer should be shown in Table 1 as it seems that RG2 is very shallow peat (<25cm). Also von post figure should be given for each peat layer. Line 350. The WT reported in Figures 5 & 6 is confusing; what are they if you are not showing your continuous measurements (which shows higher WTL?) Line 407: there in previously in this paragraph, it would help to add the DOY (as per line 410) or else include the month in your Figures. Line 415: this is the first time that the monitoring period is mention; this should be explicitly shown in Table 2 at least and therefore rather than total a per day average would be better to compare treatment.

Figure 3-6: the WT is visible in blue not in grey. Figure 7 : the statistical number on the graphics should be explained in the legend since it is not clear to which lines they apply (especially 7a).

[Figure]

---

## Referee Comment (RC2) · Anonymous Referee #2 · 21 Mar 2019

The authors evaluate subsoil concentrations and emissions of nitrous oxide (N2O) and related soil variables at sites in a raised bog in Denmark which had been identified as having high potential N2O emissions. They conduct measurements over spring and fall, 2015, in four sites (two of which are immediately adjacent to one another): two cropped with potato and two with grasses; each site was also split into fertilized and unfertilized treatments. Using graphical analysis, they determine that in most sites, the concentration of N2O at the capillary fringe was the driver of emissions, though in one season/site combination, N2O emissions were related to sub-surface soil temperature and nitrate concentration. The relationship between water table depth and N2O appeared to vary by site and by season, but included a declining water table depth triggering N2O emissions in spring, and rising water table depth triggering N2O emissions in autumn. Pyrite was largely excluded as making an important contribution to N2O production or emission, and a range of possible mechanisms of N2O production were discussed. Lower emissions from grassland plots were attributed to greater plant uptake of soil N. Annual fluxes were comparable to or higher than IPCC Tier I emission factors for drained organic soils.

**General comments:**

This is my first reading of the manuscript. This is an impressive field study with a lot of interesting data, and the importance of N2O concentrations in the capillary fringe for N2O emissions is a nice result, as is the finding that pyrite and iron monosulfide are unimportant, and the confirmation of high emissions from these soils. However, I have a number of concerns regarding the experimental design, support for some of the conclusions presented, and organization of the manuscript. The manuscript and discussion in particular could benefit from substantial revision after some contemplation of what the core advances of the study are. There seem to be three main aspects of the study (in no particular order): 1. Understanding the mechanism of N2O production, with a particular examination of the potential roles of pyrite and iron monosulfide. 2. Understanding the variables related to surface N2O fluxes. 3. Understanding the relationship between water table and soil N2O concentrations (and N2O flux). These different aspects are part of a single system, and ideally the paper will weave these aspects together into a single coherent story of the patterns and mechanisms behind N2O dynamics and its drivers. As noted below, I think it may make sense to refocus this manuscript around results related to items 2 and 3, for three reasons: a) it may help to clarify this study, b) results apparently highly relevant to item 1 have been kept for a second manuscript, and c) this study wasn't designed to be a comprehensive investigation of the specific mechanism of N2O production in these soils.

Based on the first referee's review, it sounds as though this may be a revised version of an earlier manuscript; if so, sorry to be bringing new concerns at this stage of the publication process, and apologies for the long comments. Note: I don't have expertise in graphical analysis or in the specific technique used to measure subsurface N2O concentrations.

Specific comments:

- ■ Experimental design:
  - o The experimental design is unusual, with elements of different types of partly nested designs. Essentially, the first arable potato site (AR1) and first rotational grass site (RG1) are part of what traditionally would be called a split-split-plot design, but RG2 and AR2 are in a split-plot design. What this means is that the role of site in the analysis—there are effectively three sites--varies in its relationship with the treatments. AR1 and RG2 are nested within site, but RG2 and AR2 are not. Additionally, the lack of nesting of AR2 and RG2 within site raises additional issues, since RG2 has such a big difference in organic C content, so treatment is confounded with site in the RG2/AR2 pairing (in addition to the different fertilizer types used). Although differences between the potato and grass treatments are discussed in the results, and the title suggests that these and seasonal differences are the focus of the manuscript, I don't see any methodological description of the analysis that could allow one to compare RG and AR treatments. I'm not strictly sure how it would be done, but I also don't have a problem with softening the wording of the conclusions that can be drawn about the differences between RG and AR treatments—it does look like there may well be treatment differences there, I'm just not sure that with this design that it's possible to establish that statistically. So I think it needs to be made clear that if a strict statistical comparison was not conducted, the conclusions drawn aren't statistically supported (and any related concluding statements should be softened). And if a statistical comparison was conducted, details on how the nesting within site of RG1 and AR1 but not AR2 and RG2 was handled would be helpful. There is also no replication of bog—the results can't technically be generalized beyond the Store Vildmose raised bog. This is not a problem, and the authors don't attempt to extrapolate beyond the Store Vildmose, but it is a limitation that should be explicitly acknowledged. None of these issues should affect the finding that the capillary fringe is often/typically the primary determinant of the magnitude of N2O flux.
  - o I am always concerned about field studies the present just one year of data, as it does limit insights into the degree to which patterns observed provide generalizable insights. The manuscript title suggests that comparisons of seasons is one of the central findings of the study, but since only one year of data are included, it is impossible to rigorously compare seasonal differences, as there is no replication of season. There are a lot of varying results in this study, which only strengthen this issue. I don't think these issues affect the finding on the frequent importance of capillary fringe N2O concentrations for surface emissions, though.
  - o In the end, these issues might make the conclusions of this study fairly descriptive with respect to the specific questions of the effects of season and land use on N2O emissions and subsurface dynamics. That doesn't mean that these specific results aren't valuable, just that the nature and limitations of the conclusions that can be drawn from the study on these specific questions need

to be made very clear in the manuscript. Rethinking what the central finding or findings of the manuscript are may be helpful.

- The data on the changes in the water table and subsoil concentrations of N2O are great. But the conclusions drawn in section 4.3 could be more compellingly supported (there does not appear to be any quantitative analysis of the relationship) and discussed. Looking at the figures, in some cases N2O concentrations are enhanced above the water table, in other cases, below the water table, and an overall relationship is not immediately obvious. The importance of the capillary fringe concentrations for surface emissions makes this discussion of particular interest, and worth spending some text to guide the reader through your interpretation.

- The manuscript often reports whether there was an effect of fertilization (e.g., line 380, lines 404-405, 412-413, and others) or appears to test fertilizer vs site effects (e.g., line 405), and the methods detail how generalized linear mixed models were used to analyze the temporal dynamics of N2O. However, I don't see any reporting of the statistical results of this model or its application to the impact of fertilization on N2O emissions or soil concentrations.

- A follow-on point is the very nice finding that N2O at the capillary fringe generally controlled N2O emissions, rather than any variables in the topsoil. But this result raises the question of what controls variation in N2O concentration at the capillary fringe. This question seems to be of first order importance in this system, but is not addressed quantitatively in the manuscript. It might help tighten the manuscript if the discussion in section 4.3 is tied more explicitly to N2O concentrations in the capillary fringe.

- A general comment: In presenting results, it could be helpful to start each section with a general description of the main results or patterns found instead of starting with detailed information for individual blocks; that detailed information can be presented later, to support the general patterns or describe deviations from those patterns. Anything you can do to guide readers through the results is great! In addition, the authors occasionally slip interpretation into the results section that would be more appropriate in the discussion section.

- One of the main questions addressed in this manuscript is that of the importance of FeS2 oxidation for N2O production. Line 530-531 invokes a separate but presumably related manuscript that presents results showing that FeS2 oxidation is unimportant in this peat soil. It is difficult to say without knowing what the focus of that manuscript is, but my hunch is that it may make more sense to include the FeS2 results from this field study in the separate manuscript (presumably focused on mechanisms of N2O production in these soils), since we are effectively only getting half the story here. Something to consider, anyway. In the end, this manuscript doesn't provide much in the way of firm insights into the mechanisms of N2O production—that's not an inherent problem, it's just not something this study was designed to do--so one idea would be to cut out that part of this manuscript, and make the focus entirely on quantification of fluxes, the nice soil N2O & water table data (and capillary fringe finding), and environmental drivers more generally. It would be easy enough then to include a paragraph on mechanisms of N2O production that cites the other manuscript. Also, just to note, many journals require any related manuscripts that have been submitted

elsewhere to be included as part of a manuscript submission, so would be a good idea to check the policy of the journal in question when you submit the separate manuscript.

- The manuscript argues that the water table depth was related to surface N2O emissions (e.g., section 4.1), but that this relationship varied by site and by season, and speculates that soil properties modified the water table depth/N2O relationship. However, there doesn't seem to be any statistical/quantitative analysis to support a water table depth/N2O relationship or how other soil properties modify that relationship (and as noted above, it's not possible to statistically evaluate seasonal differences).

- section 4.1 is largely a re-statement of results; much of the actual discussion about the water table/N2O relationship is sprinkled throughout subsequent sections. A restructuring of the discussion might make the results easier to digest, with one section focused on discussing the capillary fringe result and one focused on understanding the water table/N2O relationship. Some discussion about why the patterns are so variable could be valuable, including some explanation of why water table increases stimulate N2O production at all sites in the autumn (and contrast with the results of other studies, e.g. Maljanen et al. 2003, which saw no effect of rising water table on N2O emission). One possible straw man interpretation could be that in the early spring (or late spring in the case of AR2), N2O production is limited by NH4+ (and/or NO2-/NO3-) availability, which in turn is constrained by the availability of O2. The decline of the water table may release the O2 constraint. In the autumn, in contrast, it is possible that aerobic conditions limit N2O production, and a rising water table or precipitation leads to higher N2O emissions. If indeed the case, why a possible seasonal shift from substrate to O2 limitation happens would be interesting to understand.

**Technical comments:**
**Line 1: I would change the title to reflect the focus of the revised manuscript (in addition, as noted above, it seemed to me that neither seasonal nor land use differences were able to be rigorously tested, and so it would be better to exclude phrases like "effects of land use and season" from the title)**
**Line 45: You can check my math, but it seems to me that the global warming potential is still uniformly larger for C than for N2O here. I don't think it is necessary to make the case that N2O fluxes are more important than carbon fluxes, just that the N2O fluxes are large.**
**Line 44: change ", which" to "that"**
**Line 64: "The sites" : not sure what sites are being referred to. Could you add more context?**
**~Line 120: please add a short explanation for why soil gas data were not presented for unfertilized RG2 during Autumn**
**Line 167: was this the fertilized or unfertilized block?**
**Line 176: remove "quantitatively" (not sure what it is intended to mean)**
**Line 210-211: Entirely your choice, but perhaps AVS and CRS don't need to be abbreviated**
**Line 197: specify type of filter paper used**
**Line 237: Is the instrument ever checked against a set of standards of varying concentrations?**
**Line 339: Indicate whether fertilized or unfertilized blocks were sampled**

Line 354 change "temporarily" to "temporary"

Line 363: change "trends" to "concentrations"

Line 370: Figure 3 seems to suggest that the N2O concentrations in the top 40cm of soil look to be 1-2 orders of magnitude higher in the fertilized RG1 than fertilized RG2. And unfertilized RG2 looks to be 1-2 orders of magnitude higher than unfertilized RG1 between 60 and 100cm depth for most of the spring. Yet they are described as "generally similar." I wouldn't have thought that would be considered "generally similar"—am I missing something?

Line 375-6: Since there are apparently no soil gas data from unfertilized plots in RG2 during autumn, this statement is too strongly worded (even independent of questions of whether the effect of fertilization was tested).

Line 376: The figures are out of order—I think you can swap Figures 4 and 5.

Line 380: could you be specific about what soil conditions showed significant within-site heterogeneity? Also, use "substantial" instead of "significant' if this heterogeneity wasn't tested, and if it was, consider providing P values

Line 384: this is really interpretation, and might be better placed in the discussion.

Line 400: were any measurements made of N in harvested biomass? It could certainly help support the story that differences in uptake could alter N2O emissions from different plots.

Line 405: if specific soil variables cannot be identified as causing the differences, perhaps change "soil conditions" to "site differences"

Line 411-12: I'm not sure I see this pattern clearly: emissions are already high when the water table is at 80, and in the fertilized plots of AR1, emissions are 1/3 as large on DOY 259 than DOY 252, even though the water table is at its peak on DOY 259. There's also no apparent effect of the increase in water table starting on DOY 307, and emissions look quite elevated on DOY 246, which may be before the increase in water table began. A quantitative analysis would be helpful.

Lines 473-4: could be more specific and change "a short period" to "1 to 2 weeks"

Line 481 and following: Section 4.3 draws a number of conclusions that don't appear to be supported by any statistical analyses.

Line 516-517: I'm not sure I see this rapid increase in N2O around the water table depth in all the blocks in figure 6?

Lines 533-534: I think it might be better to say that "denitrification in topsoil was the main source. . . " since there is no explanation of how the N2O in the capillary fringe is produced.

Line 544: I'd change "drive" to "regulate".

Line 532 and following: this is interesting discussion, but if there are supplementary data that could support application of the ideas to this study (e.g., water filled pore space, acetylene reduction experiments, etc), it would strengthen it considerably. If the manuscript in preparation on FeS2 oxidation includes any detailed examinations of these questions, it may be better to limit the speculation here.

Line 544-558: looking at Tables S1 and S2, it seems that there is generally more NO3- or NH4+ in these soils on April 22 and/or May 13 than there is NO2- on April 23 (much more if these were the fertilized plots—I could not see any indication of whether the undisturbed core was from fertilized or unfertilized plots). If correct, that suggests that perhaps there is not an imbalance between ammonia oxidation and nitrite oxidation? Perhaps all nitrifier

populations are temporarily saturated by the increase in available NH4+?  The discussion in Lines 486-489 also seems to suggest that denitrification was cranking along pretty well in the AR sites.  And perhaps there's reason to be cautious about inferring processes from snapshots of concentrations, whether a single depth profile or weekly measurements of NH4+ and NO3-.  Presumably, high concentrations could indicate anything from slow loss rates of each compound (whatever the pathway may be), or could reflect rapid N mineralization rates.  If, by chance, total N concentrations were measured at each sampling date, calculations of net mineralization and net nitrification might be able to provide additional insight into whether and where reactive N might be accumulating.

Line 550-51: This is partly covered in the note immediately above.  I see that NH4+ remains at high concentrations, but NO3- does as well, which is why I'm unsure about the suggestion that there is a lack of a mechanism to remove NO2-.

Figure 2 caption: indicate whether cores were taken from fertilized or unfertilized blocks.

Figures 3 through 7: I think it might be easier to evaluate these data if the entire year of data are presented in a single plot, rather than separating spring and fall data—I don't think it would make it any more difficult to read the data.  An axis break could be included between DOY 167 and DOY 246.  You could also explore presenting surface N2O flux in a log scale—there may be a variation that would be visible on a log scale that is difficult to discern on the current linear scale because of the dates with very high fluxes.

The manuscript text switches freely between using DOY, month, and terms like "early spring" to describe time, which makes it challenging for the reader to compare the text and figures.  Sometimes the DOY is included parenthetically when month names are used, which is great, but this practice should be extended throughout the text.  Alternatively, the x axis labels could be changed to month names and days.

Tables S1 through S4 would be much easier to read in figure form (possibly in a single figure), though I appreciate the inclusion of the summary data here.  Actually, why not explore adding these data as a second y axis in figures 3-6, sharing the panels used for N2O.  Since topsoil nitrate is presented as a significant predictor of N2O flux, it could be valuable to be able to compare the data in the figures.

---

## Author Comment (AC1) · 20 Apr 2019

I have copy the response letter below. However, the comment and responses to both reviewers' comments was uploaded in the form of a supplement, in pdf file.
This manuscript investigated N2O emissions and concentrations in peat soils under 2 agricultural crops: grassland and potato) at 2 distinct site locations during spring and

autumn season of one year only. All combination (site x crop) treatment received different management in terms of fertilisation and harvesting etc. The N2O production measurements were characterized with static chambers and soil N2O diffusion probes placed at 5, 10, 20, 50 and 100 cm depths. All potential environmental factors (climatic or edaphic) were also monitored during this period. This manuscript has been resubmitted and is substantially improved and conclusions are now validated. It would appear that a lot of fieldwork and indeed field data have been processed and are not equally discussed here but is focused on the title of the manuscript. It is well written and sufficient information provided to allow their reproduction. Some minor comments below would help clarify some details of the experiment and the results. Response 1: Thank you for this positive feedback and helpful review. We have carefully considered all comments, and below we respond in more detail.

Line 11: rephrase or change word 'extensively' or 'intensive'; there is nothing extensive about growing cereals or potatoes on organic soils given the cultivation/fertilisation inputs. Perhaps it was meant to be 'widely' used? Response 2: We will change the word 'extensively' to 'widely' to avoid ambiguity.

Line 24: emissions could be given per unit of time, either day or season. Where are those days in terms of season? Response 3: The sentence will be modified for clarity. We propose the following wording: "Spring and autumn monitoring periods together accounted for 152-174 days, and during this time the cumulative N2O emissions were 3-6 kg N2O-N ha-1 for rotational grass, and 19-21 kg N2O-N ha-1 for potato sites."

Line 79-83 belongs to methods; go straight to your hypothesis questioning the role of crop type and seasonal variation Response 4: Thank you for this suggestion. We would like to keep the first sentence to establish the context, but agree to remove the detailed information about measurements (Line 81-83).

Line 130; the fertilisation treatment is different in each site and therefore do not act as replicate but different treatment. Response 5: It is true that, in order to follow the actual

management at each site, different fertiliser types and rates were used. However, the fertiliser treatment was only represented in the statistical analysis as a categorical variable, i.e., for testing if N fertilisation of, grassland and the potato crop influenced the emission of N2O.

L149 each field trip being a day sampling so 2 sites were sampled per day maximum but all were sampled during the same week? Rephrase please. Response 6: We propose to rephrase the text as follows: "Field trips included sampling at two sites, either AR1 + RG1 or AR2 + RG2, and thus all four sites were visited during two field trips on consecutive days."

L258 it is not clear that cumulative N2O emissions are here total or on a daily average. Response 7: We propose the following rewording to clarify: "The model for daily N2O emission described above was used to estimate cumulative emissions by integrating the flux curves over time. Treatment effects were then analysed by specially designed linear contrasts as described in detail by Duan et al. (2017), who showed that models with untransformed responses (when using adequate distributions) allow simple statistical inference of the time-integrated N2O emissions."

Line 301 : average deviation of soil temp from air temp is given; could it be better described in terms of sign L304. Response 8: It is not quite clear if some text is missing here. We report average deviations, as well as the largest positive and negative deviations observed.

It stats in Lin 166 that soil samples were taken at the start of each season April and Sept? Response 9: The soil characteristics reported in Table 1 were based on analyses of soil cores sampled in April except for AVS and CRS, which were analysed with soil cores sampled in September (as stated in the Table caption). This will be specified in the text also.

Depth of total peat layer should be shown in Table 1 as it seems that RG2 is very shallow peat ( (<25cm). Also von post figure should be given for each peat layer.

Response 10: Unfortunately peat depth was not determined (but general information given in L. 95), and as stated the degradation at site RG2 was extensive. We will refer to von Post values determined at a grassland site located on the same East-West axis as the four monitoring sites of this study (Schäfer et al., 2012. Seasonal methane dynamics in three temperate grasslands. Plant and Soil 357: 339-353).

Line 350. The WT reported in Figures 5 & 6 is confusing; what are they if you are not showing your continuous measurements (which shows higher WTL?) Response 11: The WT reported in Figures 5 and 6 were the weekly values measured concurrently with flux measurements. By referring to the continuous measurements, we wanted to stress that soil conditions were highly dynamic, and that the nitrate reduction potential could have been influenced by this. For consistency across seasons, we would prefer to show the values obtained at the time of flux measurements in all contour plots.

Line 407: there in previously in this paragraph, it would help to add the DOY (as per line 410) or else include the month in your Figures. Response 12: We will add this information, so that the last sentence reads: "The highest emissions occurred, independent of fertilisation, in June when a WT rise to 35 cm depth was observed on DOY154."

Line 415: this is the first time that the monitoring period is mention; this should be explicitly shown in Table 2 at least and therefore rather than total a per day average would be better to compare treatment. Response 13: Please note that the overall monitoring periods in spring and autumn were defined in Line 143. We will include the specific DOY for each site in Table 2.

Figure 3-6: the WT is visible in blue not in grey. Response 14: We acknowledge that black and white reproduction will make it difficult to see WT lines in AR plots with dark grey areas. We will consider an alternative colour combination.

Figure 7 : the statistical number on the graphics should be explained in the legend since it is not clear to which lines they apply (especially 7a). Response 15: Thank you for the comment. We will revise the legend and consider a way to clarify further the

link between lines and statistical information.

Please also note the supplement to this comment:
https://www.biogeosciences-discuss.net/bg-2019-14/bg-2019-14-AC1-supplement.pdf

———————————————————

[Figure]

**Supplement:**

This manuscript investigated N$_2$O emissions and concentrations in peat soils under 2 agricultural crops: grassland and potato) at 2 distinct site locations during spring and autumn season of one year only. All combination (site x crop) treatment received different management in terms of fertilisation and harvesting etc. The N$_2$O production measurements were characterized with static chambers and soil N$_2$O diffusion probes placed at 5, 10, 20, 50 and 100 cm depths. All potential environmental factors (climatic or edaphic) were also monitored during this period. This manuscript has been resubmitted and is substantially improved and conclusions are now validated. It would appear that a lot of fieldwork and indeed field data have been processed and are not equally discussed here but is focused on the title of the manuscript. It is well written and sufficient information provided to allow their reproduction. Some minor comments below would help clarify some details of the experiment and the results.

Response 1: Thank you for this positive feedback and helpful review. We have carefully considered all comments, and below we respond in more detail.

Line 11: rephrase or change word 'extensively' or 'intensive'; there is nothing extensive about growing cereals or potatoes on organic soils given the cultivation/fertilisation inputs. Perhaps it was meant to be 'widely' used?

Response 2: We will change the word 'extensively' to 'widely' to avoid ambiguity.

Line 24: emissions could be given per unit of time, either day or season. Where are those days in terms of season?

Response 3: The sentence will be modified for clarity. We propose the following wording:
"Spring and autumn monitoring periods together accounted for 152-174 days, and during this time the cumulative N$_2$O emissions were 3-6 kg N$_2$O-N ha$^{-1}$ for rotational grass, and 19-21 kg N$_2$O-N ha$^{-1}$ for potato sites."

Line 79-83 belongs to methods; go straight to your hypothesis questioning the role of crop type and seasonal variation

Response 4: Thank you for this suggestion. We would like to keep the first sentence to establish the context, but agree to remove the detailed information about measurements (Line 81-83).

Line 130; the fertilisation treatment is different in each site and therefore do not act as replicate but different treatment.

Response 5: It is true that, in order to follow the actual management at each site, different fertiliser types and rates were used. However, the fertiliser treatment was only represented in the statistical analysis as a categorical variable, i.e., for testing if N fertilisation of, grassland and the potato crop influenced the emission of $N_2O$.

L149 each field trip being a day sampling so 2 sites were sampled per day maximum but all were sampled during the same week? Rephrase please.

Response 6: We propose to rephrase the text as follows:

"Field trips included sampling at two sites, either *AR1 + RG1* or *AR2 + RG2*, and thus all four sites were visited during two field trips on consecutive days."

L258 it is not clear that cumulative N2O emissions are here total or on a daily average.

Response 7: We propose the following rewording to clarify:

"The model for daily $N_2O$ emission described above was used to estimate cumulative emissions by integrating the flux curves over time. Treatment effects were then analysed by specially designed linear contrasts as described in detail by Duan et al. (2017), who showed that models with untransformed responses (when using adequate distributions) allow simple statistical inference of the time-integrated $N_2O$ emissions."

Line 301 : average deviation of soil temp from air temp is given; could it be better described in terms of sign L304.

Response 8: It is not quite clear if some text is missing here. We report average deviations, as well as the largest positive and negative deviations observed.

It stats in Lin 166 that soil samples were taken at the start of each season April and Sept?

Response 9: The soil characteristics reported in Table 1 were based on analyses of soil cores sampled in April except for AVS and CRS, which were analysed with soil cores sampled in September (as stated in the Table caption). This will be specified in the text also.

Depth of total peat layer should be shown in Table 1 as it seems that RG2 is very shallow peat ( (<25cm). Also von post figure should be given for each peat layer.

Response 10: Unfortunately peat depth was not determined (but general information given in L. 95), and as stated the degradation at site *RG2* was extensive. We will refer to von Post values determined at a grassland site located on the same East-West axis as the four monitoring sites of this study (Schäfer et al., 2012. Seasonal methane dynamics in three temperate grasslands. Plant and Soil 357: 339-353).

Line 350. The WT reported in Figures 5 & 6 is confusing; what are they if you are not showing your continuous measurements (which shows higher WTL?)

Response 11: The WT reported in Figures 5 and 6 were the weekly values measured concurrently with flux measurements. By referring to the continuous measurements, we wanted to stress that soil conditions were highly dynamic, and that the nitrate reduction potential could have been influenced by this. For consistency across seasons, we would prefer to show the values obtained at the time of flux measurements in all contour plots.

Line 407: there in previously in this paragraph, it would help to add the DOY (as per line 410) or else include the month in your Figures.

Response 12: We will add this information, so that the last sentence reads:

"The highest emissions occurred, independent of fertilisation, in June when a WT rise to 35 cm depth was observed on DOY154."

Line 415: this is the first time that the monitoring period is mention; this should be explicitly shown in Table 2 at least and therefore rather than total a per day average would be better to compare treatment.

Response 13: Please note that the overall monitoring periods in spring and autumn were defined in Line 143.  We will include the specific DOY for each site in Table 2.

Figure 3-6: the WT is visible in blue not in grey.

Response 14: We acknowledge that black and white reproduction will make it difficult to see WT lines in *AR* plots with dark grey areas. We will consider an alternative colour combination.

Figure 7 : the statistical number on the graphics should be explained in the legend since it is not clear to which lines they apply (especially 7a).

Response 15: Thank you for the comment. We will revise the legend and consider a way to clarify further the link between lines and statistical information.
The authors evaluate subsoil concentrations and emissions of nitrous oxide (N2O) and related soil variables at sites in a raised bog in Denmark which had been identified as having high potential N2O emissions. They conduct measurements over spring and fall, 2015, in four sites (two of which are immediately adjacent to one another): two cropped with potato and two with grasses; each site was also split into fertilized and unfertilized treatments. Using graphical analysis, they determine that in most sites, the concentration of N2O at the capillary fringe was the driver of emissions, though in one season/site combination, N2O emissions were related to sub-surface soil temperature and nitrate concentration. The relationship between water table depth and N2O appeared to vary by site and by season, but included a declining water table depth triggering N2O emissions in spring, and rising water table depth triggering N2O emissions in autumn. Pyrite was largely excluded as making an important contribution to N2O production or emission, and a range of possible mechanisms of N2O production were discussed. Lower emissions from grassland plots were attributed to greater plant uptake of soil N. Annual fluxes were comparable to or higher than IPCC Tier I emission factors for drained organic soils.

Response 16: This is a concise summary of the study. Thank you for the many detailed comments, which we have tried to address below. There was a request from a reviewer of the previous version of the manuscript to provide the detailed account of potential mechanisms of $N_2O$ production, including the possible role of pyrite. Therefore, we would like to keep it as part of overall rational of our study.

General comments: This is my first reading of the manuscript. This is an impressive field study with a lot of interesting data, and the importance of N2O concentrations in the capillary fringe for N2O emissions is a nice result, as is the finding that pyrite and iron monosulfide are unimportant, and the confirmation of high emissions from these soils. However, I have a number of concerns regarding the experimental design, support for some of the conclusions presented, and organization of the manuscript. The manuscript and discussion in particular could benefit from substantial revision after some contemplation of what the core advances of the study are. There seem to be three main aspects of the study (in no particular order): 1. Understanding the mechanism of N2O production, with a particular examination of the potential roles of pyrite and iron monosulfide. 2. Understanding the variables related to surface N2O fluxes. 3. Understanding the relationship between water table and soil N2O concentrations (and N2O flux). These different aspects are part of a single system, and ideally the paper will weave these aspects together into a single coherent story of the patterns and mechanisms behind N2O dynamics and its drivers. As noted below, I think it may make sense to refocus this manuscript around results related to items 2 and 3, for three reasons: a) it may help to clarify this study, b) results apparently highly relevant to item 1 have been kept for a second manuscript, and c) this study wasn't designed to be a comprehensive investigation of the specific mechanism of N2O production in these soils. Based on the first referee's review, it sounds as though this may be a revised version of an earlier manuscript; if so, sorry to be bringing new concerns at this stage of the publication process, and apologies for the long comments. Note: I don't have expertise in graphical analysis or in the specific technique used to measure subsurface N2O concentrations.

Response 16: Thank you for considering an alternative structure for the paper that would eliminate aspects related to the possible role of iron sulfides for $N_2O$ formation. The wider study was planned to evaluate this hypothesis, which was first introduced in a previous study published in BGS (Petersen et al., 2012. Annual emissions of $CH_4$ and $N_2O$, and ecosystem respiration, from eight organic soils in Western Denmark managed by agriculture. Biogeosciences 8, 403-422). The present study was intended to provide field observations as reference for controlled manipulation experiments described in a separate submission. Furthermore, the detailed account of potential mechanisms now included meets a request from a reviewer of the previous version of this manuscript. We therefore find that this is important to account for the overall rationale of our work.

Specific comments:

Experimental design:

- The experimental design is unusual, with elements of different types of partly nested designs. Essentially, the first arable potato site (AR1) and first rotational grass site (RG1) are part of what traditionally would be called a split-split-plot design, but RG2 and AR2 are in a split-plot design. What this means is that the role of site in the analysis—there are effectively three sites--varies in its relationship with the treatments. AR1 and RG2 are nested within site, but RG2 and AR2 are not. Additionally, the lack of nesting of AR2 and RG2 within site raises additional issues, since RG2 has such a big difference in organic C content, so treatment is confounded with site in the RG2/AR2 pairing (in addition to the different fertilizer types used). Although differences between the potato and grass treatments are discussed in the results, and the title suggests that these and seasonal differences are the focus of the manuscript, I don't see any methodological description of the analysis that could allow one to compare RG and AR treatments. I'm not strictly sure how it would be done, but I also don't have a problem with softening the wording of the conclusions that can be drawn about the differences between RG and AR treatments—it does look like there may well be treatment differences there, I'm just not sure that with this design that it's possible to establish that statistically. So I think it needs to be made clear that if a strict statistical comparison was not conducted, the conclusions drawn aren't statistically supported (and any related concluding statements should be softened). And if a statistical comparison was conducted, details on how the nesting within site of RG1 and AR1 but not AR2 and RG2 was handled would be helpful. There is also no replication of bog—the results can't technically be generalized beyond the Store Vildmose raised bog. This is not a problem, and the authors don't attempt to extrapolate beyond the Store Vildmose, but it is a limitation that should be explicitly acknowledged. None of these issues should affect the finding that the capillary fringe is often/typically the primary determinant of the magnitude of N2O flux.

  Response 17: The description of treatment distributions is accurate, but we would like to maintain that sites *RG1* and *AR1* should be considered as independent fields with individual cropping histories (even if fields were part of the same crop rotation, as explained in Petersen et al 2012). In accordance with this, there were some differences in soil variables (Table 1), and this should not have been the case if this was a nested design, as proposed.

While we think that the analysis is valid as conducted, we do not wish to put strong emphasis on land use differences, which were evident even without any statistical support, and we will soften the wording on this contrast throughout the manuscript, as proposed.

- I am always concerned about field studies the present just one year of data, as it does limit insights into the degree to which patterns observed provide generalizable insights. The manuscript title suggests that comparisons of seasons is one of the central findings of the study, but since only one year of data are included, it is impossible to rigorously compare seasonal differences, as there is no replication of season. There are a lot of varying results in this study, which only strengthen this issue. I don't think these issues affect the finding on the frequent importance of capillary fringe N2O concentrations for surface emissions, though.

  Response 18: Please note that this study was preceded by a monitoring study of wider scope (Petersen et al., 2012), where $N_2O$ emissions were monitored during 14 months in three regions and on three (in one case: two) land uses, which were arable farming, rotational grass and permanent grass. In total 8 site-years were thus available as background for developing the research questions addressed in this new study. The association between seasonal fluctuations in WT and $N_2O$ emissions was observed in two different regions that were also both characterised by elevated groundwater sulfate. This is why we found it was acceptable to focus resources on spring and autumn periods at a limited number of sites, two of which were already known from the first study, rather than spreading resources across a full year, or across more sites.

  The reference to this previous study may have been too brief, and we will expand this a little to explain the context.

- In the end, these issues might make the conclusions of this study fairly descriptive with respect to the specific questions of the effects of season and land use on N2O emissions and subsurface dynamics. That doesn't mean that these specific results aren't valuable, just that the nature and limitations of the conclusions that can be drawn from the study on these specific questions need to be made very clear in the manuscript. Rethinking what the central finding or findings of the manuscript are may be helpful.

Response 19: We acknowledge the many limitations of a field study such as the one presented here. The ambition to conduct measurements in agroecosystems with management as closely as possible to the practical situation involved various compromises and limitations. We do not wish to overemphasize the results presented, and we stand by the statistical analyses conducted, but we will stress the limited data material, and the differences in soil conditions and management. The most interesting result, beside the evidence for different sources of $N_2O$ in spring and autumn, could be the more qualitative comparison with emission levels as currently estimated for inventories.

- The data on the changes in the water table and subsoil concentrations of N2O are great. But the conclusions drawn in section 4.3 could be more compellingly supported (there does not appear to be any quantitative analysis of the relationship) and discussed. Looking at the figures, in some cases N2O concentrations are enhanced above the water table, in other cases, below the water table, and an overall relationship is not immediately obvious. The importance of the capillary fringe concentrations for surface emissions makes this discussion of particular interest, and worth spending some text to guide the reader through your interpretation.

Response 20: Thank you for the comment. We were deliberately cautious in describing these relationships, since there seems to be much to learn about the mechanisms of $N_2O$ production and reduction in this system – see also the discussion of pathways in section 4.5. A significant relationship between $N_2O$ concentrations in the gas probe closest to, but above the WT depth was observed and is reported (Figure 7). However, the distance between probe depth and WT depth would have varied, and in periods with rainfall the WT depth could have fluctuated before a given sampling. For these reasons, we find it is difficult to take the data analysis or discussion much further at this stage. Additional studies could be contemplated, including measurements at higher resolution, and molecular analyses of populations and activities.

- The manuscript often reports whether there was an effect of fertilization (e.g., line 380, lines 404-405, 412-413, and others) or appears to test fertilizer vs site effects (e.g., line 405), and the methods detail how generalized linear mixed models were used to analyze the temporal

dynamics of N2O. However, I don't see any reporting of the statistical results of this model or its application to the impact of fertilization on N2O emissions or soil concentrations.

Response 21: The statistical results for the cumulative (integrated) emissions presented in Table 2 were obtained by calculating specially designed contrasts (linear combinations) of the parameters of the referred model (calculated in such a way that the contrast coincides with the trapezoidal approximation of the integral of the emission over time).Therefore, the generalized linear mixed models referred were indeed used in the text.

We will revise this Table to provide additional documentation for these summary results. Specifically, the results for the comparison of fertiliser effect in the autumn will be included. Additional details about the generalized linear mixed model results will be included in the online Supplementary Information.

- A follow-on point is the very nice finding that N2O at the capillary fringe generally controlled N2O emissions, rather than any variables in the topsoil. But this result raises the question of what controls variation in N2O concentration at the capillary fringe. This question seems to be of first order importance in this system, but is not addressed quantitatively in the manuscript. It might help tighten the manuscript if the discussion in section 4.3 is tied more explicitly to N2O concentrations in the capillary fringe.

Response 22: We are grateful for the acknowledgement that the relationship between $N_2O$ in the capillary fringe and $N_2O$ emissions is an interesting observation. We explained above the limitations of doing a more quantitative analysis and hope for understanding that this was an unexpected result, and that for this reason the sampling strategy was not optimal for addressing in detail the relationship. We will try to link the discussion in section 4.3 more to $N_2O$ in the capillary fringe, but this may still be a mostly qualitative discussion.

- A general comment: In presenting results, it could be helpful to start each section with a general description of the main results or patterns found instead of starting with detailed information for individual blocks; that detailed information can be presented later, to support the general patterns or describe deviations from those patterns. Anything you can do to guide readers through the results is great! In addition, the authors occasionally slip interpretation into the results section that would be more appropriate in the discussion section.

Response 23: Thanks for this comment. We will revise the Results section keeping this in mind.

- One of the main questions addressed in this manuscript is that of the importance of FeS2 oxidation for N2O production. Line 530-531 invokes a separate but presumably related manuscript that presents results showing that FeS2 oxidation is unimportant in this peat soil. It is difficult to say without knowing what the focus of that manuscript is, but my hunch is that it may make more sense to include the FeS2 results from this field study in the separate manuscript (presumably focused on mechanisms of N2O production in these soils), since we are effectively only getting half the story here. Something to consider, anyway. In the end, this manuscript doesn't provide much in the way of firm insights into the mechanisms of N2O production—that's not an inherent problem, it's just not something this study was designed to do--so one idea would be to cut out that part of this manuscript, and make the focus entirely on quantification of fluxes, the nice soil N2O & water table data (and capillary fringe finding), and environmental drivers more generally. It would be easy enough then to include a paragraph on mechanisms of N2O production that cites the other manuscript. Also, just to note, many journals require any related manuscripts that have been submitted elsewhere to be included as part of a manuscript submission, so would be a good idea to check the policy of the journal in question when you submit the separate manuscript.
  Response 24: As explained above, in the wider study we did intend to investigate soil pools of iron sulfides, and relationships with $N_2O$ in the field study, but also recognizing from the beginning that more controlled experiments would be needed for validation of any conclusions. The manipulation experiments constitute a full paper that is now in review. We also find that it is appropriate to report here a preliminary conclusion to the hypothesis presented in the previous BGS paper.

- The manuscript argues that the water table depth was related to surface N2O emissions (e.g., section 4.1), but that this relationship varied by site and by season, and speculates that soil properties modified the water table depth/N2O relationship. However, there doesn't seem to be any statistical/quantitative analysis to support a water table depth/N2O relationship or how other soil properties modify that relationship (and as noted above, it's not possible to statistically evaluate seasonal differences).

- Response 25: The graphical model results illustrated in Figure 7 do represent the results of a fairly advanced statistical analysis, and $p$ values are now included. This is the quantitative support we have for the discussion of effects of soil properties; seasonal effects were not tested. Please also see Response 48.

- section 4.1 is largely a re-statement of results; much of the actual discussion about the water table/N2O relationship is sprinkled throughout subsequent sections. A restructuring of the discussion might make the results easier to digest, with one section focused on discussing the capillary fringe result and one focused on understanding the water table/N2O relationship. Some discussion about why the patterns are so variable could be valuable, including some explanation of why water table increases stimulate N2O production at all sites in the autumn (and contrast with the results of other studies, e.g. Maljanen et al. 2003, which saw no effect of rising water table on N2O emission). One possible straw man interpretation could be that in the early spring (or late spring in the case of AR2), N2O production is limited by NH4+ (and/or NO2-/NO3-) availability, which in turn is constrained by the availability of O2. The decline of the water table may release the O2 constraint. In the autumn, in contrast, it is possible that aerobic conditions limit N2O production, and a rising water table or precipitation leads to higher N2O emissions. If indeed the case, why a possible seasonal shift from substrate to O2 limitation happens would be interesting to understand.

  Response 26: Thanks for this comment. The present structure was in fact an attempt to highlight plausible mechanisms and interactions, which could potentially account for the observations with respect to soil $N_2O$ concentrations and emissions. The interpretation offered by the reviewer above was in fact also proposed in the manuscript (Line 541f). Other pieces of evidence, which can help understand the observed differences between land uses and seasons with respect to mineral N, soil $N_2O$ and $N_2O$ emissions may be found in the different parts of the discussion, but we acknowledge it this may have been at the expense of the larger picture.

  We would prefer to keep the close links between $N_2O$ emissions and supporting data, and the links to previous studies. However, we propose to include in section 4.1 a brief introduction to the interpretation that we wish to put forward, and what aspects will be discussed in subsequent sections.

Technical comments:

- Line 1: I would change the title to reflect the focus of the revised manuscript (in addition, as noted above, it seemed to me that neither seasonal nor land use differences were able to be rigorously tested, and so it would be better to exclude phrases like "effects of land use and season" from the title)

  Response 27: The hypothesis of the manuscript was to focus on wet seasons and management practices. In addition, this manuscript is a resubmitted manuscript, and we would like to keep the same title in order to make it possible for the readers to access to the final revised version later on. Therefore, we considered the reviewer's comment and will remove the last part of the title which will read: "Regulation of $N_2O$ emissions from acid organic soil drained for agriculture".

- Line 45: You can check my math, but it seems to me that the global warming potential is still uniformly larger for C than for N2O here. I don't think it is necessary to make the case that N2O fluxes are more important than carbon fluxes, just that the N2O fluxes are large.

  Response 28: The point here was that $N_2O$ emissions are more influenced by site management compared to soil C losses.

- Line 44: change ", which" to "that"

  Response 29: This will be done.

- Line 64: "The sites" : not sure what sites are being referred to. Could you add more context?

  Response 30: This will be rephrased to specify that these were the arable sites with extremely high N2O emissions in two regions investigated by Petersen et al. (2012):

  "The two arable sites showing extreme $N_2O$ emissions in the study of Petersen et al. (2012) had both developed from marine forelands …"

- Line 120: please add a short explanation for why soil gas data were not presented for unfertilized RG2 during Autumn

  Response 31: Some probes were damaged during handling and removal in connection with field operations at the various sites during spring, and replacements were not available. The measurements at *RG2*-NF were sacrificed; a note will be added to explain this.

- Line 167: was this the fertilized or unfertilized block?

  Response 32: Intact soil cores to 100 cm depth were obtained from both fertilised and unfertilised subplots, and the analyses represent both fertiliser treatments. See also reponse no. 37.

- Line 176: remove "quantitatively" (not sure what it is intended to mean)

  Response 33: We will remove "quantitatively".

- Line 210-211: Entirely your choice, but perhaps AVS and CRS don't need to be abbreviated

  Response 34: We prefer to keep them abbreviated.

- Line 197: specify type of filter paper used

  Response 35: 1.6 µm glass microfibers filter, 691 was used. That information was added to the text.

- Line 237: Is the instrument ever checked against a set of standards of varying concentrations?

  Response 36: Calibration standards (0 to 2000 ppb) were included before and after each sequence run and used for determination of sample concentrations. Also, extra calibration samples were included after every 10 unknown samples to verify signal stability.  We believe these are standard procedures and prefer not to spell this out.

- Line 339: Indicate whether fertilized or unfertilized blocks were sampled

  Response 37: Will be rephrased to:

  "Nitrite-N concentrations were determined in soil profiles from the cores sampled at ites *RG1* and *AR1* on 23 April and 2 September 2015. Both fertilised and unfertilised subplots were represented, although at site *AR1* no fertilisation had taken place at the time of sampling in April. There was variation at depth in the soil, which could not be linked with fertilisation".

- Line 354 change "temporarily" to "temporary" Line 363: change "trends" to "concentrations"

  Response 38: These changes will be adopted.

- Line 370: Figure 3 seems to suggest that the N2O concentrations in the top 40cm of soil look to be 1-2 orders of magnitude higher in the fertilized RG1 than fertilized RG2. And unfertilized RG2 looks to be 1-2 orders of magnitude higher than unfertilized RG1 between 60 and 100cm depth for most of the spring. Yet they are described as "generally similar." I wouldn't have thought that would be considered "generally similar"—am I missing something?

  Response 39: Please note that the color scale for *RG2*+F was not correct, and thus concentrations were in general higher at site *RG2*. When referring to "generally similar", this should be seen in the context of concentrations of several hundred ppm N2O being observed especially at *AR* sites, but also around DOY150 in *RG2*+F.

- Line 375-6: Since there are apparently no soil gas data from unfertilized plots in RG2 during autumn, this statement is too strongly worded (even independent of questions of whether the effect of fertilization was tested).

  Response 40: Thank you for the comment. We will change the sentence to read: "During autumn, $N_2O$ concentrations in the soil profile at the *RG1* and *RG2* sites varied between 0 and 12 $\mu$L L-1, with a tendency for highest concentrations at 10-20 cm depth (Figure 5). At site *RG1*, where both fertilised and unfertilised subplots could be sampled, this was independent of fertilization".

- Line 376: The figures are out of order—I think you can swap Figures 4 and 5.

  Response 41: Figure 4 presents the results of *AR* fields in spring, and Figure 5 presents the results of *RG* fields in autumn. We believe the order is correct. Perhaps the confusion is related to the error in contour lines for site *RG1*-fertilised?

- Line 380: could you be specific about what soil conditions showed significant within-site heterogeneity? Also, use "substantial" instead of "significant' if this heterogeneity wasn't tested, and if it was, consider providing P values

Response 42: The wording of this sentence was unfortunate, as in fact heterogeneity was inferred from $N_2O$ concentrations. This will be revised. We will also change "significant" to "substantial".

- Line 384: this is really interpretation, and might be better placed in the discussion.
  Response 43: Although strictly speaking this is correct, we find that it is helpful to bring up the coincidence between N2O accumulation and WT depth in this place, and so we would prefer to keep this statement.

- Line 400: were any measurements made of N in harvested biomass? It could certainly help support the story that differences in uptake could alter N2O emissions from different plots.
  Response 44: The point we made here was that N uptake probably mediated against effects of fertilisation on $N_2O$ emissions. Unfortunately, we were not able to include manual cuts of the grass before harvest, in order to measure N in harvested biomass.

- Line 405: if specific soil variables cannot be identified as causing the differences, perhaps change "soil conditions" to "site differences"
  Response 45: We will change "soil conditions" to "site characteristics differences".

- Line 411-12: I'm not sure I see this pattern clearly: emissions are already high when the water table is at 80, and in the fertilized plots of AR1, emissions are 1/3 as large on DOY 259 than DOY 252, even though the water table is at its peak on DOY 259. There's also no apparent effect of the increase in water table starting on DOY 307, and emissions look quite elevated on DOY 246, which may be before the increase in water table began. A quantitative analysis would be helpful.
  Response 46: Thank you for pointing out the complexity of N2O-WT relationships in this period. We should in this place have stressed the importance of soil NO3 availability and referred to Table S2. Also, we will mention that rainfall may induce N2O emissions in unsaturated soil layers, when nitrate is present.

- Lines 473-4: could be more specific and change "a short period" to "1 to 2 weeks"
  Response 47: Will be done.

- Line 481 and following: Section 4.3 draws a number of conclusions that don't appear to be supported by any statistical analyses.

  Response 48: It is true that this discussion is based on observations and interpretation of interactions between potential drivers of $N_2O$ emissions. Please note that soil $N_2O$ concentrations are equivalent gas phase concentrations, and hence information about soil bulk density and air-filled porosity at the individual gas sampling position would be needed to calculate absolute amounts of $N_2O$. With measurements in only one or two blocks per sites, such a discussion is necessarily qualitative, but we believe it may still be useful as basis for development of new research questions or testable hypotheses.

- Line 516-517: I'm not sure I see this rapid increase in N2O around the water table depth in all the blocks in figure 6?

  Response 49: It is true that in some plots the accumulation of $N_2O$ in the soil is not very closely linked to WT depth, but there was a concurrent increase in $N_2O$ below, and in some cases above, the WT depth.  We will modify the sentence to clarify this.

- Lines 533-534: I think it might be better to say that "denitrification in topsoil was the main source. . . " since there is no explanation of how the N2O in the capillary fringe is produced.

  Response 50: We will change the wording to read:

  "Bacterial nitrification, denitrification, and nitrifier-denitrification are all potential pathways of $N_2O$ formation (Braker and Conrad, 2011), and the significant relationship with $NO_3^-$ at *AR* sites in the autumn (Figure 7) suggested that denitrification in the top soil was the main source in this period".

- Line 544: I'd change "drive" to "regulate".

  Response 51: The word "drive" will be changed to "regulate".

- Line 532 and following: this is interesting discussion, but if there are supplementary data that could support application of the ideas to this study (e.g., water filled pore space, acetylene reduction experiments, etc), it would strengthen it considerably. If the manuscript

in preparation on FeS2 oxidation includes any detailed examinations of these questions, it may be better to limit the speculation here.

Response 52: The reviewer has a valid point that supplementary data or experiments could probably eliminate some of the pathways discussed here. On the other hand, it is extremely difficult to characterise, let alone quantify, soil conditions at the micro- to mm-scale relevant to microorganisms in the undisturbed soil, and therefore firm conclusions may be difficult to reach. We did not attempt to quantify bulk soil properties in this study, or conduct manipulation experiments. The short summary of related literature was intended as inspiration to the planning of future studies to identify pathways.

- Line 544-558: looking at Tables S1 and S2, it seems that there is generally more NO3- or NH4+ in these soils on April 22 and/or May 13 than there is NO2- on April 23 (much more if these were the fertilized plots—I could not see any indication of whether the undisturbed core was from fertilized or unfertilized plots). If correct, that suggests that perhaps there is not an imbalance between ammonia oxidation and nitrite oxidation? Perhaps all nitrifier populations are temporarily saturated by the increase in available NH4+?

Response 53: The fact that nitrite accumulated in samples collected in April, but not in September, suggested that there was a difference in the balance between ammonia oxidation and nitrite reduction between the two sampling dates. Soil nitrogen pools are not necessarily uniform, and hence the accumulation of nitrite could be associated with microsites with only a fraction of the total $NH_4$ and $NO_3$ pools. If indeed detoxification is important in this system, it is conceivable that turnover rates for nitrite would also be high. See also Response no. 55.

- The discussion in Lines 486-489 also seems to suggest that denitrification was cranking along pretty well in the AR sites. And perhaps there's reason to be cautious about inferring processes from snapshots of concentrations, whether a single depth profile or weekly measurements of NH4+ and NO3- . Presumably, high concentrations could indicate anything from slow loss rates of each compound (whatever the pathway may be), or could reflect rapid N mineralization rates. If, by chance, total N concentrations were measured at each sampling date, calculations of net mineralization and net nitrification might be able to provide additional insight into whether and where reactive N might be accumulating.

- Response 54: Unfortunately, total soil N concentrations were just measured once (Table 1). Concern is raised about the limitations of point measurements as basis for the interpretation of N transformation processes. It may be argued that the soil gas probe measurements of $N_2O$ represent a more time-integrated measure of N transformations, insofar as the equilibration time is in the order of days (cf. Petersen, 2014; Diffusion probe for gas sampling in undisturbed soil. Eur. J. Soil Sci. 65: 663-671). High concentrations thus indicate a sustained production and not just transient episodes.

- Line 550-51: This is partly covered in the note immediately above. I see that NH4+ remains at high concentrations, but NO3- does as well, which is why I'm unsure about the suggestion that there is a lack of a mechanism to remove NO2-.

- Response 55: Peat soil is an extremely heterogeneous environment and may be dominated by dead-end pores (Hoag R.S., Price J.S., 1997. The effects of matrix diffusion on solute transport and retardation in undisturbed peat in laboratory columns. J Contam Hydrol 28:193-205). It is therefore difficult to infer microbial activities from bulk soil concentrations. The graphical model analysis showed relationships between top soil $NO_3^-$ concentration and $N_2O$ flux, hence there is some support for the quantitative importance of denitrification. We will include a short discussion on the possible constraints on solute and gas exchange.

- Figure 2 caption: indicate whether cores were taken from fertilized or unfertilized blocks. Response 56: As it was mentioned in section 2.4.2 'Soil sampling', soil samples were collected to 1 m depth within 1 m distance from the positions of flux measurements in Block 3 of sites *RG1* and *AR1*. As stated previously, a soil core was collected from both fertilised and unfertilised subplots, but at site *AR1* the soil was not fertilised until later in the spring.

- Figures 3 through 7: I think it might be easier to evaluate these data if the entire year of data are presented in a single plot, rather than separating spring and fall data—I don't think it would make it any more difficult to read the data. An axis break could be included between DOY 167 and DOY 246. You could also explore presenting surface N2O flux in a log scale— there may be a variation that would be visible on a log scale that is difficult to discern on the current linear scale because of the dates with very high fluxes.

Response 57: Thank you for these suggestions. We feel that each Figure already has a lot of information, and certainly the presentation of results per block would become more difficult, if both seasons were to be combined into one. In order to keep Figures readable, we would like to maintain the design we have developed.

The manuscript text switches freely between using DOY, month, and terms like "early spring" to describe time, which makes it challenging for the reader to compare the text and figures. Sometimes the DOY is included parenthetically when month names are used, which is great, but this practice should be extended throughout the text. Alternatively, the x axis labels could be changed to month names and days.

Response 58: In order to make time traceable and consistent throughout the manuscript, we will include DOY in parentheses when month names or season names are used. However, we prefer to keep DOY numbers in x axis.

- Tables S1 through S4 would be much easier to read in figure form (possibly in a single figure), though I appreciate the inclusion of the summary data here. Actually, why not explore adding these data as a second y axis in figures 3-6, sharing the panels used for N2O. Since topsoil nitrate is presented as a significant predictor of N2O flux, it could be valuable to be able to compare the data in the figures.

Response 59: We would like to keep that information as presented now, in tables of supplementary materials. There is already much information presented in each figure, and adding the mineral N information with standard errors at different depths to those figures would result in reduced readability.

---

## Author Response (AR1)

Dear editor,

we hereby submit a thoroughly revised version of our manuscript, now titled **"Regulation of N2O emissions from acid organic soil drained for agriculture"** by Arezoo Taghizadeh-Toosi et al., for further consideration.

We appreciate the constructive comments from both reviewers to the Discussion paper. The reviews acknowledged that novel results are presented, and that overall our conclusions are valid. However, it was also clear that the presentation of this complex story could be improved. The manuscript has therefore been thoroughly revised from the previous submission.

Both reviewers had many specific suggestions for clarification and improvement, which have been addressed. Our actions are described in detail below. Please note that all responses to reviewer comments below are rewritten and extended compared to the preliminary responses uploaded at the web site.

Reviewer #2 had some additional recommendations and concerns. One idea was to disregard aspects related to the role of pyrite oxidation, but since it was a main hypothesis when planning this study, we have kept this element. However, in the Discussion it is now addressed in the first part, before turning to alternative mechanisms. We believe this has given a more logical flow, from the original research questions towards actual findings.

The two neighbouring sites *AR1* and *RG1* were proposed to be part of a split-split plot design. However, these two sites have different cropping histories many years back, and we used the same mixed effects model to analyse results from these and other organic soils in a previous publication (Petersen et al., 2012). On the other hand, we agree that this study was not designed to document land use and season effects, but to study mechanisms under contrasting conditions, and therefore, as proposed by reviewer #2, "Effects of land use and season" was removed from the title. Please also note that in the graphical model analyses each land use and season were analysed separately.

A final recommendation was to restructure the Discussion, and this recommendation we have followed. As a result, much of the text is rewritten and reorganised. The number of subsections was reduced in an attempt to develop a more coherent discussion addressing first environmental controls, and secondly possible pathways.

We hope that these improvements to the presentation, and clarification of various aspects, has made the manuscript suitable for publication.

Kind regards, Arezoo Taghizadeh-Toosi on behalf of all co-authors Anonymous Referee #1 Received and published: 28 February 2019

This manuscript investigated N2O emissions and concentrations in peat soils under 2 agricultural crops: grassland and potato) at 2 distinct site locations during spring and autumn season of one year only. All combination (site x crop) treatment received different management in terms of fertilisation and harvesting etc. The N2O production measurements were characterized with static chambers and soil N2O diffusion probes placed at 5, 10, 20, 50 and 100 cm depths. All potential environmental factors (climatic or edaphic) were also monitored during this period. This manuscript has been resubmitted and is substantially improved and conclusions are now validated. It would appear that a lot of fieldwork and indeed field data have been processed and are not equally discussed here but is focused on the title of the manuscript. It is well written and sufficient information provided to allow their reproduction. Some minor comments below would help clarify some details of the experiment and the results.

Response 1: Thank you for the positive evaluation and helpful comments. We are pleased to learn that you find conclusions supported by the results presented. While this study covers only one year, it builds on a 14-month monitoring study in eight locations, with three land use categories (Petersen et al., 2012), which is why we found it appropriate to focus this study on specific research questions emerging from the previous study. We wanted to keep soil conditions as closely as possible to farming practices at the individual sites, which is why management differed in timing and types of fertilisers, but the effect of fertilisation was an experimental factor at all sites. Below we describe in detail how we have reacted to specific comments [note: line numbers refer to the track-changes copy of the revised manuscript].

Line 11: rephrase or change word 'extensively' or 'intensive'; there is nothing extensive about growing cereals or potatoes on organic soils given the cultivation/fertilisation inputs. Perhaps it was meant to be 'widely' used?

Response 2: We have changed the word 'extensively' to 'widely' to avoid ambiguity.

Line 24: emissions could be given per unit of time, either day or season. Where are those days in terms of season?

Response 3: The sentence was modified for clarity to the following:

"Spring and autumn monitoring periods together represented between 152 and 174 days at the four sites, and during this time the total  $N_2O$  emissions were 3-6 kg  $N_2O$ -N ha-1 at rotational grass, and 19-21 kg  $N_2O$ -N ha-1 at potato sites." (**1. 21**)

Line 79-83 belongs to methods; go straight to your hypothesis questioning the role of crop type and seasonal variation

Response 4: Thank you for this suggestion. The detailed information about measurements (Line 81-83 in the Discussion paper) was deleted, but we have kept the first sentence to establish the context.

**Line 130; the fertilisation treatment is different in each site and therefore do not act as replicate but different treatment.**

Response 5: Please note that fertilisation was only represented in the statistical analysis as a categorical variable. We chose to follow the actual management at each site, and therefore different fertiliser types, rates and timing of application were used, but all four sites received a significant input of fertiliser N in one of two subplots, which allowed us to test for the effect of N fertilisation on cumulative emissions of  $N_2O$ .

**L149 each field trip being a day sampling so 2 sites were sampled per day maximum but all were sampled during the same week? Rephrase please.**

**Response 6: We have rephrased the text as follows:**

"Field trips included sampling at two sites, either ARI + RGI or AR2 + RG2, and thus all four sites were visited during two field trips on consecutive days." (**l. 162**)

**L258 it is not clear that cumulative N2O emissions are here total or on a daily average.**

**Response 7: We have revised the wording as follows to clarify:**

"The model for daily  $N_2O$  emission described above was used to estimate cumulative emissions by integrating the flux curves over time. Treatment effects were then analysed by specially designed linear contrasts as described in detail by Duan et al. (2017), who showed that models with untransformed responses (when using adequate distributions) allow simple statistical inference of the time-integrated  $N_2O$  emissions." (**I. 279**)

**Line 301 : average deviation of soil temp from air temp is given; could it be better described in terms of sign L304.**

Response 8: We report average deviations, but also the largest positive and negative deviations observed. No changes made.

**It states in Lin 166 that soil samples were taken at the start of each season April and Sept?**

Response 9: The soil characteristics reported in Table 1 were based on analyses of soil cores sampled in April except for AVS and CRS, which were analysed on soil cores sampled in September. This was stated in the Table caption, but now it has been specified also in the text (l. 231). Please note that the description of total reactive Fe analyses (l. 226) was moved up to further emphasize this.

Depth of the total peat layer should be shown in Table 1 as it seems that RG2 is very shallow peat (<25cm). Also von post figure should be given for each peat layer.

Response 10: Unfortunately peat depth was not mapped (but general information is given in **l. 106-108**). It is true, as stated, that the degradation at site *RG2* was extensive, but still met the definition of organic soil. The following sentence was added:

"According to Kandel et al. (2018), the peat at 0-25 cm depth in arable soil in this area has a high degree of humification at H8 on the Von Post scale." (**l. 109**)

**Line 350. The WT reported in Figures 5 & 6 is confusing; what are they if you are not showing your continuous measurements (which shows higher WTL?)**

Response 11: The WT reported in Figures 5 and 6 were the weekly values measured concurrently with flux measurements. By referring to the continuous measurements, we wanted to stress that soil conditions were highly dynamic, and that the nitrate reduction potential could have been influenced by this. For consistency across seasons, we have chosen to show the values obtained at the time of flux measurements in all contour plots.

Line 407: there in previously in this paragraph, it would help to add the DOY (as per line 410) or else include the month in your Figures.

Response 12: We added this information, so that the last sentence now reads:

"The highest emissions occurred, independent of fertilisation, in June following a WT rise to 35 cm depth on DOY154." (**l. 448**)

Line 415: this is the first time that the monitoring period is mention; this should be explicitly shown in Table 2 at least and therefore rather than total a per day average would be better to compare treatment.

Response 13: The monitoring periods in spring and autumn were defined in **l. 157-160**. We have included information of periods (DOY) in a separate column in Table 2. Also, the cumulative emissions and treatment effects are now shown for both fertiliser treatments also in the autumn.

Figure 3-6: the WT is visible in blue not in grey.

Response 14: The colour of WT lines was changed to bright green, which is visible in black and white reproduction.

Figure 7 : the statistical number on the graphics should be explained in the legend since it is not clear to which lines they apply (especially 7a).

Response 15: We have moved results of the statistical test to the legend; they are referred to by numbers 1-5 i brackets.

The authors evaluate subsoil concentrations and emissions of nitrous oxide (N2O) and related soil variables at sites in a raised bog in Denmark which had been identified as having high potential N2O emissions. They conduct measurements over spring and fall, 2015, in four sites (two of which are immediately adjacent to one another): two cropped with potato and two with grasses; each site was also split into fertilized and unfertilized treatments. Using graphical analysis, they determine that in most sites, the concentration of N2O at the capillary fringe was the driver of emissions, though in one season/site combination, N2O emissions were related to sub-surface soil temperature and nitrate concentration. The relationship between water table depth and N2O appeared to vary by site and by season, but included a declining water table depth triggering N2O emissions in spring, and rising water table depth triggering N2O emissions in autumn. Pyrite was largely excluded as making an important contribution to N2O production or emission, and a range of possible mechanisms of N2O production were discussed. Lower emissions from grassland plots were attributed to greater plant uptake of soil N. Annual fluxes were comparable to or higher than IPCC Tier I emission factors for drained organic soils.

This is a concise summary of our study, i.e., based on a previous multi-site study (Petersen et al., 2012) we investigated possible mechanisms of extremely high N2O emission rate during periods with fluctuating groundwater. In the end, our hypothesis regarding involvement of pyrite could not be confirmed, and alternative explanations were explored based on the supporting data obtained. Thank you for the many detailed comments, which we have tried to address below [note: line numbers refer to the track-changes copy of the revised manuscript].

General comments: This is my first reading of the manuscript. This is an impressive field study with a lot of interesting data, and the importance of N2O concentrations in the capillary fringe for N2O emissions is a nice result, as is the finding that pyrite and iron monosulfide are unimportant, and the confirmation of high emissions from these soils. However, I have a number of concerns regarding the experimental design, support for some of the conclusions presented, and organization of the manuscript. The manuscript and discussion in particular could benefit from substantial revision after some contemplation of what the core advances of the study are. There seem to be three main aspects of the study (in no particular order): 1. Understanding the mechanism of N2O production, with a particular examination of the potential roles of pyrite and iron monosulfide. 2. Understanding the variables related to surface N2O fluxes. 3. Understanding the relationship between water table and soil N2O concentrations (and N2O flux). These different aspects are part of a single system, and ideally the paper will weave these aspects together into a single coherent story of the patterns and mechanisms behind N2O dynamics and its drivers. As noted below, I think it may make sense to refocus this manuscript around results related to items 2 and 3, for three reasons: a) it may help to clarify this study, b) results apparently highly relevant to item 1 have been kept for a second manuscript, and c) this study wasn't designed to be a comprehensive investigation of the specific mechanism of N2O production in these soils. Based on the first referee's review, it sounds as though this may be a revised version of an earlier manuscript; if so, sorry to be bringing new

concerns at this stage of the publication process, and apologies for the long comments. Note: I don't have expertise in graphical analysis or in the specific technique used to measure subsurface N2O concentrations.

Response 16: We appreciate the acknowlegdement that interesting results came out of this study. You emphasize that the role of N2O accumulation above the WT for N2O emissions in spring, but also the lack of importance of pyrite and iron monosulfide, are both nice results. We take this as an indication that the study merits publication, and we hope that the extensive revision now makes this possible. It was proposed to omit findings related to pyrite as a possible driver for N2O emissions, which was a hypothesis presented in a previous paper published in Biogeosciences (Petersen et al., 2012). We did not find a proper way to construct a different rationale for the study and therefore present this work along the lines of the original research questions. However, the Discussion section has been completely restructured and partly rewritten in order to address the pyrite hypothesis first, before turning towards alternative mechanisms to explain the extremely high N2O emission rates observed at arable sites. Additional changes were also made in Abstract and Introduction to strengthen the focus of the paper.

**Specific comments:**

Experimental design:

• The experimental design is unusual, with elements of different types of partly nested designs. Essentially, the first arable potato site (AR1) and first rotational grass site (RG1) are part of what traditionally would be called a split-split-plot design, but RG2 and AR2 are in a split-plot design. What this means is that the role of site in the analysis—there are effectively three sites--varies in its relationship with the treatments. AR1 and RG2 are nested within site, but RG2 and AR2 are not. Additionally, the lack of nesting of AR2 and RG2 within site raises additional issues, since RG2 has such a big difference in organic C content, so treatment is confounded with site in the RG2/AR2 pairing (in addition to the different fertilizer types used). Although differences between the potato and grass treatments are discussed in the results, and the title suggests that these and seasonal differences are the focus of the manuscript, I don't see any methodological description of the analysis that could allow one to compare RG and AR treatments. I'm not strictly sure how it would be done, but I also don't have a problem with softening the wording of the conclusions that can be drawn about the differences between RG and AR treatments—it does look like there may well be treatment differences there, I'm just not sure that with this design that it's possible to establish that statistically. So I think it needs to be made clear that if a strict statistical comparison was not conducted, the conclusions drawn aren't statistically supported (and any related concluding statements should be softened). And if a statistical comparison was conducted, details on how the nesting within site of RG1 and AR1 but not AR2 and RG2 was handled would be helpful. There is also no replication of bog-the results can't technically be generalized beyond the Store Vildmose raised bog. This is not a problem, and the authors don't attempt to extrapolate beyond the Store Vildmose, but it is a limitation that should be explicitly acknowledged. None of these issues should affect the finding that the capillary fringe is often/typically the primary determinant of the magnitude of N2O flux.

Response 17: Although located in neighbouring fields, the sites *RG1* and *AR1* should be considered as independent units, in accordance with the statistical analysis of Petersen et al. (2012). The two sites had different cropping histories many years back, and different crops in the experimental year. Also soil N dynamics and N2O emissions were evidently very different for these two land use categories.

Please note that the statistical analysis did not address land use effects, except that cumulative emissions were compared, and here the differences were evident even without any statistical support. In order to reduce the focus on the factorial setup, the title was changed, as proposed (see reponse 27).

• I am always concerned about field studies the present just one year of data, as it does limit insights into the degree to which patterns observed provide generalizable insights. The manuscript title suggests that comparisons of seasons is one of the central findings of the study, but since only one year of data are included, it is impossible to rigorously compare seasonal differences, as there is no replication of season. There are a lot of varying results in this study, which only strengthen this issue. I don't think these issues affect the finding on the frequent importance of capillary fringe N2O concentrations for surface emissions, though.

Response 18: Please note that this study does not stand alone. It was preceded by a study (Petersen et al., 2012), where N2O emissions were monitored during 14 months in three regions (including the area investigated in the present study), and including three different land uses, which were arable farming, rotational grass and permanent grass. In total 8 site-years were thus available as background for developing the research questions addressed in this new study.

The previous study also showed strong seasonal dynamics, which is why we found it was acceptable to focus resources on spring and autumn periods, rather than spreading resources across a full year, or across more sites. The results of the previous study are now highlighted in the discussion (**l. 484ff**).

• In the end, these issues might make the conclusions of this study fairly descriptive with respect to the specific questions of the effects of season and land use on N2O emissions and subsurface dynamics. That doesn't mean that these specific results aren't valuable, just that the nature and limitations of the conclusions that can be drawn from the study on these specific questions need to be made very clear in the manuscript. Rethinking what the central finding or findings of the manuscript are may be helpful.

Response 19: We acknowledge the many limitations of a field study kept as closely as possible to the practice of each farmer, but as stated above, this study was developed from a larger, multi-site study with reproduction of land uses and covering more than a year. We agree that this is a predominantly descriptive study. Still, the graphical models did reveal a constrast in the regulation of N2O emissions, which appears to be a novel result.

• The data on the changes in the water table and subsoil concentrations of N2O are great. But the conclusions drawn in section 4.3 could be more compellingly supported (there does not appear to be any quantitative analysis of the relationship) and discussed. Looking at the figures, in some cases N2O concentrations are enhanced above the water table, in other cases, below the water table, and an overall relationship is not immediately obvious. The importance of the capillary fringe concentrations for surface emissions makes this discussion of particular interest, and worth spending some text to guide the reader through your interpretation.

Response 20: Please note that the identification of a significant relationship between N2O concentrations in the gas probe closest to, but above the WT depth and N2O emissions (Figure 7) was the result of a quantitative analysis. The distance between probe depth and WT depth would have varied between samplings, and in periods with rainfall the WT depth may have fluctuated prior to a given sampling. For these reasons it was not possible to analyse the results in greater detail at this time.

• The manuscript often reports whether there was an effect of fertilization (e.g., line 380, lines 404-405, 412-413, and others) or appears to test fertilizer vs site effects (e.g., line 405), and the methods detail how generalized linear mixed models were used to analyze the temporal dynamics of N2O. However, I don't see any reporting of the statistical results of this model or its application to the impact of fertilization on N2O emissions or soil concentrations.

•

Response 21: The statistical results for cumulative emissions are presented in Table 2; we have revised the Table to include DOY periods of monitoring, and cumulative emissions in the autumn are now shown for each fertiliser treatment separately (previously they were pooled because they showed nearly identical N2O emissions).

• A follow-on point is the very nice finding that N2O at the capillary fringe generally controlled N2O emissions, rather than any variables in the topsoil. But this result raises the question of what controls variation in N2O concentration at the capillary fringe. This question seems to be of first order importance in this system, but is not addressed quantitatively in the manuscript. It might help tighten the manuscript if the discussion in section 4.3 is tied more explicitly to N2O concentrations in the capillary fringe.

Response 22: In the new, restructured Discussion, the paragraph referred to is part of section 4.1. There is specific reference to peat decomposition (early spring, with low soil mineral N status) and precipitation (after fertilisation and in the autumn, i.e. situations with high mineral N status) as proposed mechanisms behind N2O accumulation. Actual production pathways are now discussed in section 4.2. Again, the graphical models do represent a quantitative data analysis.

• A general comment: In presenting results, it could be helpful to start each section with a general description of the main results or patterns found instead of starting with detailed information for individual blocks; that detailed information can be presented later, to

support the general patterns or describe deviations from those patterns. Anything you can do to guide readers through the results is great! In addition, the authors occasionally slip interpretation into the results section that would be more appropriate in the discussion section.

Response 23: Text of the Results section has been revised and most subsections amended with an introductory statement to highlight a main aspect of the results.

• One of the main questions addressed in this manuscript is that of the importance of FeS2 oxidation for N2O production. Line 530-531 invokes a separate but presumably related manuscript that presents results showing that FeS2 oxidation is unimportant in this peat soil. It is difficult to say without knowing what the focus of that manuscript is, but my hunch is that it may make more sense to include the FeS2 results from this field study in the separate manuscript (presumably focused on mechanisms of N2O production in these soils), since we are effectively only getting half the story here. Something to consider, anyway. In the end, this manuscript doesn't provide much in the way of firm insights into the mechanisms of N2O production-that's not an inherent problem, it's just not something this study was designed to do--so one idea would be to cut out that part of this manuscript, and make the focus entirely on quantification of fluxes, the nice soil N2O & water table data (and capillary fringe finding), and environmental drivers more generally. It would be easy enough then to include a paragraph on mechanisms of N2O production that cites the other manuscript. Also, just to note, many journals require any related manuscripts that have been submitted elsewhere to be included as part of a manuscript submission, so would be a good idea to check the policy of the journal in question when you submit the separate manuscript.

Response 24: The field study was planned to explore a hypothesis coming out of a previous study (Petersen et al., 2012), and therefore the present study was a direct follow-up. In the restructured Discussion section, the pyrite hypothesis are now addressed first, and rejecting this hypothesis then leads on to a discussion of alternative mechanisms to explain N2O emissions. The unexpected evidence for different sources of N2O in spring and autumn periods in turn leads on to a discussion of biotic and abiotic pathways.

We note in the comment above a statement – "Something to consider, anyway" – and take this to say that a decision not to remove this part of the study is also acceptable.

• The manuscript argues that the water table depth was related to surface N2O emissions (e.g., section 4.1), but that this relationship varied by site and by season, and speculates that soil properties modified the water table depth/N2O relationship. However, there doesn't seem to be any statistical/quantitative analysis to support a water table depth/N2O relationship or how other soil properties modify that relationship (and as noted above, it's not possible to statistically evaluate seasonal differences).

Response 25: The particular statement in the former section 4.1, about soil properties modifying the effect of WT on N2O emissions, was intended as a transition to the next subsections, in which interactions between e.g. soil N status and WT dynamics were

discussed. The graphical model results (Figure 7) did show increasing N2O emissions with declining, as well as increasing WT depth that depended on soil N status, and various other studies were highlighted showing similar results.

The statement referred to above was deleted as part of the rewriting of the Discussion section, for which a more linear story line was developed.

section 4.1 is largely a re-statement of results; much of the actual discussion about the water ٠ table/N2O relationship is sprinkled throughout subsequent sections. A restructuring of the discussion might make the results easier to digest, with one section focused on discussing the capillary fringe result and one focused on understanding the water table/N2O relationship. Some discussion about why the patterns are so variable could be valuable, including some explanation of why water table increases stimulate N2O production at all sites in the autumn (and contrast with the results of other studies, e.g. Maljanen et al. 2003, which saw no effect of rising water table on N2O emission). One possible straw man interpretation could be that in the early spring (or late spring in the case of AR2), N2O production is limited by NH4+ (and/or NO2-/NO3-) availability, which in turn is constrained by the availability of O2. The decline of the water table may release the O2 constraint. In the autumn, in contrast, it is possible that aerobic conditions limit N2O production, and a rising water table or precipitation leads to higher N2O emissions. If indeed the case, why a possible seasonal shift from substrate to O2 limitation happens would be interesting to understand.

Response 26: We appreciate the suggestion to restructure the Discussion and have in fact completely reworked this section. An initial discussion of the hypothesis related to pyrite is now followed by two main subsections follow, which address environmental controls and possible pathways, respectively

The interpretation offered by the reviewer above ("N2O production is limited by NH4+ (and/or NO2-/NO3-) availability") was in fact also proposed in the manuscript (Line 541f in the BGD paper). The study of Maljanen et al. (2003) referred to described a rather different situation, where WT rise never exceeded 50 cm soil depth, and the crop was barley. There would have been less potential for N mineralisation and nitrate accumulation from barley after harvest compared to harvest of a potato crop with intensive tillage, and less potential for interaction between WT and soil mineral N in the top soil. Due to these differences we have not discussed this previous study.

Technical comments:

• Line 1: I would change the title to reflect the focus of the revised manuscript (in addition, as noted above, it seemed to me that neither seasonal nor land use differences were able to be rigorously tested, and so it would be better to exclude phrases like "effects of land use and season" from the title)

Response 27: Thank you for this suggestion – we agree and have changed the title accordingly.

• Line 45: You can check my math, but it seems to me that the global warming potential is still uniformly larger for C than for N2O here. I don't think it is necessary to make the case that N2O fluxes are more important than carbon fluxes, just that the N2O fluxes are large.

Response 28: Please note that we did not compare the importance of C and N2O fluxes, instead we wanted to highlight that N2O emissions appear to be more influenced by site management compared to soil C losses. The sentence has been changed to:

"Thus, while CO2 emissions are overall more important, site conditions appear to be more critical for N2O." (1. 54)

• Line 44: change ", which" to "that"

Response 29: Has been changed to "and this"

• Line 64: "The sites" : not sure what sites are being referred to. Could you add more context?

Response 30: This has been rephrased to specify that these were the arable sites with extremely high N2O emissions in two regions investigated by Petersen et al. (2012):

"The two regions showing extreme N2O emissions from arable soil had both developed from marine forelands ..." (**l. 74**)

• Line 120: please add a short explanation for why soil gas data were not presented for unfertilized RG2 during Autumn

Response 31: Some probes were damaged during handling (and one visit by heifers) during spring, and replacements were not available. The measurements at *RG2*-NF with unintended slurry application in July were sacrificed. The following was added:

"Due to damage of some probes it was decided to discontinue soil gas sampling in the unfertilised subplot at site RG2, which had by mistake received slurry on DOY 183" (**l. 197**)

• Line 167: was this the fertilized or unfertilized block?

Response 32: Intact soil cores to 100 cm depth were obtained from both fertilised and unfertilised subplots, and the analyses represent both fertiliser treatments. See also reponse no. 37.

• Line 176: remove "quantitatively" (not sure what it is intended to mean)

**Response 33: Done.**

• Line 210-211: Entirely your choice, but perhaps AVS and CRS don't need to be abbreviated

Response 34: We prefer to use abbreviations.

• Line 197: specify type of filter paper used

Response 35: 1.6  $\mu$ m glass microfibre filters was used. The information was added to the text (**l. 213**).

• Line 237: Is the instrument ever checked against a set of standards of varying concentrations?

Response 36: Calibration standards (0 to 2000 ppb) were included before and after each sequence run and used for determination of sample concentrations. Also, extra calibration samples were included after every 10 unknown samples to verify signal stability. We believe these are standard procedures and we have chosen not to spell out the details of analytical runs.

• Line 339: Indicate whether fertilized or unfertilized blocks were sampled

Response 37: Has been rephrased to:

"Nitrite-N concentrations were determined in undisturbed soil collected profiles from the cores sampled at sites RG1 and AR1 on 23 April (DOY 113) and 2 September (DOY 245) 2015. Both fertilised and unfertilised subplots were represented, although at site AR1 the fertilisation had not yet taken place at the time of sampling in April. There was variation at depth in the soil, which could not be linked with fertilisation." (**I. 367**)

• Line 354 change "temporarily" to "temporary" Line 363: change "trends" to "concentrations"

**Response 38: Done.**

• Line 370: Figure 3 seems to suggest that the N2O concentrations in the top 40cm of soil look to be 1-2 orders of magnitude higher in the fertilized RG1 than fertilized RG2. And unfertilized RG2 looks to be 1-2 orders of magnitude higher than unfertilized RG1 between 60 and 100cm depth for most of the spring. Yet they are described as "generally similar." I wouldn't have thought that would be considered "generally similar"—am I missing something?

Response 39: The color scale for RG2+F was not correct, and concentrations were in general higher at site RG2. When referring to "generally similar", this should be seen in the context of concentrations of several hundred ppm N2O being observed especially at AR sites, but also around DOY150 in RG2+F.

• Line 375-6: Since there are apparently no soil gas data from unfertilized plots in RG2 during autumn, this statement is too strongly worded (even independent of questions of whether the effect of fertilization was tested).

Response 40: Thank you for pointing this out. We have changed the sentence to:

"During autumn, N2O concentrations in the soil profile at the *RG1* and *RG2* sites varied between 0 and 12  $\mu$ L L-1, with a tendency for higher concentrations at 10-20 cm depth (Figure 5). At site *RG1*, where both fertilised and unfertilised subplots could be samples, this was independent of fertilization." (**1. 424**)

• Line 376: The figures are out of order—I think you can swap Figures 4 and 5.

Response 41: We have moved the text relating to *RG* sites in autumn down, and the Figure numbering is now correct.

• Line 380: could you be specific about what soil conditions showed significant within-site heterogeneity? Also, use "substantial" instead of "significant' if this heterogeneity wasn't tested, and if it was, consider providing P values

Response 42: In fact, heterogeneity was inferred from N2O concentrations. Has been rephrased to:

"The soil N2O concentrations suggested that there was considerable within-site heterogeneity in soil conditions, as the highest concentrations were observed in the unfertilised subplot." (**l. 415**)

• Line 384: this is really interpretation, and might be better placed in the discussion.

Response 43: Has been deleted, addressed in the Discussion.

• Line 400: were any measurements made of N in harvested biomass? It could certainly help support the story that differences in uptake could alter N2O emissions from different plots.

Response 44: Unfortunately, we were not able to include manual cuts of the grass before harvest, in order to measure N in harvested biomass.

• Line 405: if specific soil variables cannot be identified as causing the differences, perhaps change "soil conditions" to "site differences"

Response 45: Has been changed to: "site differences other than fertilisation". (l. 447)

• Line 411-12: I'm not sure I see this pattern clearly: emissions are already high when the water table is at 80, and in the fertilized plots of AR1, emissions are 1/3 as large on DOY 259 than DOY 252, even though the water table is at its peak on DOY 259. There's also no apparent effect of the increase in water table starting on DOY 307, and emissions look quite

elevated on DOY 246, which may be before the increase in water table began. A quantitative analysis would be helpful.

Response 46: Thank you for pointing out that the rise in WT was not a strong predictor of N2O emissions in the beginning of September. Previously the precipitation data shown in Figures 5 and 6 started by DOY246, i.e., the first measurement day in the autumn campaign, and this unfortunately left out the information that 10 mm rain fell on DOY244 and 22 mm on DOY245. The two Figures were updated to include this information.

The rainfall on DOY244 and 245 was probably absorbed by peat in the top soil and therefore did not directly affect WT depth, but very likely gas transport near the soil surface. Hence, the high N2O emissions from sites *AR1* and *AR2* on DOY246 was a result of anoxic conditions in the presence of soil NO3. The text has been modified to clarify this:

"High fluxes were observed on the first sampling day of this monitoring period, DOY246, while WT depth was still at 40 to 80 cm depth. This followed 10 and 22 mm rainfall on the previous two days that was initially absorbed by the peat. Rainfall the following days then led to a rise in WT. The subsequent decline in N2O emissions at AR sites coincided with WT withdrawal." (**1. 453**)

and in the Discussion section:

"Despite 32 mm rainfall on DOY244 and 245, the WT depth was still at 40 to 80 cm and could not account for the very high N2O emissions observed on DOY246 (Figure 6). According to Kandel et al. (2018), the peat of arable soil in this area is highly degraded (H8 at 0-25 cm depth on the von Post scale), and well-degraded peat will release as little as 10% of its water to drainage (Rezanezhad et al., 2016). It is therefore likely that the rain was absorbed by peat above the WT and created conditions suitable for denitrification." (**1. 528**)

• Lines 473-4: could be more specific and change "a short period" to "1 to 2 weeks"

Response 47: Done.

• Line 481 and following: Section 4.3 draws a number of conclusions that don't appear to be supported by any statistical analyses.

Response 48: soil N2O concentrations are equivalent gas phase concentrations, and information about soil bulk density and air-filled porosity at the individual gas sampling positions would be necessary to analyse N2O results. With measurements in only one or two blocks per sites, the discussion is necessarily qualitative.

• Line 516-517: I'm not sure I see this rapid increase in N2O around the water table depth in all the blocks in figure 6?

Response 49: As discussed above, the effect of rainfall was also partly to wet upper soil layers and thereby create conditions supporting denitrification. The text has been modified to emphasize this.

• Lines 533-534: I think it might be better to say that "denitrification in topsoil was the main source. . . " since there is no explanation of how the N2O in the capillary fringe is produced.

Response 50: We have changed the wording to read:

"Bacterial nitrification, denitrification, and nitrifier-denitrification are all potential pathways of N2O formation (Braker and Conrad, 2011), and the). The significant relationship with NO3- at AR sites in the autumn (Figure 7) suggested that denitrification wasactivity in the main source in top soil controlled N2O emissions during this period." (**1. 624**)

• Line 544: I'd change "drive" to "regulate".

Response 51: The sentence was changed in the rewritten Discussion.

• Line 532 and following: this is interesting discussion, but if there are supplementary data that could support application of the ideas to this study (e.g., water filled pore space, acetylene reduction experiments, etc), it would strengthen it considerably. If the manuscript in preparation on FeS2 oxidation includes any detailed examinations of these questions, it may be better to limit the speculation here.

Response 52: It is true that more data or experiments could probably eliminate some of the pathways discussed here. The incubation experiments referred to, however, only look at the potential for iron sulfides to stimulate N2O emissions. And results could not confirm their involvement.

We do find that the differences in environmental controls identified by the quantitative analysis of the graphical models warrant a bit of speculation about possible pathways. They provide some support for the proposed role of ammonia oxidation coupled with either chemodenitrification or nitrifier-denitrification during spring, and of heterotrophic denitrification in the autumn, where nitrate was a strong predictor of N2O emissions in arable soil. The short summary of related literature showed some support for this interpretation, and can hopefully inspire future studies to confirm pathways.

• Line 544-558: looking at Tables S1 and S2, it seems that there is generally more NO3- or NH4+ in these soils on April 22 and/or May 13 than there is NO2- on April 23 (much more if these were the fertilized plots—I could not see any indication of whether the undisturbed core was from fertilized or unfertilized plots). If correct, that suggests that perhaps there is not an imbalance between ammonia oxidation and nitrite oxidation? Perhaps all nitrifier populations are temporarily saturated by the increase in available NH4+?

Response 53: Soil nitrogen pools are not necessarily uniform, and the fact that nitrite accumulated in samples collected in April, but not in September, suggested that there was a difference in the balance between ammonia oxidation and nitrite reduction between the two sampling dates. We have added the following sentence:

"Total concentrations of NH4+ and NO3- at 25-50 cm depth were similar or higher (Tables S1 and S2), but well-decomposed peat is dominated by dead-end pores (Hoag and Price, 1997), and it is likely that N mineralisation and ammonia oxidation to a large extent took place in such pores having a slow exchange of solutes with active pore volumes." (**1. 642**)

• The discussion in Lines 486-489 also seems to suggest that denitrification was cranking along pretty well in the AR sites. And perhaps there's reason to be cautious about inferring processes from snapshots of concentrations, whether a single depth profile or weekly measurements of NH4+ and NO3- . Presumably, high concentrations could indicate anything from slow loss rates of each compound (whatever the pathway may be), or could reflect rapid N mineralization rates. If, by chance, total N concentrations were measured at each sampling date, calculations of net mineralization and net nitrification might be able to provide additional insight into whether and where reactive N might be accumulating.

Response 54: We have in fact tried to be cautious about assigning N2O accumulation and emission to specific processes. Concern is raised about the limitations of point measurements as basis for the interpretation of N transformation processes. It may be argued that the soil gas probe measurements of N2O represent a more time-integrated measure of N transformations, insofar as the equilibration time is in the order of days (cf. Petersen, 2014). High concentrations thus indicate a sustained production and not just transient episodes.

• Line 550-51: This is partly covered in the note immediately above. I see that NH4+ remains at high concentrations, but NO3- does as well, which is why I'm unsure about the suggestion that there is a lack of a mechanism to remove NO2-.

Response 55: Peat soil is an extremely heterogeneous environment and may be dominated by dead-end pores (Hoag R.S., Price J.S., 1997). It is therefore difficult to infer microbial activities from bulk soil concentrations. As mentioned, a sentence has been added to stress this possible spatial heterogeneity (Response 53)

• Figure 2 caption: indicate whether cores were taken from fertilized or unfertilized blocks.

Response 56: It is mentioned in section 2.4.2 'Soil sampling' that soil samples were collected to 1 m depth within 1 m distance from the positions of flux measurements in Block 3 of sites RG1 and AR1. A soil core collected from both fertilised and unfertilised subplots went into the analyses, but at site AR1 the soil was not fertilised until 21 May.

• Figures 3 through 7: I think it might be easier to evaluate these data if the entire year of data are presented in a single plot, rather than separating spring and fall data—I don't think it would make it any more difficult to read the data. An axis break could be included between DOY 167 and DOY 246. You could also explore presenting surface N2O flux in a log scale— there may be a variation that would be visible on a log scale that is difficult to discern on the current linear scale because of the dates with very high fluxes.

Response 57: Thank you for these suggestions. We feel that each Figure already has a lot of information, and we wanted to present results from each land use type side by side. In order to not sacrifice the temporal resolution, we have not revised these Figures.

The manuscript text switches freely between using DOY, month, and terms like "early spring" to describe time, which makes it challenging for the reader to compare the text and figures. Sometimes the DOY is included parenthetically when month names are used, which is great, but this practice should be extended throughout the text. Alternatively, the x axis labels could be changed to month names and days.

Response 58: In order to make time traceable and consistent throughout the manuscript, we have included DOY in parentheses throughout the text, as well as in Figures, but often together with actual dates for reference.

• Tables S1 through S4 would be much easier to read in figure form (possibly in a single figure), though I appreciate the inclusion of the summary data here. Actually, why not explore adding these data as a second y axis in figures 3-6, sharing the panels used for N2O. Since topsoil nitrate is presented as a significant predictor of N2O flux, it could be valuable to be able to compare the data in the figures.

Response 59: It would be useful to show mineral N data along with N2O emissions etc. in Figures 3-6, but they are already very busy, and mineral N data included two N species, two treatments and two depths. We have therefore kept these data in tables as supplementary information.

- 20 weekly field campaigns. In late AprilNO32 and early September, intact cores were collected ammonium (NH4+) concentrations. At all sites, the soil was acidic with pH ranging from 4.7 to 5.4. Spring and autumn monitoring periods together represented between 152 and 174 d, with cumulative emissions of 3-6 kg N2O-N ha-1 m depth at adjacent grassland and sites with rotational grass and 19-21 kg N2O-N ha-1 at sites with a potato sites for analysis of soil properties, which included acid volatile sulfide (AVS) and chromium reducible sulfur (CRS) to quantify, respectively,
- 25 iron monosulfide (FeS) and FeS2, as well as total reactive iron (TRFe) and nitrite (NO2-). Soil organic matter composition and total reduction capacity was also determined. The soil pH varied between 4.7 and 5.4.crop. Equivalent soil gas phase concentrations of N2O ranged from around 10 μL L-1 at grassland sites to several hundred μL L-1 at potato sites, in accordance with lower soil mineral N concentrations at grassland sites. Total N2O emissions during 152-174 days were 3-6 kg N2O. N ha+ for rotational grass, and 19-21 kg N2O. N ha+ for potato sites. Statistical analyses
- 30 byStatistical analyses using graphical models showed that soil N2O concentration in the capillary fringe was the strongest predictor forof N2O emissions in spring; and, for grassland sites, also in the autumn. For potato sites in the autumn, nitrate (the analysis found that NO3-) availability in the top soil, together with temperature, were the main controls on N2O emissions. Pyrite oxidation coupled with NO3- reduction could not be dismissed as a source of N2O,

butChemical analyses of intact soil cores, collected to 1 m depth at adjacent grassland and potato sites in spring and
 autumn, showed that the total reduction capacity of the peat soil (assessed by cerium (IV) reduction) was much higher than explained that represented by the FeS2-concentration, and the concentrations of total reactive iron (TRFe) were much higher than pyrite concentrations. The potential for chemodenitrification being a source of N2O during WT drawdown in spring is discussed. In contrast, those of FeS2. Based on the statistical graphical models and the tentative estimates of reduction capacities, FeS2 oxidation was found unlikely to be important for N2O emissions-associated with rapid soil wetting and WT rise. Possible pathways of N2O production in autumn were consistent with biological denitrification. Soilspring and autumn periods, and the potential sources of N availability and seasonal WT changes were important controls of N2O emissions.-, are further discussed.

Key words: Drained peat, potentially acid sulfate soil, rotational grass, potato, nitrous oxide, reactive iron

**45 1 Introduction**

50

55

60

Worldwide, 25.5 million ha of organic soils have been drained for agricultural use, mainly as cropland (Tubiello et al., 2016), which and this accelerates decomposition of soil organic matter and net carbon (C) and nitrogen (N) mineralisation above the water table (WT) (Schothorst, 1977). Drained organic soils are significant net sources of greenhouse gas (GHG) emissions due to emissions of as carbon dioxide (CO2) and nitrous oxide (N2O) (Goldberg et al., 2010; Maljanen et al., 2003). A recent supplement to the 2006 IPCC Guidelines for National Greenhouse Gas Inventories on Wetlands (IPCC, 2014) proposed average annual emission factors of 4.3 and 8.2 kg N2O-N ha-1 yr-1 for temperate grassland on drained organic soil with low and high nutrient status, respectively, and an emission factor of 13 kg N2O-N ha-1 yr-1 for cropland. For soil C losses, the emission factors proposed for the three land use categories were between 5.3 and 7.9 Mg CO2-C ha-1 yr-1 (Hiraishi et al., 2014). This implies that Thus, while CO2 emissions are overall more important, site conditions are potentiallyappear to be more critical for N2O than for CO2 emissions.

Site conditions are defined by land use, management, inherent soil properties and climate (Mander et al., 2010; Leppelt et al., 2014). Both WT drawdown (Aerts and Ludwig, 1997) and WT rise (Goldberg et al., 2010) may enhance N2O emissions, but such effects depend on soil nitrogenN status (Martikainen et al., 1993; Aerts and Ludwig, 1997). Maljanen et al. (2003) found that WT, CO2 emissions and temperature at 5 cm depth<del>, together</del>, explained 55% of the observed variability in N2O 
[revised manuscript text omitted]

**3.5 Soil N2O concentration profiles**

395

375

380

The distribution and temporal dynamics of  $N_2O$  in the soil profiles showed important contrasts between grassland and arable sites. Equivalent gas phase concentrations of  $N_2O$ , as determined by in passive diffusion samplers, were determined concurrently with gas sampling, and results are presented as contour plots (Figures 3-6) based on; data compiled in Table S5. A logarithmic grey scale had to be used in order to show trends within both *RG* and *AR* treatments, as concentrations sometimes differed by). Concentrations in many cases varied by several orders of magnitude, this was also true between sites and sampling days, and between depths within individual profiles in many cases. Some, and therefore a logarithmic grey scale was used to show trends. The gaps occurin Figures 3-6 indicate periods, where diffusion probes could not be installed or were temporarily removed due to field operations.

400

Under the rotational grass at site RG1, soil N2O concentrations during spring were mostly between 0.1 and 3  $\mu$ L L-1 (Figure 3). A higher concentration (15  $\mu$ L L-1) was observed at 40-80 cm depth in the fertilised subplot around DOY139, but only in Block 3 of the field plot. At site RG2, the concentrations of N2O in the soil during spring were generally similar to those of RG1, although there were more values in the 1-10  $\mu$ L L-1 concentration range (Table S5).

405 However, on 3 June (DOY154) a significant increase in N2O concentration occurred in the fertilised part of the plot with a maximum of 560 µL L-1 at 100 cm depth (i.e., well below the WT). This occurred during a period with frequent rainfall and could have been caused by NO2- leaching from the top soil. Soil N2O concentrations in the unfertilised plot also increased around this time, but only to c. 15 µL L-1 and mainly near the soil surface.

During autumn, N2O concentrations in the soil profile at the *RG1* and *RG2* sites varied between 0 and 12  $\mu$ L L+ independent of fertilisation and with a tendency for highest concentrations at 10-20 cm depth (Figure 5).

The arable site *AR1*, with sampling positions located in a different field, but only 10-20 m from those of site *RG1*, showed very different soil N2O concentration dynamics during spring (Figure 4). There was a consistent accumulation of N2O at 50 and 100 cm depth where seasonal concentrations averaged 340 and 424  $\mu$ L L-1, respectively. In contrast, at 5, 10 and 20 cm depth the average N2O concentrations were 10-30  $\mu$ L L-1, and there was no clear response to

fertilisation on DOY141 in terms of soil N2O accumulation. There The soil N2O concentrations suggested that there was significant considerable within-site heterogeneity in soil conditions, and as the highest concentrations were observed in the unfertilised subplot. Between DOY75 and DOY100, the concentrations of N2O peaked at nearly 1500  $\mu$ L L-1 at 50 cm depth and were 2-3 fold higher than at 100 cm depth, indicating that N2O was produced in the capillary fringe as WT in this period was around 60 cm depth. At site *AR2*, the highest soil N2O concentrations during early spring were consistently observed at 20 cm depth, but then gradually declining to reach the background level of 0.3  $\mu$ L L-1 in mid-May (around DOY130). In the unfertilised field plot, the N2O concentration then increased again at 20 cm depth to reach 272  $\mu$ L L-1 following rainfall, and a WT rise to 35 cm depth. With fertilisation, soil N2O concentrations were even higher at 10 cm depth and reached nearly 400  $\mu$ L L-1 in mid-June.

During autumn, N2O concentrations in the soil profile at the *RG1* and *RG2* sites varied between 0 and 12  $\mu$ L L-1, with a tendency for higher concentrations at 10-20 cm depth (Figure 5). At site *RG1*, where both fertilised and unfertilised subplots could be sampled, this was independent of fertilisation.

September was characterised by heavy rainfall (114 mm in total), and at site *AR1* a substantial rise in the WT from 80 to 40 cm depth was observed (Figure 6). Soil N2O concentrations showed a dual pattern, with maxima at 10 and 100 cm depth through to DOY266 (end of September), and after this time soil N2O rapidly declined as the WT withdrew.
Nitrous oxide concentrations equivalent to several hundred μL L-1 were measured even at 5 cm depth during this period. During late autumn, the N2O concentration at 0-50 cm depth varied between 0 and 20 μL L-1, whereas at 100 cm depth it remained high at 100-850 μL L-1. At site *AR2*, the groundwater level was higher than at *AR1* and reached the soil surface by mid-September5 (DOY 260). Soil N2O accumulated in both fertilised and unfertilised subplots following saturation of the soil, again with the highest concentrations at 20 cm depth. A secondary increase was observed near the soil surface at the last sampling on DOY314 in November, in response to a period with rainfall and a rapid WT rise.

3.6 Nitrous oxide emissions

410

415

420

425

The weekly sampling campaigns during spring and autumn showed much higher N2O emissions at arable compared to grassland sites independent of season and fertiliser N application. At site *RG1*, N2O emissions during spring ranged

from 0 to 550 μg N2O m-2 h-1, with no effect of fertiliser amendment (Figure 3). Growth of the The grass swardin the
 fertilised subplot showed a strongclear response to fertilisation (not shown), and presumably there was a rapidthat
 indicated uptake of thefertiliser N-added. At site *RG2*, however, a peak in N2O emissions occurred on DOY154, and the flux was still elevated at the next two samplings. This high flux coincided with the elevated-accumulation of N2O in the soil profile N2O concentrations-described above.

At site *AR1*, the N2O fluxes were generally much higher than at the *RG1* site grassland sites during spring (Figure 4).
Fluxes during early spring reached 2000-6000 μg N2O m-2 h-1 and were higher than in late spring where, as for site *RG1*, no effect of N fertilisation was observed. Hence, the higher emissions were associated with soil conditions and notsite differences other than fertilisation. The potato field at site *AR2* showed a different pattern, with N2O fluxes remaining low during early spring, and for several weeks after fertilisation. The highest emissions occurred, independent of fertilisation, in June whenfollowing a WT rise to 35 cm depth was observed n DOY154.

In the autumn, N2O fluxes from site *RG1* were consistently low (Figure 5). The first sampling at site *RG2* was on DOY260DOY259 in mid-September, where a high flux of 3000 μg N2O m-2 h-1 was seen, which dropped to near zero within 1-2 weeks. Nitrous oxide emissions at site *AR1* were high during September at 4000-10,000 μg N2O m-2 h-1 independent of N fertilisationτ(Table S2), and subsequently declined to near zero (Figure 6). The highHigh fluxes coincided with a rise in were observed on the WT from 80 to 40 first sampling day of this monitoring period, DOY246, while WT depth was still at 40 to 80 cm depthτ. However, this followed 10 and 22 mm rainfall on the previous two days. Rainfall the following days then was accompanied by a rise in WT. The subsequent decline in fluxes N2O emissions at *AR* sites coincided with WT withdrawal. At site *AR2* the pattern in N2O emissions was similar, and again the dynamics of N2O fluxes aligned with WT dynamics.

Cumulative N2O emissions were calculated for the 99-105 days of monitoring in spring, and for the 47-69 d period in autumn (Table 2). At *RG* sites, the average N2O flux from fertilised grassland was significantly higher than from unfertilised grass (7.3 *vs.* 2.0 kg N2O ha-1) during spring. At *AR* sites with potato, there was no significant effect of N fertilisation, but the cumulative N2O emissions of 15-17 kg N2O ha-1 were much higher than from *RG* sites. In the autumn, the there were no residual effects of N fertiliser application in spring, and average cumulative emissions at the *RG* and *AR* sites were 2 and 15 kg N2O ha-1, respectively.

**465 3.7 Interrelationships between driving variables of N2O production**

Graphical models were used to study the dependence structure among selected soil variables and N2O fluxes.
Interestingly, at *RG* sites in both spring (Figure 7a) and autumn (Figure 7b), and at *AR* sites in spring (Figure 7c), the only variable with a direct link to N2O flux was soil N2O concentration in the capillary fringe (N2OWT), indicating that N2OWT carried information on the N2O flux that could not be explained by indirect correlations between the other variables. Moreover, the variable N2OWT separated N2O flux from the other variables in the graph which, according to the separation principle (an instance of the general theory of graphical models), indicates that information about this variable rendered all the other variables uninformative with respect to N2O flux. For example, in the analysis of *AR* sites in spring (Figure 7c), the variables N2O flux and Temp5 were not directly connected, and therefore any correlation

475

between Temp5 and N2O flux could be completely explained by other variables. The only exception to this pattern was AR sites in the autumn (Figure 7d), where instead two other variables showed a significant relationship with N2O flux; one variable was NitrateT, i.e., NO3-N concentration in the top soil, and the other variable was soil temperature at 30 cm depth. All other relationships were unrelated to N2O flux, or could be accounted for by other variables.

**4** Discussion**

WeThis study investigated seasonal dynamics of N2O emissions and soil conditions in a region in Northern Denmark that wasan area, which has been designated as a hotspot for N2O emissions in a meta-analysis of organic soils across Europe (Leppelt et al., 2014). Spring and autumn monitoring periods together covered between 152- and 174 d, and cumulative N2O emissions during these periods were in total 3-6 kg N2O-N ha-1 for rotational grass, and 19-21 kg N2O-N ha-1 for arable sites with a potato crop. These numbers indicate, representing <6 month periods, thus confirmed</li>
previous results (Petersen et al., 2012) that annual N2O emissions werein this area are comparable to (*RG*), or clearly above (*AR*), the IPCC emission factors for drained organic soil of 8 and 13 kg N2O-N ha-1 yr-1 for nutrient rich grassland and cropland, respectively (IPCC, 2014). Hence, the observations confirmed that organic soil drained for agriculture in this region constitutes a high risk for N2O emissions, but also showed that this risk depends on land use. The area has been characterised as potentially acid sulfate soil (Madsen and Jensen, 1988), and a previous study showed groundwater sulfate concentrations in excess of 100 mg L-1 (Petersen et al., 2012). We therefore hypothesised that NO5- reduction coupled with FeS2 oxidation could be a pathway of N2O formation in this acid organic soil,

Leppelt et al. (2014) concluded that high N2O emissions are associated with cropped land having a pH below 4.7, C:N ratios below 30-35, and WT depths of 0.2–0.9 m, and they found a significant positive relationship with annual precipitation. This sites investigated here largely fit this description, but the specific mechanisms behind high N2O emissions are not easily derived from average annual conditions. The present study was therefore planned to examine high emission periods at higher spatial and temporal resolution to elucidate environmental controls and possible pathways, such as FeS2 oxidation being a driver of N2O emissions.

500

505

495

**4.1 Nitrous oxide**Pyrite, measured as CRS, was quantified at selected depths (Table 1), and with bulk density of the peat varying between 0.15 and 0.3 g cm-3 (data not shown), the total amount of CRS at 0 to 50 cm depth would thus be 200-350 mmol FeS2 m-2. The N2O emissions observed during spring and autumn monitoring periods constituted up to 145 mmol N m-2 in total (site *ARI*), and it is thus theoretically possible that the process described by Eq. 2 contributed to emissions of N2O. However, the FeS2 concentration (0.7-2.4 mmol kg-1) represented a minor part of the total reduction capacity (>11,500 meq kg-1 at 27-30 cm depth). Also, the concentration of total reactive Fe was 25-120 times higher than that of FeS2 (though less in terms of reduction equivalents). Reducing agents other than FeS2 were therefore likely to be more important, a conclusion that was later supported by a laboratory study in which peat amended with FeS2 did not show enhanced N2O production (Taghizadeh-Toosi et al., submitted).

4.1 Environmental drivers of N2O emissions and water table dynamics

15

The regulation of N2O emissions was investigated using a statistical method represented by graphical models. It identified N2O concentration in the capillary fringe as the strongest predictor of N2O emissions from both grassland and 510 arable soil in spring, and from grassland soil in the autumn. The implication is that N transformations at depth in the soil, and not in the top soil, were the main source of N2O escaping to the atmosphere in these cases. In accordance with this, there was no immediate effect of N fertilisation on emissions of N2O independent of land use. Other studies also found a limited response to fertilisation (Maljanen et al., 2003; Regina et al., 2004), although Regina et al. (2004) later observed a peak in N2O emissions after rainfall. Goldberg et al. (2010) reported that N2O emissions from a 515 minerotrophic fen were produced at 30-50 cm depth, in accordance with the observations presented here, where the highest concentrations of N2O were mostly observed at 20 or 50 cm depth (Table S5).

Peat decomposing in the capillary fringe during WT drawdown could have been the source of N for N2O production. It is well established that N2O emissions from organic soil may be enhanced by drainage (Martikainen et al., 1993; Taft et al., 2017). The), and the response will appear within days, as shown by Aerts and Ludwig (1997) in an 520 incubation study with an oscillating WT. A stimulation of N2O emissions by WT drawdown was also observed by Goldberg et al. (2010) when simulating drought under field conditions, but in additionalthough a pulse of N2O also occurred after rewetting. In the present study, the response to WT drawdown was complex, i.e., at sites RG1 and AR1 there was a stimulation of N2O emissions as WT declined in early spring, while this was not evident at sites RG2 and AR2. During autumn there was generally no effect of WT drawdown on N2O emissions. In contrastaccordance with this, 525 rising WT and/or increasing soil wetness in late spring and in the autumn resulted in a consistent increase in N2O emissions at all sites. Hence, the relationship between WT depth and N2O emission showed seasonal patterns and sitespecific effects, which indicated that other soil properties modified the effect of WT on N2O emissionsearly autumn consistently enhanced N2O emissions at all sites in the present study. Despite 32 mm rainfall on DOY244 and 245, the WT depth was still at 40 to 80 cm and could not account for the very high N2O emissions observed on DOY246 (Figure

530 6). Well-degraded peat will release as little as 10% of its water to drainage (Rezanezhad et al., 2016). It is therefore likely that the rain was absorbed by peat above the WT and created conditions suitable for denitrification.

**4.2 Nutrient status and land use**

The repeated-increase in N2O emissions afterduring WT drawdowncycles reported by Aerts and Ludwig (1997) was observed only with eutrophic peat, whereas a mesotrophic peat showed no effect of WT treatmentdynamics on N2O 535 emissions, which were consistently low. A similar interaction between nutrient status and WT depth was observed in field studies comparing N2O emissions from minerotrophic and ombrotrophic boreal peatlands (Martikainen et al., 1993; Regina et al., 1996). Thus, nutrient status, and N availability in particular, was probably a driver for the higher N2O emissions at AR sites used for potatoes. The RG sites with rotational grass, in contrast, showed much lower N2O emissions despite similar soil conditions and N fertiliser input. In the present study, soil NH4+-N and NO3--N RG1 increased to 133 and 120 µg g-1 dry weight soil upon fertilisation, respectively, concentrations at site but largely returned to the background level of around 5 and 10  $\mu$ g g-1 dry weight soil, respectively, within a week (Table S1). In contrast, at site ARI there was significant accumulation of NH4+-N and NO3--N even before fertilisation on DOY141, and soil mineral N remained high for several weeks (Table S2). This accumulation of soil mineral N

540

around the time of potato crop establishment could have stimulated N2O emissions in the arable soil. Grasslands on

545 organic soil generally show lower emissions of N2O compared to arable organic soil (Eickenscheidt et al., 2015; Petersen et al., 2012), presumably because plants compete successfully with microorganisms for available N. Schothorst (1977) estimated peat decomposition indirectly from the N-content in herbage yield of grassland and concluded that the soil supplied 96 kg N ha-1 when the drainage depth was 25 cm, but 160 and 224 kg N ha-1 with the WT in-drainage ditchesepth at 70 and 80 cm-depth, respectively. In the present study, soil NH4+-N and NO3-N 550 concentrations at RG1 remained mostly below 5 µg g+1 dry weight soil except for a short period after fertilisation (Table S1). In contrast, at site ARI the NO3-N concentrations were mostly at 15-25 µg g+1 dry weight soil during spring, and it declined more slowly after fertilisation, where soil mineral N peaked at c. 500 and 200 µg g-1 dry weight at 0-25 and 25-50 cm depth, respectively (Table S2). This does indicate that the grass sward effectively took up N mineralised from soil organic matter above the WT. Hence, plant uptake of N mineralised from soil organic matter above the WT likely

555

Independent of land use there was no immediate effect of N fertiliser application on emissions of N2O. Other studies also found a limited response to fertilisation (Maljanen et al., 2003; Regina et al., 2004), although Regina et al. (2004) observed a peak in N2O emissions in late spring after rainfall.

4.3 Nitrogen dynamics and N2ONitrous oxide concentration in soil profiles

caused the much lower N2O emissions from rotational grass in this study.

560

Only pooled soil samples from 0.25 and 25.50 cm depth were available for characterisation of provided indirect information about soil mineral N dynamics-(Table S1-S4), but additional information can be derived from soil N2O concentration profiles (Goldberg et al., 2008). The soil gas diffusion probes used in this study were installed vertically and thus did not disturb soil stratification prior to monitoring. At RG sites, soil N2O concentrations were generally low and did not provide clear evidence for microbial N transformations. In contrast, at AR sites there was during spring an 565 accumulation of N2O in, which supports the soil; the highest concentrations at ARI-conclusion above that plant uptake was a main sink for the N released during peat decomposition. At site RG2 an accumulation of N2O was seen at 1 m depth in late May (Figure 3), which could have been caused by leaching of mineral N from the acidified cattle slurry following extensive rain. In contrast, at AR sites there was significant accumulation of N2O in the soil; at site AR1 the highest concentrations occurred at 50 to 100 cm depth, while at site AR2 the highest concentrations were at 20 cm 570 depth, in accordance with the higher groundwater table. This suggests that peat decomposition was a significant source

of mineral N, and that biotic or abiotic processes led to extensive N2O accumulation. At site RG2, accumulation of N2O at 1 m depth in late May suggested that mineral N from the acidified cattle slurry had leached from the top soil (Figure 3). Soil N2O concentration profiles thus These observations indicated that emissions of N2O at AR sites before fertilisation were due to an interaction betweenwas produced in the capillary fringe, consistent with peat decomposition 575 as a source of mineral N, and declining WT. Inpossibly also in the period followingsaturated zone (see next section). Following N fertilisation, the accumulation of N2O in the soil profile was mostly associated with precipitation and rising WT.

580

Precipitation was high during September 2015, and the rapid rise in WT toward the soil surface resulted in accumulation of N2O in the top soil at all sites. However, N2O concentrations peaked at around 10 µL L+ at RG sites, as opposed to several hundred µL L+ at AR sites. Soil NO3-N concentrations at 0-50 cm depth in early September (DOY145) were 5 µg g+ dry weight soil at site RG1 (Table S1), but 40-150 µg g+ dry weight soil at site AR1 (Table

S2), which could have supported denitrification activity. It is not clear if the source of NO3- was decomposing potato erop residues or accelerated peat decomposition following harvest, or both.

4.4 Environmental controls

The previous sections have indicated that effects of land use and climate on N2O emissions (Leppelt et al., 2014; Mu et al., 2014) are modified by soil conditions that may vary across the year. We investigated possible drivers of N2O emissions using a statistical method represented by graphical models, which identified N2O concentration in the capillary fringe as the strongest predictor of N2O emissions from both grassland and arable soil in spring, and from grassland soil in the In the autumn. The implication is that N transformations at depth in the soil, and not in the top soil (despite fertilisation in some treatments) were the main source of N2O emissions from a minerotrophic fen were produced at 30–50 cm depth. Peat decomposing in the capillary fringe during WT drawdown could thus have been a driver of N2O production, and indeed the highest concentrations of N2O were mostly observed at 20 or 50 cm depth (Table S5). On the other hand, at site *AR1* the soil N2O concentration was high even at 100 cm depth, indicating that N2O was also

At the arable sites, the regulation of N2O emissions in the autumn was different from that in spring, since the graphical model identified NO3- in the top soil, and soil temperature at 30 cm depth, as significant predictors of N2O emissions at arable sites (Figure 7), although it should be noted that). The accumulation of NO3- was much greater at site AR1 compared to AR2-, suggesting differences in N mineralisation potentials. It is not clear if the source of N was 600 decomposing potato crop residues or accelerated peat decomposition following soil disturbance at harvest, or both. Rainfall most likely triggered denitrification at the AR sites, by rapidly increasing WT depth and soil water-filled pore space, thereby impeding the oxygen supply to much of the soil profile (Barton et al., 2008). This interpretation is supported by  $N_2O$  concentrations increasing dramatically around the WT depth  $N_2O$  concentrations below, as well as above the WT depth depending on site and block, and in fertilised as well as unfertilised subplots (Figure 6). In an 605 annual study, conducted in other parts of the Store Vildmose bog, Kandel et al. -In an annual study, conducted in other parts of the Store Vildmose bog, Kandel et al. (2018) also measured high peak emissions of N2O from a potato cropping systemcrop, i.e., around 2000 µg N2O m-2 h-1 in October 2014 and 6000 µg N2O m-2 h-1 in June 2015, which coincided with NO3- accumulation- and rainfall. Precipitation was also high during September 2015, and the rapid rise in WT toward the soil surface resulted in accumulation of N2O in the top soil at all sites. However, N2O concentrations reached 610 only around 10 µL L-1 at RG sites, as opposed to several hundred µL L-1 at AR sites, confirming that soil mineral N availability was a limiting factor for N2O emissions.

**4.5 Possible pathways 2 Pathways of N2O formation**

We hypothesised that NO3- reduction coupled with FeS2 oxidation could be a pathway of N2O formation in this acid organic soil, and FeS2, measured as CRS, was quantified at selected depths (Table 1). Assuming a bulk density of peat
 615 in this area of 0.15 g cm-3 (Schäfer et al., 2012), the amount of CRS at 0.50 cm depth at site *AR1* would correspond to around 180 mmol FeS2 m-2, whereas the N2O emission observed during spring and autumn monitoring periods together

620

constituted 145 mmol N m2. It is thus possible that the process described by Eq. 2 contributed to N2O emissions; though probably not during spring, where N2O emissions were unrelated to soil NO2- dynamics. The total reduction capacity of the peat was much higher than that represented by FeS2, i.e., >11,500 meq kg+ at 27–30 cm depth. Also, the concentration of total reactive Fe was 25–90 times higher than that of CRS. Together this indicates that reducing agents other than FeS2 were more important. Subsequent incubation experiments with addition of FeS2 together with different electron acceptors also suggest that FeS2 oxidation is not a driver of N2O emissions in this peat soil (manuscript in preparation), and hence alternative pathways should be considered.

 Bacterial nitrification, denitrification, and nitrifier-denitrification are all potential pathways of N2O formation
 (Braker and Conrad, 2011), and the). The significant relationship with NO3- at *AR* sites in the autumn (Figure 7) suggested that denitrification wasactivity in the main source in top soil controlled N2O emissions during this period. This was different in early spring, where however, the emissions soil mineral N concentrations were more strongly related to N2O accumulatinglow and N2O accumulated near the WT depth. Here, ammonia oxidation activity may therefore have controlled N2O emissions either directly, or indirectly *via* production of NO2- or NO3-. Ammonia

630 oxidising bacteria (AOB) are scarce in acid peat despite the presence of nitrite oxidising bacteria (NOB) (Regina et al., 1996), and some studies indicate that ammonia oxidising archaea (AOA) predominate in both abundance and activity (Herrmann et al., 2012; Stopnišek et al., 2010). Stieglmeier et al. (2014) isolated an AOA from soil that emitted N2O at a rate corresponding to 0.09% of the NO2- produced independent of O2 availability, but it is not known if this organism is present in acid organic soil-, and at this time an indirect control of denitrification activity seems more plausible.

Stopnišek et al. (2010) found that AOA activity was not stimulated by an external source of NH4+ and concluded that the activity was associated with N released from decomposing soil organic matter. Thus, in early spring the The anaerobic conditions of saturated peat may have been a limiting factor for N mineralisation and in turntherefore ammonia oxidation activity during early spring, a constraint which could have beenwas alleviated as the WT declined and oxygen entered deeper soil layers.

640

645

650

635

Ammonia oxidation may drive N2O emissions indirectly *via* production of NO2- or NO3-. Nitrite had accumulated at 20-50 cm depth in late April at both *RG1* and *AR1* sites (Figure 2)<del>. This</del>), which was consistent with peat decomposition in connection with and ammonia oxidation following WT drawdown. Total concentrations of NH3+ and NO3- at 25-50 cm depth were significant (Tables S1 and S2), but also suggestswell-decomposed peat is dominated by dead-end pores (Hoag and Price, 1997), and it is likely that ammonia oxidation to a large extent took place in such pores having a slow exchange of solutes with active pore volumes. The accumulation of NO2- suggested there was an imbalance between ammonia oxidation and nitrite oxidation activity. Estop-Aragonés et al. (2012) found that oxicanoxic interfaces in peat soil were located above the WT depth, and hence the capillary fringe in this study may have been still partly anoxic. Oxygen affinity differs between nitrifiers, with AOA>AOB>NOB (Yin et al., 2018), and <del>thus</del> oxygen limitation could thus have caused the accumulation of NO2-. In acid soil, this would result in product inhibition by HNO2, if there were no mechanism to remove NO2-; this would be especially true for *AR* sites, where mineral N accumulation was three to four times higher compared to *RG* sites (Tables S3-S6). Nitrifier-denitrification is one mechanism by which ammonia oxidisers can avoid HNO2 accumulation, and this process leads to N2O formation (Braker and Conrad, 2011). Another potential sink for  $NO_2^{-}$  is chemodenitrification, an abiotic reaction in which  $NO_2^{-}$  reacts with Fe2+ to produce N2O (Jones et al., 2015):

655
$$4Fe^{2+} + 2NO_2^- + 5H_2O \to 4FeOOH + N_2O + 6H^+$$
(3)

where in Eq. 3 Fe(OH)3 is shown as anhydrous FeOOH. Some depletion of TRFe was indicated at 50 cm depth at site *AR1*, which coincided with a similar depletion in NO2- (Figure 2). Nitrifier-denitrification and chemodenitrification are both sinks for NO2-, and therefore both pathways wereare potential sources of N2O emissions induring early spring.

The observation that TRFe concentrations were much higher than those of AVS or CRS (Table 1) was unexpected, but makes it relevant to consider alternative reactions withinvolving iron oxides/hydroxides, which have a potential to produce N2O that involve iron oxides/hydroxides rather than FeS272. One such recently described pathway is Feanmox, a process whereby ammonia oxidation coupled with ferric iron reduction can produce NO2- below pH 6.5 (Yang et al., 2012):

$$Fe(OH)_3 + 10H^+ + NH_4^+ \rightarrow 6Fe^{2+} + 16H_2O + NO_2^-$$
 (4)

665 Nitrate can also be produced under these conditions (Yang et al., 2012; Guan et al., 2018):

$$8Fe(OH)_3 + 14H^+ + NH_4^+ \to 8Fe^{2+} + 21H_2O + NO_3^-$$
(5)

A shuttle of  $Fe^{2+}$  between Feanmox and chemodenitrification (Eq. 3 and Eq. 4) could have caused explain the accumulation of N2O under anoxic conditions in the saturated zone, where presumably the availability of NH4+ from peat mineralisation would be a limiting factor. The confirmation of pathways will require more detailed investigations that should also involve include molecular analyses targeting microbial communities in the soil profile.

**5** Conclusion**

670

6

As hypothesised, there was an effect of land use on N2O-Nitrous oxide emissions, which were clearly higher from arable sites with a potato crop compared to rotational grassland-grass. This was independent of fertilisation, and instead 675 N2O emissions could be associated with soil N mineralisation, rainfall patterns and temperature, as hypothesized. There were strong seasonal dynamics in N2O emissions that were associated with WT dynamics. In spring there was no direct responseConcentrations of pyrite were low compared to the input of fertiliser N, and instead N2O emissions mainly reflected the accumulation of N2O near the WT. At sites used for a potato crop, NO3- accumulated after harvest and was significantly related to N2O emissions. Pyrite was present at low concentrations, and hence some N2O emission from 680 NO3- reduction coupled with FeS2 oxidation could not be dismissed, at least in the autumn. However, the total reduction capacity of the peat-was much higher than that represented by FeS25, and reactive Fe was predominantly in forms other than pyrite, probably as oxyhydroxides. While the hypothesis, that N2O was produced by NO3 reduction coupled with FeS2 oxidation, could not be dismissed, it is likely that other processes were more important. There were strong seasonal dynamics in N2O emissions, and evidence that different pathways were involved. We propose that oxidation of N mineralised from decomposing peat was the main source of N2O during after WT drawdown in spring, where 685

ammonia oxidation together with was followed by chemodenitrification was a likely pathway to N2O formation. In the autumn, (or nitrifier-denitrification of), whereas in the autumn, where NO3 derived from residues or decomposing peat following WT riseaccumulated in arable soil after heavy rainfall was probably harvest, N2O emissions were associated with rising WT and heterotrophic denitrification as the main pathway. Mitigating N2O emissions from the acid organic soil investigated here is challenged by the apparent complexity of underlying processes. However, reducing surplusmineral N in the soil, for example accumulation by ensuring a vegetation cover throughoutoutside the yearmain 
[revised manuscript text omitted]
     | pН        | EC          | TOC                      | Total N                  | C:N   | TRFe                     | AVS                     | CRS                     |
|---------|-----------|-----------|-------------|--------------------------|--------------------------|-------|--------------------------|-------------------------|-------------------------|
|         | (cm)      |           |             | (g 100 g -1 ) | (g 100 g -1 ) | ratio | (mg Fe g -1 ) | (µg S g -1 ) | (µg S g -1 ) |
| RG1     |           |           |             |                          |                          |       |                          |                         |                         |
| Depth 1 | 2.5-7.5   | 5.1 (0.2) | 0.26 (0.10) | 37.4 (0.2)               | 1.75 (0.00)              | 21.3  | 3.63 (0.11)              | 2.51 (0.86)             | 155 (62)                |
| Depth 2 | 7.5-12.5  | 5.3 (0.1) | 0.15 (0.02) | 38.2 (0.2)               | 1.79 (0.01)              | 21.3  | 4.03 (0.44)              | NA                      | NA                      |
| Depth 3 | 17.5-22.5 | 5.3 (0.5) | 0.37 (0.18) | 39.7 (0.3)               | 1.80 (0.04)              | 22.1  | 4.14 (0.32)              | NA                      | NA                      |
| Depth 4 | 36-40     | 4.8 (0.1) | 0.55 (0.02) | 43.1 (2.7)               | 1.85 (0.03)              | 23.3  | 3.04 (0.26)              | 2.60 (0.87)             | 133 (64)                |
| Depth 5 | 47.5-52.5 | 5.1 (0.3) | 0.42 (0.13) | 31.0 (15.6)              | 1.47 (0.64)              | 21.1  | 2.50 (0.55)              | 4.86 (1.07)             | 24 (17)                 |
| Depth 6 | 93-98     | 5.4 (0.0) | 0.51 (0.06) | 0.6 (0.3)                | 0.01 (0.01)              | ND    | 0.14 (0.04)              | NA                      | NA                      |
| RG2     |           |           |             |                          |                          |       |                          |                         |                         |
| Depth 1 | 0-25      | 5.0       | NA          | 19.8 (3.4)               | 1.34 (0.13)              | 14.8  | 2.29 (0.56)              | NA                      | NA                      |
| Depth 2 | 25-50     | 5.1       | NA          | 8.9 (3.0)                | 0.63 (0.23)              | 14.2  | 4.48 (NA)                | 1.71 (0.00)             | 33 (7.3)                |
| AR1     |           |           |             |                          |                          |       |                          |                         |                         |
| Depth 1 | 2.5-7.5   | 5.0 (0.1) | 0.45 (0.04) | 35.9 (0.1)               | 1.81 (0.02)              | 19.9  | 4.57 (0.09)              | 1.74 (0.02)             | 141 (9)                 |
| Depth 2 | 7.5-12.5  | 5.2 (0.1) | 0.42 (0.06) | 34.2 (0.2)               | 1.76 (0.02)              | 19.4  | 4.66 (0.15)              | NA                      | NA                      |
| Depth 3 | 17.5-22.5 | 5.2 (0.1) | 0.34 (0.04) | 41.0 (2.2)               | 1.93 (0.11)              | 21.3  | 4.99 (0.43)              | NA                      | NA                      |
| Depth 4 | 36-40     | 4.7 (0.5) | 0.37 (0.05) | 41.1 (5.8)               | 1.84 (0.05)              | 22.4  | 3.23 (0.41)              | 2.17 (0.29)             | 49 (3)                  |
| Depth 5 | 47.5-52.5 | 4.7 (0.3) | 0.48 (0.08) | 5.9 (1.7)                | 0.37 (0.13)              | 16.3  | 1.19 (0.19)              | 1.98 (0.41)             | 137 (39)                |
| Depth 6 | 93-98     | 5.4 (0.2) | 0.91 (0.03) | 0.3 (0.1)                | 0.00 (0.00)              | ND    | 0.18 (0.02)              | NA                      | NA                      |
| AR2     |           |           |             |                          |                          |       |                          |                         |                         |
| Depth 1 | 0-25      | 5.1       | NA          | 33.4 (1.2)               | 1.45 (0.03)              | 23.1  | 4.11 (0.03)              | NA                      | NA                      |
| Depth 2 | 25-50     | 5.1       | NA          | 38.4 (0.2)               | 1.46 (0.02)              | 26.2  | 3.78 (0.14)              | 1.65 (0.02)             | 45 (8)                  |

ND - Not determined due to TOC and total N concentrations being at the limit of detection.

NA - Not analysed.

**Table 2.** Cumulative emissions of N2O (kg N2O ha-1) during the spring (99-105 days) and autumn (47-69 days) monitoring period. Estimation for each season was performed using the trapezoidal approximation of the integral of the emission curve. Numbers in parentheses indicate 95% confidence intervals, and significant differences, corrected for multiple testing by the single-step method, are indicated by asterisks. *RG*, rotational grass; *AR*, arable crop (potato); F, fertilised; NF, unfertilised.

|                     | DOY              | Cumulative N 2 O_                               | RG-NF         | RG-F | AR-NF     |
|---------------------|------------------|------------------------------------------------------------|---------------|------|-----------|
| Spring # |                  | kg ha-1                                                    |               |      |           |
| RG-NF               | 63-162    | 2.0_(1.5-2.5)                                              |               |      |           |
| RG-F                | 63-162    | 7.3_(4.9-9.6)                                              | ***\$         |      |           |
| AR-NF               | 63-167    | 17.1_(13.9-20.2)                                           | ***           | ***  |           |
| AR- $F$             | 63-167           | 15.0_(12.2-17.8)                                           | ***           | ***  | NS        |
|                     |                  |                                                            |               |      |           |
| Autumn              |                  |                                                            | <del>RG</del> |      |           |
| RG -NF       | 252-314          | 2. 20 (1.61 -2. 79 )                         |               |      |           |
| RG-F         | 252-314          | 1.9 (1.4-2.4)                                       | NS            |      |           |
| AR -NF       | - 246-314 | <del>14.813.6(11.6_(10.2</del> -
17.9 1 ) | ***           | _*** | -         |
| AR-F         | 246-314          | 15.3 (11.2-19.4)                                    | ***           | ***  | NS |

§ \*\*\*, *p* < 0.001; NS, not significant (*p* > 0.05)

**The monitoring periods (spring and autumn) were: DOY63-162 and DOY252-314 (*RG1*); DOY64-169 and DOY260-307 (*RG2*); DOY63-162 and DOY246-308 (*AR1*); DOY64-169 and DOY245-314 (*AR2*).**

**Figure captions**

**Figure 1.** A. Location of sites *AR1* and *RG1* (both at 57°13'59.7"N, 9°50'40.3E), *RG2* (57°13'55.9"N, 9°52'20.2E) and *AR2* (57°13'7.6"N, 9°46'26.9E). B. Experimental design at each of the four sites, with three blocks centered around piezometers ( $\bullet$ ) and two subplots, one of which received N fertiliser at the rate of the surrounding field. Six collars for gas flux measurements (S1-S6) were distributed as indicated, and sets of 5 diffusion probes for soil gas sampling were installed near collars in selected positions (see text).

**Figure 2.** Nitrite-N (a, c) and total reactive iron, TRFe (b, d), in undisturbed soil cores collected at sites *RG1* and *AR1* on 23 April (DOY113; white symbols) and 2 September (DOY245; grey symbols). Results shown are mean and standard error (n = 2). The dotted lines indicate WT level on the two sampling dates.

**Figure 3.** The top panel shows rainfall, air temperature and management (F – fertilisation) at sites *RG1* (left panels) and *RG2* (right panels) during spring7. 3 March (DOY63) to 16 June (DOY169). The middle section shows N2O fluxes (black circles; mean  $\pm$  standard error, *n* = 3) and contour plots of soil N2O concentrations in fertilised subplots, and the lower section the corresponding results for unfertilised subplots. A logarithmic grey scale was used in order to show trends within both *RG* and *AR* treatments, and between depths. Soil gas sampling positions are indicated in the contour plots; numbers shown are N2O concentrations ( $\mu$ L L-1). Grey Green lines show the WT depth (which varied slightly between blocks). B2 and B3 refer to block number of diffusion probe positions.

**Figure 4.** The top panel shows rainfall, air temperature and management (T – tillage; F – fertilisation) at sites *AR1* (left panels) and *AR2* (right panels) during spring7, 3 March (DOY63) to 16 June (DOY169). The middle section shows N2O fluxes (black circles; mean  $\pm$  standard error, *n* = 3) and contour plots of soil N2O concentrations in fertilised subplots, and the lower section the corresponding results for unfertilised subplots. A logarithmic grey scale was used in order to show trends within both *RG* and *AR* treatments, and between depths. Soil gas sampling positions are indicated in the contour plots; numbers shown are N2O concentrations ( $\mu$ L L-1). Gaps are indicated where soil gas sampling probes were installed late, or removed due to field operations. Grey-Green lines show the WT depth (which varied slightly between blocks). B2 and B3 refer to block number of diffusion probe positions.

**Figure 5.** The top panel shows rainfall, air temperature and management (H - harvest) at sites *RG1* (left panels) and *RG2* (right panels) during autumn7, 3 September (DOY245) to 10 November (DOY314). The middle section shows N2O fluxes (black circles; mean  $\pm$  standard error, n = 3) and contour plots of soil N2O concentrations in fertilised subplots, and the lower section the corresponding results for unfertilised subplots. A logarithmic grey scale was used in order to show trends within both *RG* and *AR* treatments, and between depths. Soil gas sampling positions are indicated in the contour plots; numbers shown are N2O concentrations ( $\mu$ L L-1); the probes were absent in the unfertilised subplot after harvest. Grey-Green lines show the WT depth (which varied slightly between blocks). B2 and B3 refer to block number of diffusion probe positions.

**Figure 6**. The top panel shows rainfall, air temperature and management (H - harvest) at sites *AR1* (left panels) and *AR2* (right panels) during autumn<del>., 3 September (DOY245) to 10 November (DOY314).</del> The middle section shows N2O

fluxes (black circles; mean  $\pm$  standard error, n = 3) and contour plots of soil N2O concentrations in fertilised subplots, and the lower section the corresponding results for unfertilised subplots. A logarithmic grey scale was used in order to show trends within both *RG* and *AR* treatments, and between depths. Soil gas sampling positions are indicated in the contour plots; numbers shown are N2O concentrations ( $\mu$ L L-1). Grey Green lines show the WT depth (which varied slightly between blocks). B2 and B3 refer to block number of diffusion probe positions.

**Figure 7.** Statistical results from Using graphical models, a statistical analysis was conducted for the four eombination of erops (*RG*, *AR*) and season (spring, autumn). a. *RG*, spring; b. *RG*, autumn; c. *AR*, spring; and d. *AR*, autumn. The edges ("lines") connecting vertices ("points") and edges ("lines") indicate significant relationships between explanatory variables and the response variable, i.e., (N2O flux7). Statistical results for effects on N2O flux are: [1] 2.32 (0.12-9.11, *p* = 0.011); [2] 0.74 (0.06-3.05, *p* = 0.034); [3] 0.78 (0.41-2.47, *p* = 0.0002); [4] 1.34 (0.78-4.08, *p* = 0.008); and [5] 2.45 (1.10-9.90, *p* = 0.0002). Key to variables: AmmoniumT – NH4+ at 0-25 cm depth; NitrateT – NO3- at 0-25 cm depth; N2O WT – equivalent soil gas phase concentration closest to, but above the water table depth; Temp5 – soil temperature at 5 cm depth; Temp30 – soil temperature at 30 cm depth.

---

## Editor Decision (ED1)

I raised several issues in my previous review, and thank the authors for their efforts at addressing them. Overall, I think the discussion is much improved. Unfortunately, I believe my concern regarding the experimental design needs to be further addressed. I do believe that revision is possible, and although it will likely require new statistical analyses, it shouldn't be overly difficult once the appropriate design is identified.

Below I list four general comments, of which the first two are most important, followed by responses to some of the authors' responses to my earlier comments, followed by specific comments. Apologies again for the rather verbose review.

GENERAL COMMENTS

1. One of my main concerns was regarding the robustness of the conclusions that can be drawn from this experimental design (Author response 17). The authors have removed the seasonality analysis, which I appreciate, but I believe that important issues remain. Some of the problem is in the language used to present and discuss results rather than in the underlying science, but science issues remain.

- ■ This remains a study using only one year of data, though I don't think this is in itself a problem, it is a limitation. The authors argue in their response 18 that a previous 14-month study (Petersen et al. 2012) should also be considered, and that study does provide context for the results presented here (though note that any results from Petersen et al 2012 that are important for understanding the rationale for this study or for understanding its results should be summarized in this manuscript). However, Petersen 2012 is not part of the analysis, and the current manuscript doesn't provide any insight into the effects of interannual variability (also it seems that this was perhaps an abnormally warm and wet year). That's not a problem--I think the authors have done a good job of removing generalizing statements about seasonality from the revised manuscript--but I do think that a statement noting that the analyses of spring and fall are based on data from a single year should be included in the conclusion as part of an acknowledgment of potential limitations of the study.
- ■ This remains a study that lacks traditional replication of treatments, and I think this issue requires substantial additional revision.
  - o The authors argue that RG1 and AR1 should be considered as independent units because they differ in land use history, and were treated as such in Petersen et al. 2012. I'm afraid I'm not convinced—the design is an illustration of the traditional definition of a split plot: two adjacent fields with different treatments (or, as in this case, treatment histories), but a shared geographic location, and thus conditions that have been formed by a history of shared state factors (sensu Hans Jenny). As such they are simply not independent sites. The map in figure 1a, which presents AR1/RG1 as a single site, illustrates the point rather well, and as I understand, sampling positions in the two fields were only 10-20m from one another. One of

the clearest issues here is the nearly identical temporal patterns of the water table depth—the kind of shared environmental variable that split plot analyses were created to accommodate. I think it's fine for the authors to describe patterns at each of the RG and AR locations, but I think the experimental design makes any formal comparison between land uses impossible. If the authors want to examine site differences, I think it would have to done using an n of 3, treating RG1/AR1 as a single site with higher within-site replication, but it would be a good idea to consult a statistician—there may be other options.  Because of this issue, I think the graphical analysis may also need to be redone—perhaps organized around sites rather than land uses—if I am correct in my understanding that independence is an underlying assumption (as I mentioned in my previous review, I am not an expert in graphical analysis).  In any event, I recommend consulting with a statistician before proceeding with additional revisions.

- o A related note: the very big site differences between RG2 and AR2 (e.g., in carbon stocks) highlight the rationale for using split plot approaches (such as the one the authors use for their fertilizer treatments): such approaches avoid treatment or land use history differences being confounded with site differences.
- o The authors also argue that differences between land use was evident without any statistical support, which is a fair argument.  I think it's fine to discuss possible effects of land use qualitatively, but important to include and clearly highlight the caveat that the study lacks independent replicates for the land use treatments.
- o I don't think these issues mean that the graphical model cannot be applied or provide novel results, but the model needs to be applied using the appropriate experimental design, and can't treat RG1 and AR1 as independent sites.
- o In Response 17, the authors note that land use was not analyzed statistically. However, the generalized linear mixed model includes land use ("crop") in the interaction term used as the fixed effect (line 248-249), which is inappropriate given the replication concerns regarding RG1 and AR1.

2. This leads me to the language presenting results.  The qualitative results need to be more clearly communicated as such—the way many results are presented here implies that there has been a quantitative statistical analysis conducted.  A very simple example is line 328: "Mineral N concentrations were greater at AR1 compared to AR2". In the scientific literature, such a statement implies a significant statistical difference in mineral N concentrations between the two sites.  If a statistical test was conducted, a P value should be included (in general, there should be much more inclusion of p values in the text).  If it wasn't, small changes to the language need to be made that make that clear, e.g., "Mineral N concentrations appeared to be greater at AR1 compared to AR2".  Another small example might be line 334-35: "There was variation at depth in the soil, which could not be explained by fertilization." The phrase "could not be explained" implies that a statistical test was attempted, but was not significant.  If, instead, this is qualitative interpretation of data, it should include more qualifying language, e.g., "There was variation at depth in the, which did not appear to be related to fertilization."

In fact, there are numerous instances of this kind of presentation of results throughout the manuscript; here are just a few examples (there are many more involving site comparisons, temporal comparisons within sites, fertilizer effects, etc., and all should be addressed):

L316: "There were only minor differences. . . between seasons" implies that seasonal differences were tested; can be changed to something like "If there were any differences, they were likely minor. . ."

L 326: "The residence time for mineral N in the soil solution was generally longer at AR compared to the RG sites" implies that site differences were tested; can be changed to "The residence time for mineral N in the soil solution generally appeared to be longer at AR compared to the RG sites"

L328: "Mineral N concentrations were greater at AR1 compared to AR2" implies that site differences were tested; can be changed to "It appeared that mineral N concentrations may have been greater at AR1 than AR2"

L 329: "Fertilisation increased NH4+-N and NO3- -N concentrations" implies that differences between fertilized and unfertilized split plots were tested. Can be changed to "It appeared as though NH4+-N and NO3- -N concentrations increased following fertilization"

One quick way around this issue would be to state in the methods section that all interpretations of results are qualitative unless accompanied by a p value, and then to include p values where statistical tests were conducted (in any event, I encourage the authors to include P values for all statements reflecting the results of a statistical test). But a clearer solution would be 1) to test differences statistically if they have not been and include p values, and 2) to change the language describing results to make it qualitative, using words and phrases such as "may," "appears to," "might," etc. to describe apparent patterns that cannot be tested or aren't important enough to test. I don't think it's necessarily a problem to present results qualitatively (though for any results that are important to the central hypotheses being tested, it's always good to include statistical analyses), but it needs to be very clear when differences were tested and when they were not.

3. I think there could be more detail in the discussion of figures 3-7, which represent a lot of the data presented in this manuscript. Briefly, the text often invokes a relationship between the water table and a) N2O concentrations in the soil or b) N2O fluxes at the surface, but the relationships described in the text are not obvious to me from examining the figures—the spatio-temporal relationship between the water table depth and N2O concentrations seems quite different in different sites and seasons. Additional text walking the reader through the authors' interpretation of these figures would be helpful. I made related specific comments in my last review, and detail some points below in my response to the authors' response #46, as well as in specific comments for lines 494, 501, and thereabouts.

4. It seems to me that the two conclusions that can be clearly reached from this study are the rejection of the hypothesis that that FeS$_2$ oxidation coupled with NO3$^-$ reduction was an important driver of N$_2$O emissions, and the identification of the capillary fringe as an important predictor of surface N$_2$O fluxes.  If the authors agree, the conclusion should be revised; as it currently stands, the second of these findings is not discussed at all in the conclusion, and the description of the first conclusion gives the reader the impression that the hypothesized mechanism is possibly and maybe even probably not trivial.

Minor comment: the response to reviewers letter was sloppy – a number (maybe all?) of the line numbers of revised text referred to in the response to reviewers are incorrect, there are typos in the quoted text revisions (e.g., response 50), and sometimes there's no indication of what revisions were made or where they can be found in the text (e.g., response 49).  I emphathize with these kinds of errors, especially after getting a fresh batch of comments from a new reviewer late in the publication process, but anything you can do to make the job of the reviewers easier is appreciated.

**Responses to specific author responses**

Response 20: I think the authors missed my point here, which was that the manuscript seems to only conduct a qualitative analysis of the relationship between changes in water table depth and changes in surface N2O flux, rather than including a statistical analysis of the relationship between the two variables that could provide quantitative insight into the importance of variation in water table depth for N2O surface fluxes.  I agree that the manuscript does provide a graphical analysis showing that capillary fringe N2O is the only significant predictor of N2O surface fluxes at several sites, but being statistically significant and being important are not necessarily the same thing, and the graphical analysis does not include any analysis of the relationship between water table depth per se and N2O.  I don't think that the authors need to do what I am suggesting here for the manuscript to be publishable, but it seems like a missed opportunity.

Response 21: Thank you for the response, though part of my concern was not addressed.  This concern is largely related to my general comment about the language used to describe results.  My concern here had been that there were multiple instances in the manuscript (I cited them in the original comment as including lines 380 and lines 411-412) where temporal variation in surface N2O flux at a site was attributed to fertilizer effects, or where fertilizer effects were excluded as a cause of variation in surface N2O flux.  However, the relationship between temporal variability in surface N2O fluxes and fertilizer applications appears not to have been tested statistically.

Response 25: The authors argue that "The graphical model results (Figure 7) did show increasing N2O emissions with declining, as well as increasing WT depth that depended on soil N status." I do not think that the graphical analysis included a water table depth variable?

A separate point: I also don't understand what is intended by "increasing WT depth that depended on soil N status" but if the point is that N2O_WT depends on soil N status, that would be an interesting result that deserves more discussion. If instead it's referring to AR-autumn, where the N2O flux is determined in part by soil N status, then ignore this second comment of mine.

Response 39: I'm glad this was caught. I'd encourage the authors to carefully review the script for producing all figures again if they haven't already.

Response 46: I wonder whether this explanation may also hold for DOY 252 and 259: it looks to me as though there are elevated N2O concentrations in surface soils, and no clear connection to the capillary fringe. Wouldn't that suggest a topsoil source for all of the highest surface fluxes at AR1 during Autumn—a finding supported by the graphical analysis? The text implies that the high WT is responsible for the high emissions for DOY 252 and 259 ("The subsequent decline in N2O emissions at AR sites coincided with WT withdrawal"), but I would think that anoxia in topsoil related to elevated precipitation during this period could be a more likely explanation. The variability in topsoil N2O concentrations looks fairly physically separated from the water table dynamics, and if capillary fringe is not a significant predictor for surface N2O fluxes at this site during autumn, why invoke a relationship between WT withdrawal and surface N2O emission declines?

I have further questions about the interpretation of the subsoil N2O concentration dynamics discussed in my comment on line 494 below.

Response 54: I think it's a good move to consider N2O as a time-integrated measure. My previous concern here was very much regarding the limitations of using point measurements of NH4+ and NO3- to infer N transformation rates. I think the framing of the revised discussion addresses my concerns.

Response 56: So the data presented in figure 2 are means (and standard errors) of intact cores containing either fertilized and unfertilized soils in RG1, but for AR1, it's a mean of intact cores containing only unfertilized soils? I don't understand the rationale for presenting the data this way (instead of, for example, presenting fertilized and unfertilized soils separately where appropriate). At the very least, the caption should clearly detail this odd fertilization treatment situation.

Response 57: It's your decision, but there isn't much fine-scale temporal variation in these figures; I'd think it's worth graphing up the full year and seeing what it looks like.

Response 59: Again, it's your decision, but would think it's worth including in SI as figures as well as data tables—it's just very difficult to read patterns in a data table. But thank you for including the raw data for researchers who may be interested in using it for their analyses.

**New specific comments:**

Line 23: perhaps define capillary fringe here: "…in the capillary fringe— [definition here]—was. . ."

Line 62: it would be helpful to readers if you could provide an explicit definition of capillary fringe here, given how important it is to the manuscript.

Line 105: what ridges? Probably need to provide more agronomic details

Line 124: Perhaps this should be "method of fertilizer application"? (as N fertilization was noted as an exception in the next sentence)

Line 127: I believe the erroneous NS fertilizer application in RG2 was made on the same date as the second slurry application; if correct I would clarify that fact by starting this sentence with "On the same date as the second slurry . .. " or "Immediately after the second slurry. . . " If the NS application was made on a different date, indicate the date.

Line 137: here it would be helpful to explain the choices of spring and fall as emerging from the patterns observed in Petersen 2012, and in the introduction briefly describe those patterns. It could be just 1-2 sentences in the introduction, and an introductory clause to the first sentence here.

Line 144: it would give a more complete picture if the dates of these exceptions were enumerated here.

Line 165: revise to "…at -20C until analysis, described in section 2.4.5."

Line 170-1: I'm afraid this sentence is hard to follow. It reads as though the diluted soil gas was first transferred to the exetainer, and then from the exetainer to the glass syringe. Perhaps also specify whether exetainers were filled with 10ml gas as they were in the chamber measurements.

Line 174: As a general rule I'd argue it's best to explicitly include all applicable methodological detail, rather than referring readers to an additional manuscript (within reason). In this case, my institution does not have a subscription to the European Journal of Soil Science, so my library had to obtain 24-hour access to Petersen 2014 so I could see the dilution calculations; I would include them in the manuscript.

Line 240-245: it would be helpful to specify if/how field temperature and pressure corrections were made to obtain surface flux estimates.

Line 297: It should be noted here that these characterizations were conducted after the fertilizer treatment of RG1

Line 380: Instead of "independent of fertilization" which suggests a formal analysis, how about "this variation appeared to be broadly similar between fertilized and unfertilized subplots"

Line 394 and following: this is one of many examples related to my general comment on presenting qualitative results, but the generalized linear mixed model appears to include only time, and not fertilizer, as a factor in the fixed effect, so statements regarding the effects of fertilization should be made qualitatively (e.g., rather than "with no effect of fertilizer amendment," use something like "and did not appear to vary in response to fertilization." Similarly, "no effect of N fertilisation was observed. Hence, the higher emissions were associated with site differences other than fertilization" could be changed to "fertilisation did not appear to influence N2O fluxes. Hence, the higher emissions were likely associated with site differences other than fertilization"). The paragraph starting with line 406 does a good job of qualitatively describing results.

Line 467: change to "In accordance with this effect of rewetting…" (in general, it's good to give "this" an object to make it clear what is being referred to)

Line 490: here and maybe elsewhere there are still instances where time of year are referred to without DOY (here parenthetical DOYs would be particularly helpful since its referring to figures using DOY on the x-axis). In addition, please specify that this accumulation is in the fertilized plots of RG2.

Line 491: change "significant" to "substantial" if the meaning is "a lot." Avoid using "significant" except when alluding to the result of a statistical test.

Line 492: if the contrast is that in RG2 there was only accumulation at certain times and depths but accumulation was everywhere and all the time in the AR sites, change "..accumulation of N2O in the soil" to "…accumulation of N2O across all soil depths throughout the spring" or something similar. Maybe include a reference to Figure 4.

*Line 494: as was the case with the previous version of the manuscript (see my original comment that elicited the authors' response 46 and my general comment 3), I'm not sure I see the pattern so clearly. The location of elevated N2O above 40cm in AR2 is attributed to the water table being higher than in AR1, where elevated N2O tends to be in deeper soils, and the authors argue that this points to a capillary fringe source for N2O. But in RG2 the water table is higher than it is in RG1 during the spring, but the N2O distributions are reversed: they are higher at depth in in RG2, where the water table is higher, and higher in shallower soils in RG1, where the water table is lower. The fertilized AR2 also has an inversion of N2O concentrations during

spring that seems entirely unrelated to water table depth.  In autumn, the N2O at depth declines as the water table depth increases, and the patterns in the two blocks in RG1 seem somewhat opposite of one another even though the water table is just about identical.  With so much apparent variation in how N2O concentrations vary with the water table, how do these results all relate to capillary fringe as the N2O source?  What am I missing? I suppose that this comment, in combination with my new response to the authors response 46, suggests an opportunity for the authors to revise and expand their presentation (and perhaps interpretation) of these results.  It would be a very satisfying read if the paper spent a little more time on the subtleties in the links the graphical analysis identification of the significant drivers of surface N2O fluxes to the spatio-temporal dynamics presented in figures 3-6 (these are most of the data being presented in the paper, so feel free to give them more attention in the text).

Line 501: I'm not sure why water table dynamics are invoked in the discussion of N2O emissions at AR1, if capillary fringe was not a significant predictor of surface N2O fluxes in the graphical analysis, and, as noted previously (and in my comment on the authors' response 46), surface fluxes were already elevated before the rise in WT depth? A simpler explanation consistent with the graphical analysis might be that topsoil was the source of surface N2O fluxes in AR1 during autumn, apparently independent of water table depth.  If this discussion is intended to be descriptive only of dynamics within the soil (and does not have any relation to surface N2O emissions), that needs to be made clear.

Line 528: It would be helpful to have a definition of "dead-end pores"

Line 567-559: These are all the kinds of statements that need to be made much more qualitatively in the absence of clear statistical support, and issues with experimental design and strength of statistical support for conclusions need to be highlighted in the conclusion.

Line 560: remove the comma between "hypothesis" and "that"

Line 561: my impression from the results is that NO3- reduction coupled with FeS2 reduction is likely to be a trivial mechanism of N2O production in these soils (a conclusion apparently supported by the related manuscript that has been submitted elsewhere)—this statement makes it sound as though it could be non-trivial.

---

## Author Response (AR2)

Dear editor,

We hereby submit a revision of the paper "Regulation of $N_2O$ emissions from acid organic soil drained for agriculture" (bg-2019-14) by Taghizadeh-Toosi et al. Major revisions were requested, and we are grateful to be granted the extension needed to complete this work.

Perhaps the most critical point raised was the assumptions about land use and site conditions that were necessary to test for these effects. In the revised statistical analyses, individual combinations of site and crop were instead tested separately to avoid such assumptions. Surprisingly, the effects identified in the previous analysis were more or less confirmed, indicating that the relationships were robust.

Review comments were overall constructive and helpful in clarifying many aspects. With the exception of some proposed reorganization of Figures 3-6 we believe that we have acted on all points raised, as specified below in the point by point response. In this light, we hope that it will now be possible to accept the manuscript for final publication.

Kind regards,
Arezoo Taghizadeh-Toosi

*Detailed response to reviewer comments (in blue):*

I raised several issues in my previous review, and thank the authors for their efforts at addressing them. Overall, I think the discussion is much improved. Unfortunately, I believe my concern regarding the experimental design needs to be further addressed. I do believe that revision is possible, and although it will likely require new statistical analyses, it shouldn't be overly difficult once the appropriate design is identified.

Below I list four general comments, of which the first two are most important, followed by responses to some of the authors' responses to my earlier comments, followed by specific comments. Apologies again for the rather verbose review.

*Response: We appreciate the reviewer's critical eye on the data analyses. We have made an effort to meet the different concerns, and in particular assumptions regarding experimental design have been revised, as explained below.*

GENERAL COMMENTS
1. One of my main concerns was regarding the robustness of the conclusions that can be drawn from this experimental design (Author response 17). The authors have removed the seasonality analysis, which I appreciate, but I believe that important issues remain. Some of the problem is in the language used to present and discuss results rather than in the underlying science, but science issues remain.

- This remains a study using only one year of data, though I don't think this is in itself a problem, it is a limitation. The authors argue in their response 18 that a previous 14-month study (Petersen et al. 2012) should also be considered, and that study does provide context for the results presented here (though note that any results from Petersen et al 2012 that are important for understanding the rationale for this study or for understanding its results should be summarized in this manuscript). However, Petersen 2012 is not part of the analysis, and the current manuscript doesn't provide any insight into the effects of interannual variability (also it seems that this was perhaps an abnormally warm and wet year). That's not a problem--I think the authors have done a good job of removing generalizing statements about seasonality from the revised manuscript--but I do think that a statement noting that the analyses of spring and fall are based on data from a single year should be included in the conclusion as part of an acknowledgment of potential limitations of the study.

*Response: There are five references to the study of Petersen et al. 2012 in the Introduction, and we have added additional detail to explain the context:*

"Among sites with arable crops in three regions, two sites had $N_2O$ emissions corresponding to, respectively, 38 and 61 kg N ha$^{-1}$ yr$^{-1}$; both of these sites showed distinct seasonal patterns with the highest emissions in spring and autumn periods, whereas emissions at the third site were lower (6.4 kg $N_2O$-N ha$^{-1}$ yr$^{-1}$) and much less variable. Notably, WT depth at the two sites with seasonal patterns of $N_2O$ emission fluctuated between 10-30 and 10-120 cm depth, whereas WT depth at the third site remained at 90-125 cm depth throughout the 14-month monitoring period." (l. 53 in cleaned manuscript version)

*Also, we now begin the conclusion by stressing that this is a one-year study:*

" In this one-year study, N2O emissions were…" (l. 600 in cleaned manuscript version)

- This remains a study that lacks traditional replication of treatments, and I think this issue requires substantial additional revision.

    o The authors argue that RG1 and AR1 should be considered as independent units because they differ in land use history, and were treated as such in Petersen et al. 2012. I'm afraid I'm not convinced—the design is an illustration of the traditional definition of a split plot: two adjacent fields with different treatments (or, as in this case, treatment histories), but a shared geographic location, and thus conditions that have been formed by a history of shared state factors (sensu Hans Jenny). As such they are simply not independent sites. The map in figure 1a, which presents AR1/RG1 as a single site, illustrates the point rather well, and as I understand, sampling positions in the two fields were only 10-20m from one another. One of the clearest issues here is the nearly identical temporal patterns of the water table depth—the kind of shared environmental variable that split plot analyses were created to accommodate. I think it's fine for the authors to describe patterns at each of the RG and AR locations, but I think the experimental design makes any formal comparison between land uses impossible. If the authors want to examine site differences, I think it would have to done using an n of 3, treating RG1/AR1 as a single site with higher within-site replication, but it would be a good idea to consult a statistician—there may be other options. Because of this issue, I think the graphical analysis may also need to be redone—perhaps organized around sites rather than land uses—if I am correct in my understanding that independence is an underlying assumption (as I mentioned in my previous review, I am not an expert in graphical analysis). In any event, I recommend consulting with a statistician before proceeding with additional revisions.

*Response: We acknowledge that site conditions and distribution of treatments represent factors that warrant great caution in data interpretation, and we have indeed acted to meet the concern raised here. Note that the effect on N2O emission of the fertilisation treatment was not the same at the four sites; therefore, one of the greatest advantages of using split plot designs (estimating a common effect of the fertilization treatment) vanishes. In the revised analyses we have instead used a single model describing the effect of fertilisation for each site. Therefore, we now report the N2O emissions separately per site (both for the temporal dynamics and the cumulative emissions). We believe that this approach documents the results of our experiment better. The conclusions reached previously still stand, with a few modifications which are described.*

    o A related note: the very big site differences between RG2 and AR2 (e.g., in carbon stocks) highlight the rationale for using split plot approaches (such as the one the authors use for their fertilizer treatments): such approaches avoid treatment or land use history differences being confounded with site differences.

*Response: See previous comment. In the revised Table 2, results are now presented for each combination of site, crop, fertiliser treatment and season (n=8).*

    o The authors also argue that differences between land use was evident without any statistical support, which is a fair argument. I think it's fine to discuss possible

effects of land use qualitatively, but important to include and clearly highlight the caveat that the study lacks independent replicates for the land use treatments.

*Response:  We have re-analysed cumulative N2O emissions for individual site-crop combinations by season (see revised Table 2). Results are now expressed as average daily rates, which allowed us to test for differences during spring and autumn, respectively. Furthermore, the temporal dynamics were analysed with a suitable mixed model. Results are now presented and discussed with reference to statistical support where possible, and otherwise wording has been changed to be more qualitative where relevant.*

o I don't think these issues mean that the graphical model cannot be applied or provide novel results, but the model needs to be applied using the appropriate experimental design, and can't treat RG1 and AR1 as independent sites.

*Response:  As for cumulative N2O emissions, graphical models were reanalysed for individual site-crop combinations by season, see revised Fig. 7 (now split into Figure 7 (spring) and 8 (autumn)). Notably, even without the statistical power of treatment replication, the effects remained largely unchanged with respect to treatment differences.*

o In Response 17, the authors note that land use was not analyzed statistically. However, the generalized linear mixed model includes land use ("crop") in the interaction term used as the fixed effect (line 248-249), which is inappropriate given the replication concerns regarding RG1 and AR1.

*Response:  In the revised analysis, land use treatments were analysed separately.*

2. This leads me to the language presenting results. The qualitative results need to be more clearly communicated as such—the way many results are presented here implies that there has been a quantitative statistical analysis conducted. A very simple example is line 328: "Mineral N concentrations were greater at AR1 compared to AR2". In the scientific literature, such a statement implies a significant statistical difference in mineral N concentrations between the two sites. If a statistical test was conducted, a P value should be included (in general, there should be much more inclusion of p values in the text). If it wasn't, small changes to the language need to be made that make that clear, e.g., "Mineral N concentrations appeared to be greater at AR1 compared to AR2". Another small example might be line 334-35: "There was variation at depth in the soil, which could not be explained by fertilization." The phrase "could not be explained" implies that a statistical test was attempted, but was not significant. If, instead, this is qualitative interpretation of data, it should include more qualifying language, e.g., "There was variation at depth in the, which did not appear to be related to fertilization."

In fact, there are numerous instances of this kind of presentation of results throughout the manuscript; here are just a few examples (there are many more involving site comparisons, temporal comparisons within sites, fertilizer effects, etc., and all should be addressed):
L316: "There were only minor differences. . . between seasons" implies that seasonal differences were tested; can be changed to something like "If there were any differences, they were likely minor. . ."
L 326: "The residence time for mineral N in the soil solution was generally longer at AR compared to the RG sites" implies that site differences were tested; can be changed to

"The residence time for mineral N in the soil solution generally appeared to be longer at AR compared to the RG sites"

L328: "Mineral N concentrations were greater at AR1 compared to AR2" implies that site differences were tested; can be changed to "It appeared that mineral N concentrations may have been greater at AR1 than AR2"

L 329: "Fertilisation increased NH4+-N and NO3- -N concentrations" implies that differences between fertilized and unfertilized split plots were tested. Can be changed to "It appeared as though NH4+-N and NO3- -N concentrations increased following fertilization"

One quick way around this issue would be to state in the methods section that all interpretations of results are qualitative unless accompanied by a p value, and then to include p values where statistical tests were conducted (in any event, I encourage the authors to include P values for all statements reflecting the results of a statistical test). But a clearer solution would be 1) to test differences statistically if they have not been and include p values, and 2) to change the language describing results to make it qualitative, using words and phrases such as "may," "appears to," "might," etc. to describe apparent patterns that cannot be tested or aren't important enough to test. I don't think it's necessarily a problem to present results qualitatively (though for any results that are important to the central hypotheses being tested, it's always good to include statistical analyses), but it needs to be very clear when differences were tested and when they were not.

*Response: Regrettably, with limited resources we had made the decision to not collect and analyse soil samples at the block level, but instead pool soil samples from each block (by depth interval) for analysis of mineral N. For this reason, we have been unable to formally test effects of fertilisation or temporal dynamics.*

  *We agree with the reviewer that the text should have reflected this more clearly. We have adjusted wording throughout the manuscript and further stress the qualitative nature of the mineral N dynamics with the following sentence:*

"Since subsamples were pooled for analysis, only a qualitative description of the effects of treatments and temporal dynamics is possible." (l. 346 in cleaned manuscript version)

3. I think there could be more detail in the discussion of figures 3-7, which represent a lot of the data presented in this manuscript. Briefly, the text often invokes a relationship between the water table and a) N2O concentrations in the soil or b) N2O fluxes at the surface, but the relationships described in the text are not obvious to me from examining the figures—the spatiotemporal relationship between the water table depth and N2O concentrations seems quite different in different sites and seasons. Additional text walking the reader through the authors' interpretation of these figures would be helpful. I made related specific comments in my last review, and detail some points below in my response to the authors' response #46, as well as in specific comments for lines 494, 501, and thereabouts.

*Response: In the presentation of results, we gave priority to the comparison of land uses at the four sites, which is why fluxes and subsurface N2O concentrations (and seasons) are presented separately. We have revised the description of these results to better link observations to specific treatments or days, and to better link fluxes with subsurface N2O dynamics.*

  *However, due to the delay in gas exchange through saturated and/or highly tortuous peat soil (a point now stressed in the Discussion l. 489f* in cleaned manuscript version*), a close relationship between fluxes*

*and subsurface concentrations can not be expected. We have added a new paragraph to the discussion explaining this (see below).*

4. It seems to me that the two conclusions that can be clearly reached from this study are the rejection of the hypothesis that that FeS2 oxidation coupled with NO3- reduction was an important driver of N2O emissions, and the identification of the capillary fringe as an important predictor of surface N2O fluxes. If the authors agree, the conclusion should be revised; as it currently stands, the second of these findings is not discussed at all in the conclusion, and the description of the first conclusion gives the reader the impression that the hypothesized mechanism is possibly and maybe even probably not trivial.

*Response: N mineralisation from decomposing peat after WT drawdown was mentioned, but we now specifically refer to the capillary fringe. With respect to FeS2, we have strengthened the wording:*

"The hypothesis that NO3- reduction coupled with FeS2 oxidation was an important source of N2O could not be confirmed." (l. 603 in cleaned manuscript version)

*Also, the Discussion now includes a proper reference to the laboratory study confirming this conclusion – this study has now been published (doi.org/10.1080/01490451.2019.1666192).*

Minor comment: the response to reviewers letter was sloppy – a number (maybe all?) of the line numbers of revised text referred to in the response to reviewers are incorrect, there are typos in the quoted text revisions (e.g., response 50), and sometimes there's no indication of what revisions were made or where they can be found in the text (e.g., response 49). I emphathize with these kinds of errors, especially after getting a fresh batch of comments from a new reviewer late in the publication process, but anything you can do to make the job of the reviewers easier is appreciated.

*Response: An annotated copy with tracked changes was in fact submitted together with the revision, and we specifically referred to in the response letter. The critical comments above therefore appears to be based on a misunderstanding.*

Responses to specific author responses
Response 20: I think the authors missed my point here, which was that the manuscript seems to only conduct a qualitative analysis of the relationship between changes in water table depth and changes in surface N2O flux, rather than including a statistical analysis of the relationship between the two variables that could provide quantitative insight into the importance of variation in water table depth for N2O surface fluxes. I agree that the manuscript does provide a graphical analysis showing that capillary fringe N2O is the only significant predictor of N2O surface fluxes at several sites, but being statistically significant and being important are not necessarily the same thing, and the graphical analysis does not include any analysis of the relationship between water table depth per se and N2O. I don't think that the authors need to do what I am suggesting here for the manuscript to be publishable, but it seems like a missed opportunity.

*Response: We did not test possible direct effects of WT depth on N2O flux. However, WT depth was indirectly, in the analysis of N2O flux patterns with graphical models, where N2O concentration above the WT table carried information about WT depth at the time of sampling. As pointed out by the reviewer in a comment to the earlier Response no. 46 (see below), in early autumn the N2O fluxes were greatly*

*stimulated at a time where WT depth was still low, presumably due to wetting of the upper soil layers in which nitrate had accumulated. This exemplifies that WT depth per se may not be a strong predictor of emissions.*

Response 21: Thank you for the response, though part of my concern was not addressed. This concern is largely related to my general comment about the language used to describe results. My concern here had been that there were multiple instances in the manuscript (I cited them in the original comment as including lines 380 and lines 411-412) where temporal variation in surface N2O flux at a site was attributed to fertilizer effects, or where fertilizer effects were excluded as a cause of variation in surface N2O flux. However, the relationship between temporal variability in surface N2O fluxes and fertilizer applications appears not to have been tested statistically.

*Response: We have analysed the temporal dynamics of N2O emissions, as well as WT depth. Information about statistical significance has been included where relevant.*

Response 25: The authors argue that "The graphical model results (Figure 7) did show increasing N2O emissions with declining, as well as increasing WT depth that depended on soil N status." I do not think that the graphical analysis included a water table depth variable?
A separate point: I also don't understand what is intended by "increasing WT depth that depended on soil N status" but if the point is that N2O_WT depends on soil N status, that would be an interesting result that deserves more discussion. If instead it's referring to AR-autumn, where the N2O flux is determined in part by soil N status, then ignore this second comment of mine.

*Response: This wording was not too well-considered. In fact the effect of WT depth was indirect, by shifting the position and magnitude of soil volume (in the capillary fringe) with a potential for N2O production. It is true, as stated, that we were thinking about the NO3 accumulation and subsequent wetting of the top soil. No changes made.*

Response 39: I'm glad this was caught. I'd encourage the authors to carefully review the script for producing all figures again if they haven't already.

*Response: All Figures were double-checked.*

Response 46: I wonder whether this explanation may also hold for DOY 252 and 259: it looks to me as though there are elevated N2O concentrations in surface soils, and no clear connection to the capillary fringe. Wouldn't that suggest a topsoil source for all of the highest surface fluxes at AR1 during Autumn—a finding supported by the graphical analysis? The text implies that the high WT is responsible for the high emissions for DOY 252 and 259 ("The subsequent decline in N2O emissions at AR sites coincided with WT withdrawal"), but I would think that anoxia in topsoil related to elevated precipitation during this period could be a more likely explanation. The variability in topsoil N2O concentrations looks fairly physically separated from the water table dynamics, and if capillary fringe is not a significant predictor for surface N2O fluxes at this site during autumn, why invoke a relationship between WT withdrawal and surface N2O emission declines?
I have further questions about the interpretation of the subsoil N2O concentration dynamics discussed in my comment on line 494 below.

*Response: We largely agree with the interpretation presented above, and the reference to WT dynamics did not imply a causal relationship. We have modified the text to put more emphasis on the wetting of the soil:*

"High fluxes were observed on the first day of this monitoring period, DOY 246, at a time where WT depth was still at 40 to 80 cm depth. Instead the high $N_2O$ fluxes may have been triggered by saturation of the top soil after 10 and 22 mm rainfall the previous two days. Additional rainfall during the following days then was accompanied by a rise in WT. The subsequent decline in $N_2O$ emissions at AR sites coincided with WT draw-down and drainage of the top soil." (l. 437 in cleaned manuscript version)

Response 54: I think it's a good move to consider $N_2O$ as a time-integrated measure. My previous concern here was very much regarding the limitations of using point measurements of $NH_4+$ and $NO_3-$ to infer N transformation rates. I think the framing of the revised discussion addresses my concerns.

*Response: Thank you.*

Response 56: So the data presented in figure 2 are means (and standard errors) of intact cores containing either fertilized and unfertilized soils in RG1, but for AR1, it's a mean of intact cores containing only unfertilized soils? I don't understand the rationale for presenting the data this way (instead of, for example, presenting fertilized and unfertilized soils separately where appropriate). At the very least, the caption should clearly detail this odd fertilization treatment situation.

*Response: The management of grassland required that fertilisation took place before we had an opportunity to collect intact soil cores at the remote field site. Subsequent inspection of $NO_2-$ results upon sampling one week after fertilisation, however, did not suggest any effect of fertilisation, and this was confirmed by a paired-sample t-test across the six depths. We therefore went ahead and presented soil profiles across both fertiliser treatments.*
*As recommended, we have revised the Figure caption to describe this procedure.*

Response 57: It's your decision, but there isn't much fine-scale temporal variation in these figures; I'd think it's worth graphing up the full year and seeing what it looks like.

*Response: Whether seasons or land use treatments should be shown side by side is not an easy decision. However, since grassland sites, and arable sites, are mostly discussed together, we wanted to facilitate a direct comparison with this side by side presentation. Although a case could also be made for other solutions, we have decided not to change the format of Figures 3-6.*

Response 59: Again, it's your decision, but would think it's worth including in SI as figures as well as data tables—it's just very difficult to read patterns in a data table. But thank you for including the raw data for researchers who may be interested in using it for their analyses.

*Response: Thank you for this suggestion. Given the many variables involved (sites, crops, fertiliser treatments, soil depths and N species), we believe graphic presentations could be either too busy or too disaggregated to have much added value, and therefore we prefer to focus on the Table format.*

New specific comments:

Line 23: perhaps define capillary fringe here: "…in the capillary fringe— [definition here]—was.

. . "

*Response: Done – the following text was added:*

"(i,e, the soil volume above the water table influenced by tension saturation)" (l. 23 in cleaned manuscript version)

Line 62: it would be helpful to readers if you could provide an explicit definition of capillary fringe here, given how important it is to the manuscript.

*Response: text was added (l. 67) to stress that this zone is defined by capillary rise of groundwater, but that it can still be partly unsaturated depending on pore size distribution:*

"The capillary fringe of organic soils represents an interface between saturated and unsaturated soil conditions, in which the extent of tension saturation depends on pore size distribution (Gillham, 1984)." (l. 68 in cleaned manuscript version)

Line 105: what ridges? Probably need to provide more agronomic details

*Response: It has been specified that these are established around potato rows.*

Line 124: Perhaps this should be "method of fertilizer application"? (as N fertilization was noted as an exception in the next sentence)

*Response: OK, has been included to clarify this distinction. (l. 131* in cleaned manuscript version*)*

Line 127: I believe the erroneous NS fertilizer application in RG2 was made on the same date as the second slurry application; if correct I would clarify that fact by starting this sentence with "On the same date as the second slurry . .. " or "Immediately after the second slurry. . . " If the NS application was made on a different date, indicate the date.

*Response: Done.*

Line 137: here it would be helpful to explain the choices of spring and fall as emerging from the patterns observed in Petersen 2012, and in the introduction briefly describe those patterns. It could be just 1-2 sentences in the introduction, and an introductory clause to the first sentence here.

*Response: The following was added to the Introduction, and is referred to in section 2.4, as requested:*

"Among sites with arable crops in three regions, two sites had N2O emissions corresponding to, respectively, 38 and 61 kg N ha-1 yr-1; both of these sites showed distinct seasonal patterns with the highest emissions in spring and autumn periods, whereas emissions at the third site were lower (6.4 kg N2O-N ha-1 yr-1) and much less variable. Notably, WT depth at the two sites with seasonal patterns of N2O emission fluctuated between 10-30 and 10-120 cm depth, whereas WT depth at the third site remained at 90-125 cm depth throughout the 14-month monitoring period." (l. 53 in cleaned manuscript version)

Line 144: it would give a more complete picture if the dates of these exceptions were enumerated here.

*Response: The following was added:*

"With a few exceptions (DOY 169 and 265 at site *RG1*, DOY 132 and 314 at site *AR2*), each campaign was initiated between 9:00 and 12:00; the order of sites visited in each trip alternated from week to week." (l. 151 in cleaned manuscript version)

Line 165: revise to "…at -20C until analysis, described in section 2.4.5."

*Response: Done.*

Line 170-1: I'm afraid this sentence is hard to follow. It reads as though the diluted soil gas was first transferred to the exetainer, and then from the exetainer to the glass syringe. Perhaps also specify whether exetainers were filled with 10ml gas as they were in the chamber measurements.

*Response: Section 2.4.3 has been edited to clarify the procedure.* (l. 176 in cleaned manuscript version)

Line 174: As a general rule I'd argue it's best to explicitly include all applicable methodological detail, rather than referring readers to an additional manuscript (within reason). In this case, my institution does not have a subscription to the European Journal of Soil Science, so my library had to obtain 24-hour access to Petersen 2014 so I could see the dilution calculations; I would include them in the manuscript.

*Response: Equation 5 from Petersen 2014 is now shown and explained in section 2.5.*

Line 240-245: it would be helpful to specify if/how field temperature and pressure corrections were made to obtain surface flux estimates.

*Response: The following sentence was added:*

"Nitrous oxide mixing ratios were converted to units of mass per volume using the ideal gas law and values of pressure and air temperature recorded by the weather station." (l. 257 in cleaned manuscript version)

Line 297: It should be noted here that these characterizations were conducted after the fertilizer treatment of RG1

*Response: We did not locate the text referred to above, but we have assumed that it was soil N characterisation. We have added the following sentence:*

"At site *RG1*, fertilisation had taken place one week earlier, but a two-sample *t* test did not find evidence for an effect on $NO_2^-$ availability ($p = 0.19$), and therefore the results from both *AR1* and *RG1* subplots are presented together." (l. 355 in cleaned manuscript version)

Line 380: Instead of "independent of fertilization" which suggests a formal analysis, how about "this variation appeared to be broadly similar between fertilized and unfertilized subplots"

*Response: OK, paragraph has been revised.*

Line 394 and following: this is one of many examples related to my general comment on presenting qualitative results, but the generalized linear mixed model appears to include only time, and not fertilizer, as a factor in the fixed effect, so statements regarding the effects of fertilization should be made qualitatively (e.g., rather than "with no effect of fertilizer

amendment," use something like "and did not appear to vary in response to fertilization." Similarly, "no effect of N fertilisation was observed. Hence, the higher emissions were associated with site differences other than fertilization" could be changed to "fertilisation did not appear to influence N2O fluxes. Hence, the higher emissions were likely associated with site differences other than fertilization"). The paragraph starting with line 406 does a good job of qualitatively describing results.

*Response: We have revised the text to make sure that results are presented in a qualitative way where necessary.*

Line 467: change to "In accordance with this effect of rewetting…" (in general, it's good to give "this" an object to make it clear what is being referred to)

*Response: Done.*

Line 490: here and maybe elsewhere there are still instances where time of year are referred to without DOY (here parenthetical DOYs would be particularly helpful since its referring to figures using DOY on the x-axis). In addition, please specify that this accumulation is in the fertilized plots of RG2.

*Response: OK, several instances were amended with DOY information.*

Line 491: change "significant" to "substantial" if the meaning is "a lot." Avoid using "significant" except when alluding to the result of a statistical test.

*Response: Done.*

Line 492: if the contrast is that in RG2 there was only accumulation at certain times and depths but accumulation was everywhere and all the time in the AR sites, change "..accumulation of N2O in the soil" to "…accumulation of N2O across all soil depths throughout the spring" or something similar. Maybe include a reference to Figure 4.

*Response: OK, the following wording is now used:*

*"In contrast, at AR sites N$_2$O consistently accumulated in the soil throughout spring" (l. 536 in cleaned manuscript version)*

*Line 494: as was the case with the previous version of the manuscript (see my original comment that elicited the authors' response 46 and my general comment 3), I'm not sure I see the pattern so clearly. The location of elevated N2O above 40cm in AR2 is attributed to the water table being higher than in AR1, where elevated N2O tends to be in deeper soils, and the authors argue that this points to a capillary fringe source for N2O. But in RG2 the water table is higher than it is in RG1 during the spring, but the N2O distributions are reversed: they are higher at depth in in RG2, where the water table is higher, and higher in shallower soils in RG1, where the water table is lower. The fertilized AR2 also has an inversion of N2O concentrations during spring that seems entirely unrelated to water table depth. In autumn, the N2O at depth declines as the water table depth increases, and the patterns in the two blocks in RG1 seem somewhat opposite of one another even though the water table is just about identical. With so much apparent variation in how N2O concentrations vary with the water table, how do these results all

relate to capillary fringe as the N2O source? What am I missing? I suppose that this comment, in combination with my new response to the authors response 46, suggests an opportunity for the authors to revise and expand their presentation (and perhaps interpretation) of these results. It would be a very satisfying read if the paper spent a little more time on the subtleties in the links the graphical analysis identification of the significant drivers of surface N2O fluxes to the spatiotemporal dynamics presented in figures 3-6 (these are most of the data being presented in the paper, so feel free to give them more attention in the text).

*Response: We agree that patterns of soil N2O concentration profiles were complex and not readily aligned with emissions. It is probably important to acknowledge the importance of soil wetness as a barrier towards transport, and we have added a new paragraph to the discussion (section 4.1) to motivate the focus on N2O concentration above the water table, which was a potential site of N2O production:*

"A limited number of potential drivers (rainfall, temperature, soil mineral N and soil concentrations of N2O) were monitored to help explain N2O emission dynamics. Soil N2O concentration profiles showed complex patterns where, for example, the highest concentrations were sometimes observed above, and sometimes below the WT depth at both RG (Figure 3) and AR sites (Figure 4). Fertilisation in spring was associated with higher concentrations of N2O below the WT depth at sites RG2 and AR2, which indicated downward transport of fertiliser N, but this was not reflected in elevated N2O emissions. The reason may be that in wet soil the time required to reach a steady state between N2O production and emissions from the soil surface can be significant and increases with distance (Jury et al., 1982). In accordance with this, Clough et al. (1999) observed a delay of 11 days before 15N enriched N2O produced at 80 cm depth was released from the soil surface at a corresponding rate. Since the presence of air-filled porosity is critical for the exchange of gases between soil and the atmosphere (Jury et al., 1982), the soil N2O concentration closest to, but above the WT depth (N2OWT), was taken to represent "subsoil" processes stimulating N2O emissions." (l. 484 in cleaned manuscript version)

Line 501: I'm not sure why water table dynamics are invoked in the discussion of N2O emissions at AR1, if capillary fringe was not a significant predictor of surface N2O fluxes in the graphical analysis, and, as noted previously (and in my comment on the authors' response 46), surface fluxes were already elevated before the rise in WT depth? A simpler explanation consistent with the graphical analysis might be that topsoil was the source of surface N2O fluxes in AR1 during autumn, apparently independent of water table depth. If this discussion is intended to be descriptive only of dynamics within the soil (and does not have any relation to surface N2O emissions), that needs to be made clear.

*Response: We agree, and we have adjusted the text accordingly, as explained above.*

Line 528: It would be helpful to have a definition of "dead-end pores"

*Response: These are associated with plant cell remains. This information has been added.*

Line 567-559: These are all the kinds of statements that need to be made much more qualitatively in the absence of clear statistical support, and issues with experimental design and strength of statistical support for conclusions need to be highlighted in the conclusion.

*Response: The Conclusion section has been rewritten to reflect where statistical support for statement was available, and to reflect better this is our interpretation of the results.*

Line 560: remove the comma between "hypothesis" and "that"

*Response: Done – and sentence edited.*

Line 561: my impression from the results is that NO3- reduction coupled with FeS2 reduction is likely to be a trivial mechanism of N2O production in these soils (a conclusion apparently supported by the related manuscript that has been submitted elsewhere)—this statement makes it sound as though it could be non-trivial.

*Response: We agree that this conclusion should be expressed more clearly. Has been rephrased as:*

"The hypothesis that NO3- reduction coupled with FeS2 oxidation was an important source of N2O was not confirmed." (l. 603 in cleaned manuscript version)

[revised manuscript text omitted]
* sites (Figure 2), which was consistent with peat decomposition and ammonia oxidation following WT drawdown. Total concentrations of $NH_4^+$ and $NO_2^-$ at 25-50 cm depth were significant (Tables S1 and S2), but well-decomposed peat is dominated by dead-end pores (Hoag and Price, 1997), and it is likely that ammonia oxidation to a large extent took place in such pores having a slow exchange of solutes with active pore volumes. The accumulation of $NO_2^-$ suggested there was an imbalance between ammonia oxidation and nitrite oxidation activity.in this study.~~, which can explain the correlation between $N_2O_{WT}$ and $N_2O$ emissions in graphical models. In late April (DOY 113), $NO_2^-$ had accumulated at 20-50 cm depth at both *RG1* and *AR1* sites (Figure 2), suggesting that there was an imbalance between ammonia oxidation and nitrite oxidation activity. Oxygen affinity differs between nitrifiers, with AOA>AOB>NOB (Yin et al., 2018), and

620 hence $O_2$ limitation could  have caused the accumulation of $NO_2^-$. In acid soil, this would result in product inhibition by $HNO_2$ if there were no mechanism to remove $NO_2^-$, this would be especially true for *AR* sites, where mineral N accumulation was three to four times higher compared to *RG* sites (Tables S3-S6). Nitrifier-denitrification is a mechanism by which ammonia oxidisers can avoid $HNO_2$ accumulation, and this process leads to $N_2O$ formation (Braker and Conrad, 2011). Another potential sink for $NO_2^-$ is chemodenitrification, an abiotic reaction in which $NO_2^-$

625 reacts with $Fe^{2+}$ to produce $N_2O$ (Jones et al., 2015):

$$4Fe^{2+} + 2NO_2^- + 5H_2O \rightarrow 4FeOOH + N_2O + 6H^+ \tag{34}$$

where in Eq. 4 $Fe(OH)_3$ is shown as anhydrous FeOOH. A possible depletion of TRFe was indicated at 50 cm depth at site *AR1*, which coincided with a similar pattern for $NO_2^-$ (Figure 2). Nitrifier-denitrification and chemodenitrification are both sinks for $NO_2^-$, and therefore both pathways are potential sources of the $N_2O$ emissions
630 observed during early spring.

At *AR* sites there was often considerable accumulation of $N_2O$ below the WT, which suggests there was also an anaerobic pathway of $N_2O$ formation. The fact that TRFe concentrations were much higher than those of AVS or CRS (Table 1) makes it relevant to consider alternative reactions involving iron oxides/hydroxides, which have a potential to produce $N_2O$. One such recently described pathway is Feammox, a process whereby ammonia
635 oxidation coupled with ferric iron reduction can produce $NO_2^-$ below pH 6.5 (Yang et al., 2012):

$$6Fe(OH)_3 + 10H^+ + NH_4^+ \rightarrow 6Fe^{2+} + 16H_2O + NO_2^- \tag{45}$$

Nitrate can also be produced under these conditions (Yang et al., 2012; Guan et al., 2018):

$$8Fe(OH)_3 + 14H^+ + NH_4^+ \rightarrow 8Fe^{2+} + 21H_2O + NO_3^- \tag{56}$$

A shuttle of $Fe^{2+}$ between Feammox and chemodenitrification (Eq. 5 and Eq. 4) could account for the
640 accumulation of $N_2O$ under anoxic conditions in the saturated zone,  presumably with the availability of $NH_4^+$ from peat mineralisation as a limiting factor. The confirmation of pathways will require more detailed investigations that should include molecular analyses targeting microbial communities in the soil profile.

**5 Conclusion**

645 In this one-year study, $N_2O$ emissions were consistently higher from arable sites compared to rotational grass. There were strong seasonal dynamics in $N_2O$ emissions, and we present evidence that different pathways were involved. Concentrations of pyrite were low compared to the total reduction capacity of the peat, and Fe was predominantly in forms other than pyrite. The
650 hypothesis that  $NO_3^-$ reduction coupled with $FeS_2$ oxidation was an important source of $N_2O$ could therefore not be confirmed. Nitrous oxide emissions

that oxidation of N mineralised from decomposing peat after WT drawdown in during spring were independent of fertilisation, since there was followed by mostly no effect of mineral N in the top soil. The significant effect of $N_2O$ concentration in the capillary fringe indicated that emissions during spring, and for grassland during the autumn, were associated with soil N mineralisation in this environment, as modified by rainfall patterns and WT dynamics. We propose that chemodenitrification (or nitrifier-denitrification),) of $NO_2^-$ produced in the capillary fringe is a main source of $N_2O$ in acid organic soil during spring, whereas in the autumn, where $NO_3^-$ accumulated in arable soil after harvest, $N_2O$ emissions were associated with rising WT and heterotrophic denitrification as thecan 
[revised manuscript text omitted]

a.

[Figure]

b.

Figure 1

[Figure]

**Figure** 2.

[Figure]

[Figure]

**Figure** 3̶2̶.

[Figure]

**Figure 3**

[Figure]

Commented [SOP3]:
AR2: 71 d vlærdi mangler (21 cm)

**Figure 4**

[Figure]

**Figure 5**

[Figure]

**Figure 6**

[Figure]

a.

[Figure]

b.

[Figure]

c.

[Figure]

d.

Commented [SOP4]: Udskiftes

[Figure]

**Figure 7**

[Figure]

[Figure]

[Figure]

[Figure]

**Figure 8**

---

## Author Response (AR4)

Dear Dr. Park

On behalf of my coauthors, I would like to thank you for accepting the manuscript. We are grateful to you and anonymous reviewer's comments that help us improve our work scientifically.

I have added the word denitrification in the abstract.

Please find below our responses to the comments below, and also the manuscript with track changes.

With kind regards
Arezoo Taghizadeh-Toosi
Researcher
Aarhus University

Associate Editor Decision: Publish subject to minor revisions (review by editor) (14 Oct 2019) by Ji-Hyung Park

Comments to the Author:

Dear Authors,

Thank you for your efforts to conduct additional statistical analyses and prepare detailed responses to the reviewer comments on your revised manuscript.

I am pleased to let you know that your manuscript can be published after some minor revisions (mostly technical corrections) as detailed below.

- Abstract, lines 20-22: Please provide "ranges" (either XX-XX or from XX to XX).

Response: We have added the ranges and text is read as: "*Equivalent soil gas phase concentrations of N2O at grassland sites varied between 0 and 25 µL L-1 except for a sampling after slurry application at one of the sites in spring with a maximum of 560 µL L-1 at 1 m depth. At the two potato sites the levels of below-ground N2O concentrations ranged from 0.4 to 2270 µL L-1, and from 0.1 to 470 µL L-1, respectively, in accordance with the higher soil mineral N availability at grassland sites.*".

- Abstract, lines 30-31: This should be a concluding (or at least summarizing) remark. Please specify what "possible pathways of N2O production in spring and autumn periods" you would like to highlight.

Response: The summarizing remark has been added as : " *Based on the statistical graphical models and the tentative estimates of reduction capacities, FeS2 oxidation was unlikely to be important for N2O emissions. Instead, archaeal ammonia oxidation in combination with either chemodenitrification or nitrifier denitrification were considered as plausible pathways of N2O production in spring, whereas in the autumn heterotrophic may have been more important at arable sites.*".

- Conclusions, line 606: for grassland "also" during the autumn?

Response: "also" has been added.

- Table 2: Please describe in the table caption what test was used and what significant differences (between what?) different letters represent.

Response: Table 2 caption has been updated as it requested.

- Fig. 1: To guide readers on the location of your study sites in Demark or Europe, please add more geographical information (e.g., any close city name, cardinal directions,,,) on the main site map

Response: We have added the map of Denmark as well as indicating of the study place. Copenhagen as a capital has been marked in the map as well.

[revised manuscript text omitted]

a.

b.

b.

a.

b.

[Figure]

Figure 1

[Figure]

**Figure 2.**

[Figure]

**Figure 3**

[Figure]

**Figure 4**

[Figure]

**Figure 5**

[Figure]

**Figure 6**

[Figure]

**Figure 7**

[Figure]

**a**. RG1-Autumn

[Figure]

**c**. AR1-Autumn

[Figure]

**b**. RG2-Autumn

[Figure]

**d**. AR2-Autumn

[Figure]

**a**. RG1-Autumn

[Figure]

**c**. AR1-Autumn

[Figure]

**b**. RG2-Autumn

[Figure]

**d**. AR2-Autumn

**Figure 8**